# DISTRIBUTIONALLY ROBUST OPTIMIZATION VIA GENERATIVE AMBIGUITY MODELING

**Jiaqi Wen and Jianyi Yang** [*]
Department of Computer Science
University of Houston

## ABSTRACT

This paper studies Distributionally Robust Optimization (DRO), a fundamental framework for enhancing the robustness and generalization of statistical learning and optimization. An effective ambiguity set for DRO must involve distributions that remain consistent to the nominal distribution while being diverse enough to account for a variety of potential scenarios. Moreover, it should lead to tractable DRO solutions. To this end, we propose generative model-based ambiguity sets that capture various adversarial distributions beyond the nominal support space while maintaining consistency with the nominal distribution. Building on this generative ambiguity modeling, we propose DRO with Generative Ambiguity Set (`GAS-DRO`), a tractable DRO algorithm that solves the inner maximization over the parameterized generative model space. We formally establish the stationary convergence performance of `GAS-DRO`. We implement `GAS-DRO` with a diffusion model and empirically demonstrate its superior Out-of-Distribution (OOD) generalization performance in ML tasks. [1]

## 1 INTRODUCTION

Distributionally Robust Optimization (DRO) is a fundamental framework for enhancing the robustness of statistical learning and optimization problems, particularly under Out-of-Distribution (OOD) scenarios Liu et al. (2021); Arjovsky (2020); Ren & Majumdar (2022). DRO formulates a minimax optimization problem, where the inner maximization identifies the worst-case distribution within an ambiguity set, and the outer minimization optimizes the decision variable against this worst-case scenario Chen et al. (2020); Liu et al. (2022b); Chen & Paschalidis (2018); Kuhn et al. (2019); Smirnova et al. (2019). Unlike non-probabilistic robust optimization, DRO leverages probabilistic ambiguity modeling to enable improved generalization performance. This property has made DRO increasingly important in statistical learning and optimization tasks for addressing distribution shifts, noisy data, and adversarial conditions Quiñonero-Candela et al. (2022); Muratore et al. (2022); Ren & Majumdar (2022); Li et al. (2019); Veit et al. (2017).

The performance of DRO algorithms significantly depends on the design of the ambiguity set, which must contain diverse distributional variations around the nominal distribution. A common approach is to model the ambiguity set using $\phi$-divergences, such as the Kullback–Leibler (KL) divergence Hu & Hong (2013); Husain et al. (2023); Kirschner et al. (2020); Kocuk (2020); Liu et al. (2022b). Although such $\phi$-divergence-based DRO formulations can sometimes yield closed-form solutions Hu & Hong (2013); Husain et al. (2023), they require any distribution $P$ in the ambiguity set to be *absolutely continuous* with respect to the nominal distribution $P_0$ (denoted $P \ll P_0$), meaning that for any measurable set $\mathcal{A}$, if $P_0(\mathcal{A}) = 0$, then $P(\mathcal{A}) = 0$. This implicit constraint limits robustness of DRO in scenarios with support shifts Staib & Jegelka (2019); Lu et al. (2024); Gao & Kleywegt (2023a).

While the ambiguity set defined by Wasserstein distance allows for support shifts, solving Wasserstein-DRO over the infinite probability space is challenging and computationally inefficient. Some approaches Mohajerin Esfahani & Kuhn (2018); Chen et al. (2020); Gao & Kleywegt (2023b)

---

[*]Correspondence to Jianyi Yang {jyang66@uh.edu}

[1]Source Code Link: https://github.com/CIGLAB-Houston/GAS-DRO

reformulate Wasserstein-DRO as a finite-dimensional optimization problem based on the convex assumptions which typically do not hold in deep learning. Other methods approximate Wasserstein-DRO via adversarial optimization Staib & Jegelka (2017); Gao & Kleywegt (2023b); however, these relaxations are often overly conservative, which restricts their ability to exploit probabilistic modeling for effectively balancing average-case performance with worst-case robustness.

Recent advances have sought to address the challenges of ambiguity modeling in DRO. Some works Wang et al. (2025a); Yang et al. (2025); Blanchet et al. (2023) study DRO using ambiguity sets based on the Sinkhorn distance, which preserve computational tractability and enhance the expressiveness of adversarial distributions. Generative models have also been utilized in the traditional ambiguity modeling frameworks. Among them, Ren & Majumdar (2022) construct a Wasserstein-based ambiguity set in the latent space of generative models. Michel et al. (2021; 2022) exploit parameterized generative models to represent the adversarial distributions or the likelihood ratios within the KL-divergence-based ambiguity modeling framework and derive tractable solutions with additional approximations. Nevertheless, the traditional ambiguity modeling frameworks still introduce tractability issues or restricts the support space of adversarial distributions, limiting the utilization of the distributional representation capacity of generative models.

To the best of our knowledge, this is the first work to model the ambiguity sets in the parameterized space of generative models [2] [3], such as diffusion models and Variational Autoencoders (VAEs), which offers several advantages: (1) Generative models have a strong capability to represent the underlying data distribution, ensuring that the distributions in the ambiguity set remain consistent with the nominal distribution. (2) Generative models are capable of producing diverse samples beyond the nominal support space, thereby enabling the discovery of distributions with worst-case optimization performance. (3) Generative models provide a finite, parameterized optimization space, avoiding the need to solve problems over an infinite probability space.

Our main contributions are summarized below: **(1)** Based on the probability modeling of generative models, we introduce a novel *Generative Ambiguity Set* (GAS) for DRO which involves diverse distributions while preserving consistency with the nominal distribution. While GAS is introduced based on the formulations of diffusion models and VAEs, it can be flexibly adapted to other likelihood-based generative models. **(2)** We design `GAS-DRO` (Algorithm 1), which gives a direct solution to DRO with the proposed Generative Ambiguity Sets. We solve the inner maximization in `GAS-DRO` based on dual learning and policy optimization techniques, enabling tractable iterative optimization within a finite, parameterized space. **(3)** We prove in Theorem 1 that the inner maximizer of `GAS-DRO` converges to the optimal maximization oracle and, asymptotically, generates distributions consistent with the nominal distribution. Building on this inner-maximization error bound, Theorem 2 further establishes the stationary convergence of `GAS-DRO`. **(4)** We implement `GAS-DRO` on series prediction and image classification tasks and demonstrate its superiority by comparing it against state-of-the-art DRO baselines based on generative models.

## 2 RELATED WORK

**Distributionally Robust Optimization**. The performance of DRO relies largely on the design of ambiguity sets. Tractable DRO solutions can be obtained based on the ambiguity sets defined by $\phi$-divergence Husain et al. (2023); Kirschner et al. (2020); Hu & Hong (2013); Kocuk (2020). However, the definition of $\phi$-divergence requires any distribution $P$ in the ambiguity set to be absolutely continuous with respect to the nominal distribution, which can cause support mismatch problems Staib & Jegelka (2019). By contrast, Wasserstein-distance-based ambiguity set Kuhn et al. (2019); Mohajerin Esfahani & Kuhn (2018); Gao & Kleywegt (2023b) has no restrictions on the support of the distribution, but it introduces much computational difficulty. Wasserstein-based DRO is solved by either conservative relaxations Sinha et al. (2017); Staib & Jegelka (2017) or reformulations based on some assumptions on the objective Kuhn et al. (2019); Mohajerin Esfahani & Kuhn (2018); Chen et al. (2020); Gao & Kleywegt (2023b).

---

[2]This claim means designing new ambiguity sets that directly constrain the parameters of generative models without replying on traditional distributional discrepancy measures.

[3]We design concrete algorithms with likelihood-based generative models that satisfy the constraints on inclusive KL divergence in Lemma 1, but the proposed framework can employ other generative models which potentially provide other guarantees on consistency with the nominal distribution.

A line of recent studies focuses on addressing the challenges in ambiguity modeling for DRO Xu et al. (2024); Zhu & Xie (2024); Ren & Majumdar (2022); Liu et al. (2022a); Yang et al. (2025). Among them, Liu et al. (2022a); Wang et al. (2025c) inject prior knowledge into the design of ambiguity sets to mitigate excessive conservativeness. Ma et al. (2024) develop an end-to-end framework in which an ML model is trained to predict the ambiguity set for downstream DRO tasks. Moreover, recent studies Wang et al. (2025a); Yang et al. (2025); Blanchet et al. (2023) investigate DRO with Sinkhorn-distance-based ambiguity set which includes continuous adversarial distributions and preserves tractability. Different from these methods, we utilize the strong distribution learning capability of generative models to build the ambiguity set, enabling the discovery of worst-case and realistic distributions. Meanwhile, the proposed algorithm `GAS-DRO` converts DRO into a finite tractable problem in the parameterized space of generative models.

**Generative Models for Robust Learning**. While generative models are typically used for generative tasks Zhang et al. (2024); Li et al. (2025); Chen et al. (2024), they have also been used for (robust) discriminative or classification tasks Li et al. (2023); Chen et al. (2023); Tong et al. (2025). A line of works use generative models to synthesize adversarial samples for robust training Dai et al. (2024b;a); Xie et al. (2024); Du et al. (2022). The target of these works is to generate adversarial attacking examples which is fundamentally different from the worst-case distribution generation which is studied in this paper and aims to improve OOD generalization. Recently, Wang et al. (2025b) introduced a Generate-then-Optimize framework, where a diffusion model is trained to generate data for downstream statistical optimization with a focus on the conditional value-at-risk (CVaR) objective.

Prior work has utilized generative models to represent adversarial distributions within traditional ambiguity modeling frameworks. For example, DRAGEN Ren & Majumdar (2022) models adversarial distributions by a generative model. However, this approach remains within the Wasserstein-based DRO framework and continues to suffer from intractability and relaxation-related issues. In addition, Michel et al. (2021) represent adversarial distributions using generative models within a traditional KL-divergence-based ambiguity set and derive an approximated tractable objective. Similarly, Michel et al. (2022) employ generative models to parameterize the likelihood ratio between the adversarial and nominal probability densities. However, both Michel et al. (2021) and Michel et al. (2022) remain within the KL-divergence-based DRO framework, which requires the adversarial distributions to share the same support as the nominal distribution. This constraint restricts the expressive power of generative models in constructing diverse adversarial distributions and ultimately limits the robustness performance. Different from these methods, our generative ambiguity modeling overcomes the limitations of traditional ambiguity sets: It provably constrains the reverse KL divergence that does not restrict the support space of adversarial distributions and enables flexibly tractable solutions through policy-optimization techniques.

Our work is also related to studies on generative models for worst-case environment generation. In this line of research, Berdica et al. (2024) select adversarial world models from a set of world models trained on a single dataset. In addition, Hill (2025) conduct an experimental study on learning adversarial world models aimed at defeating defender agents in multi-agent reinforcement learning. Moreover, Cheng et al. (2025) learn a transport map for out-of-sample generation within the Wasserstein space and establish convergence guarantees.

## 3 AMBIGUITY MODELING IN DRO

**Distributionally Robust Optimization.** DRO optimizes for the worst-case performance given an ambiguity set constructed on a nominal distribution $P_0$. An empirical dataset $S_0$ can be drawn from the nominal distribution. Consider an objective function $f(w, x)$ with the decision variable $w \in \mathcal{W}$ and the random parameter $x \in \mathcal{X} \in \mathcal{R}^d$. Given the nominal distribution $P_0(x)$ of the random parameter $x$, DRO solves the following minimax optimization problem.

$$\min_{w \in \mathcal{W}} \max_{P \in \mathcal{B}(P_0, \epsilon)} \mathbb{E}_{x \sim P}[f(w, x)] \tag{1}$$

where $\mathcal{B}(P_0, \epsilon)$ is the ambiguity set containing possible testing distributions and is typically modeled as a distribution ball $\mathcal{B}(P_0, \epsilon) = \{P \mid D(P, P_0) < \epsilon\}$ given a distribution discrepancy measure $D$ and an adversary budget $\epsilon$.

**Challenges of Ambiguity Modeling in DRO.** The choice of the ambiguity set in DRO has a significant impact on both generalization performance and solution tractability. An effective ambiguity set should satisfy three key properties. First, it should capture a broad range of distributions with different support spaces, enabling the identification of worst-case scenarios. Second, the distributions within the ambiguity set should remain consistent with the nominal distribution since inconsistent distributions can result in overly-conservative DRO solutions with poor average-case performance. Finally, the ambiguity set should enable a tractable DRO solution despite the complexity of the infinite probability space in DRO.

The ambiguity sets based on classic distribution discrepancy measures do not satisfy one or more of the above properties. Ambiguity sets based on $\phi$-divergences Hu & Hong (2013); Husain et al. (2023); Kocuk (2020); Liu et al. (2022b) requires any distribution $P$ in the ambiguity set to share the same support $\mathcal{X}$ as the nominal distribution $P_0$, i.e. $P$ is absolutely continuous with $P_0$ ($P \ll P_0$). This implicit constraint will restrict the search of worst-case distributions and limit the robustness under testing cases with support shift. In addition, Wasserstein-based ambiguity set Mohajerin Esfahani & Kuhn (2018); Chen et al. (2020); Gao & Kleywegt (2023b); Staib & Jegelka (2017); Gao & Kleywegt (2023b) introduces too much complexity in solving the inner maximization over an infinite probability space. The relaxation of the Wasserstein-based ambiguity set can include a lot of invalid distributions that are not consistent with the nominal distribution $P_0$, leading to overly-conservative DRO solutions.

We observe that the challenge of ambiguity modeling stems from a fundamental tension between expressiveness and tractability. When probability distributions are restricted to the same support as the nominal distribution, as in $\phi$-divergence-based ambiguity sets, robustness performance is often degraded due to limited expressiveness. In contrast, when no restrictions are imposed on the support space, as in Wasserstein-based ambiguity sets, the resulting DRO is not easily tractable because of the infinite-dimensional probability space. This observation motivates us to leverage the strong distribution modeling capabilities of generative models to construct ambiguity sets over a finite, parameterized distribution space, with the goal of achieving both expressiveness and tractability.

## 4 AMBIGUITY MODELING BASED ON GENERATIVE MODELS

### 4.1 PROBABILITY MODELING VIA GENERATIVE MODELS

The foundation of the generative ambiguity sets is the probability modeling based on generative models. Here, we representatively introduce the probability models of diffusion models and VAEs since they are both widely-used generative models.

**Diffusion Models**. The diffusion models learn data distributions through a forward process and a reverse denoising process which are introduced based on the formulations of variational diffusion models Ho et al. (2020); Luo (2022).

*Forward Process.* The forward process in a diffusion model begins with an initial sample $x_0 \in \mathcal{R}^d$ drawn from the nominal distribution $P_0$, and evolves according to a Markov chain that gradually adds Gaussian noise to the data:

$$q(x_{1:T} \mid x_0) := \prod_{t=1}^{T} q(x_t \mid x_{t-1}), \ q(x_t \mid x_{t-1}) = \mathcal{N}(x_t; \sqrt{\alpha_t} \, x_{t-1}, (1 - \alpha_t)\mathbf{I}) \qquad (2)$$

where $\alpha_t \in (0, 1)$ is variance schedule at step $t$. With a slight abuse of notation, we also write the nominal distribution $P_0$ as $q(x_0)$. A sample of the forward process at step $t$ can be written as $x_t = \sqrt{\bar{\alpha}_t} \, x_0 + (1 - \bar{\alpha}_t)\nu$, where $\nu$ is a standard Gaussian noise and $\bar{\alpha}_t = \prod_{\tau=1}^{t} \alpha_\tau$.

*Reverse Process.* By reversing the forward process, a diffusion model learns to recover the original data distribution starting from $x_T \sim p'$ where $p'$ is a prior distribution usually selected as a standard Gaussian distribution. The reverse process can be represented by transitions parameterized by $\theta$:

$$P_\theta(x_{0:T}) = p'(x_T) \prod_{t=1}^{T} p_\theta(x_{t-1} \mid x_t), \ p_\theta(x_{t-1} \mid x_t) = \mathcal{N}(x_{t-1}; \mu_\theta(x_t, t), \Sigma_\theta(x_t, t)) \qquad (3)$$

where $\mu_\theta(x_t, t)$ and $\Sigma_\theta(x_t, t)$ are parameterized mean and variance.

The loss function to train the diffusion model is derived to guide the reverse process to approximate the forward process. It holds by Bayesian's rule that the denoising transition corresponding to the forward process is $q(x_{t-1} \mid x_t, x_0) = \mathcal{N}(x_{t-1}; \mu_q(x_t, t), \sigma_t \mathbf{I})$ where $\mu_q(x_t, t) = \frac{1}{\sqrt{\alpha_t}} x_t - \frac{1-\alpha_t}{\sqrt{1-\bar{\alpha}_t}\sqrt{\alpha_t}} \nu$ with $\nu$ being the standard Gaussian noise to sample $x_t$ and $\sigma_t = \frac{(1-\alpha_t)(1-\bar{\alpha}_{t-1})}{1-\bar{\alpha}_t}$. Thus, to match the denoising mean and variance, we express the parameterized denoising mean as $\mu_\theta(x_t, t) = \frac{1}{\sqrt{\alpha_t}} x_t - \frac{1-\alpha_t}{\sqrt{1-\bar{\alpha}_t}\sqrt{\alpha_t}} s_\theta(x_t, t)$, where $s_\theta(x_t, t)$ is a ML model parameterized by $\theta$ to predict the noise that determines the mean of $x_{t-1}$ from $x_t$, and $\Sigma_\theta(x_t, t) = \sigma_t^2 \mathbf{I}$. It can be proved that maximizing the log probability $\log P_\theta(x_0)$ averaged by $P_0$ is equivalent to minimizing the loss below

$$J_{\mathrm{DM}}(\theta, P_0) = \mathbb{E}_{x_0, \nu_{1:t}} \left[ \sum_{t=1}^{T} \iota_t \left[ \|\nu_t - s_\theta(x_t, t)\|_2^2 \right] \right] \tag{4}$$

where $x_t = \sqrt{\bar{\alpha}_t} x_0 + (1 - \bar{\alpha}_t)\nu_t$ with $\nu_t$ being a standard Gaussian noise, and $\iota_t = \frac{1}{2\sigma_t^2} \frac{(1-\alpha_t)^2}{(1-\bar{\alpha}_t)\alpha_t}$. Note that although the forward and reverse processes are formulated based on variational diffusion models Ho et al. (2020), the probability modeling $P_\theta$ and similar loss can also be derived by score-based generative models Song & Ermon (2019); Song et al. (2021) as detailed in Appendix F.

**Variational Autoencoders (VAEs).** VAE Kingma et al. (2019) encodes an input $x$ into a latent variable $z$ and reconstruct the input $x$ from the latent variable. The encoder of VAE is a $\varphi-$ parameterized distribution $q_\varphi(z|x)$ conditioned on the input $x$. The decoder $p_\theta(x|z)$ parameterized by $\theta$ is used to reconstruct $x$ from the latent variable $z$. The probability model of VAE is

$$p_\theta(x) = \int p_\theta(x, z) \, dz = \mathbb{E}_{z \sim p'}[p_\theta(x|z)] \tag{5}$$

where $p'(z)$ is the prior distribution.

VAE is trained to maximize the log-likelihood $\log p_\theta(x)$. Since directly maximizing $\log p_\theta(x)$ is intractable, we usually minimize the average Evidence Lower Bound (ELBO) loss:

$$\mathbb{E}_{x \sim P_0}[\mathcal{L}_{\mathrm{ELBO}}(\theta, \varphi, x)] = \mathbb{E}_{x \sim P_0} \left[ \mathbb{E}_{q_\varphi(z|x)}[-\ln p_\theta(x|z)] \right] + \mathbb{E}_{x \sim P_0} \left[ D_{\mathrm{KL}}(q_\varphi(z|x) \,\|\, p'(z)) \right] \tag{6}$$

The first term, $J_{\mathrm{VAE}}(\theta, P_0) = \mathbb{E}_{x \sim P_0} \left[ \mathbb{E}_{q_\varphi(z|x)}[-\ln p_\theta(x|z)] \right]$, is the reconstruction loss that encourages the learned model to regenerate data that is consistent with the original data distribution. The second term , $R_{\mathrm{VAE}}(p', \varphi, P_0) = \mathbb{E}_{x \sim P_0} \left[ D_{\mathrm{KL}}(q_\varphi(z|x) \,\|\, p'(z)) \right]$, is the prior matching loss which measures how similar the learned latent distribution is to a prior belief held over latent variables. In practice, the encoder is usually parameterized as a Gaussian distribution with mean and variance given by neural networks $q_\varphi(z|x) = \mathcal{N}(z; \mu_\varphi(x), \mathrm{diag}(\sigma_\varphi^2(x)))$ such that the latent variable can be sampled as $z = \mu_\varphi(x) + \sigma_\varphi(x) \odot \nu, \nu \sim \mathcal{N}(\mathbf{0}, \mathbf{I})$ where $\odot$ is element wise product.

## 4.2 GENERATIVE AMBIGUITY SET

From the probabilistic modeling perspective of generative models, a well-trained generative model with sufficient expressive capacity can closely approximate the nominal distribution, while perturbations of such a generative model can induce distributions that deviate from the nominal distribution. To build the generative ambiguity set, we need to establish the consistency of the perturbed generative models with respect to the nominal distribution. Thus, we show in Lemma 1 that for likelihood-based generative models, the discrepancy between the nominal distribution $P_0$ and the generative distribution $P_\theta$ can be formally bounded through the *inclusive* KL-divergence.

**Lemma 1.** *The inclusive KL-divergence between the nominal distribution $P_0$ and the sampling distribution $P_\theta$ of a likelihood-based generative model can be bounded as*

$$D_{\mathrm{KL}}(P_0 \| P_\theta) \leq J(\theta, P_0) + R(p', P_0) + C_1$$

*where $J(\theta, P_0)$ is the reconstruction loss which can be $J_{\mathrm{DM}}(\theta, P_0)$ in equation 4 or $J_{\mathrm{VAE}}(\theta, P_0)$ in the first term of equation 6, $R(p', P_0)$ is a prior matching loss relying on a prior distribution $p'$ but not relying on $\theta$, and $C_1$ is a constant that does not rely on $p'$ or $\theta$.*

The proof of this lemma can be found in Appendix C.1. Lemma 1 shows that if the reconstruction loss of a diffusion model or VAE is bounded, i.e., $J(\theta, P_0) \leq \epsilon$, then the KL divergence

$D_{\mathrm{KL}}(P_0 \,\|\, P_\theta)$ is bounded by $\epsilon$ plus additional terms not relying on the parameter $\theta$. The prior matching term $R(p', P_0)$ depends on the generative model designs. For example, it holds that

$$R(p', P_0) = \mathbb{E}_{P_0(x)} \left[ D_{\mathrm{KL}} \left[ q(x_T|x_0) \| p'(x_T) \right] \right]$$ for diffusion models, and a well-designed forward

process can contribute to nearly zero prior matching error Ho et al. (2020); Song et al. (2020). The constant $C_1$ relies on the negative entropy $-H(P_0)$ of the nominal distribution.

Importantly, the *inclusive* KL-divergence $D_{\mathrm{KL}}(P_0 \,\|\, P_\theta)$ in Lemma 1 is *not* the *exclusive* KL-divergence $D_{\mathrm{KL}}(P_\theta \,\|\, P_0)$ in KL-DRO Hu & Hong (2013); Kocuk (2020); Liu et al. (2022b); Michel et al. (2021). The *inclusive* KL-divergence $D_{\mathrm{KL}}(P_0 \,\|\, P_\theta)$ in Lemma 1 allows the distribution $P_\theta$ of generative models to have a broader support space than $P_0$ (i.e., $P_0 \ll P_\theta$).

Given a likelihood-based generative model satisfying Lemma 1, the induced probability distribution $P_\theta$ maintains a constrained discrepancy from the nominal distribution $P_0$ whenever its reconstruction loss $J(\theta, P_0)$ remains bounded. At the same time, $P_\theta$ is capable of capturing a broader support space than $P_0$. This observation motivates us to define ambiguity sets in DRO using generative models, leading to the formulation of DRO with Generative Ambiguity Sets (GAS).

$$\min_{w \in \mathcal{W}} \max_{\theta \in \Theta} \mathbb{E}_{x \sim P_\theta}[f(w, x)] \quad \text{s.t.} \quad J(\theta, P_0) \leq \epsilon \tag{7}$$

where $J(\theta, P_0)$ is the reconstruction loss of a likelihood-based generative model such as diffusion models or VAE, $P_\theta$ denotes the sampling distribution of the generative model, and $\epsilon$ is the adversarial budget.

The generative ambiguity set has the following advantages. Due to the distribution modeling capability of diffusion models, the constraint on the reconstruction loss $J(\theta, P_0)$ ensures that the generative distributions remain consistent with the nominal distribution $P_0$, mitigating over-conservativeness issues in DRO. In addition, it leverages generative models' ability to generate diverse samples beyond the nominal support space, enabling the identification of distributions with the worst-case optimization performance. Furthermore, the inner maximization in equation 7 operates in a finite, parameterized space, leading to tractable solutions.

## 5 ALGORITHM FOR DRO WITH GENERATIVE AMBIGUITY SET

Despite the advantages of generative ambiguity sets, it is challenging to solve DRO with generative ambiguity set in equation 7 given the complexity of generative models. In this section, we provide an algorithm to solve DRO with generative-ambiguity set in GAS-DRO (Algorithm 1). Illustrated in Figure 1, GAS-DRO includes an inner maximization ((InnerMax)) loop and an outer minimization loop which are introduced below.

**Inner Maximization (`InnerMax`).** The inner maximization of equation 7 presents the following challenges. First, the inner maximization is a constrained optimization over the diffusion parameter space, which can not be directly solved by gradient-based methods. Second, the objective is an expectation over the distribution parameterized by the generative model, which cannot be directly differentiated. Therefore, we need to reformulate the objective into a tractable form.

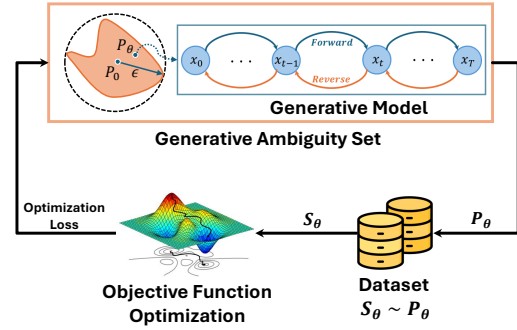

Figure 1: DRO with generative ambiguity set

To solve the constrained maximization problem, we adopt a dual learning approach: introducing a Lagrangian dual $\mu > 0$ to reformulate it as the unconstrained problem:

$$\max_\theta \ \mathbb{E}_{x \sim P_\theta}[f(w, x)] - \mu J(\theta, P_0) \tag{8}$$

and updating $\mu$ via dual gradient descent. As shown in Algorithm 1, the Lagrangian dual variable $\mu$ in equation 8 is adaptively updated to balance the maximization of the expected objective with the enforcement of the constraint. Intuitively, Algorithm 1 increases $\mu$ when $J(\theta, P_0)$ exceeds the budget $\epsilon$, thereby placing greater emphasis on reducing constraint violations, and decreases $\mu$ otherwise to prioritize objective maximization.

---

**Algorithm 1:** DRO with Generative Ambiguity Set (`GAS-DRO`)

---

**Input:** Nominal distribution $P_0$; Adversary budget $\epsilon > 0$; Step size $\eta > 0$, $\lambda > 0$.
**Init:** Initialize decision variable $w$, sampling model parameter $\theta$ and Lagrangian weight $\mu > 0$;
**for** $j = 1, 2, \cdots, I$ **do**

>    // Inner maximization (InnerMax) to update the generative model
> **for** $k = 1, 2, \cdots, K$ **do**
> > Update generative model parameter $\theta_k$ by solving equation 8 given $\mu$;
> > Update Lagrangian parameter: $\mu \leftarrow \max\{0, \mu + \eta\big(J(\theta_k, P_0) - \epsilon\big)\}$;
>
> // Outer minimization to update decision variable
> Generate dataset $S_j$ with generative model $P_{\theta^{(j)}}$, $\theta^{(j)} \sim \text{Unif}\{\theta_1, \ldots, \theta_K\}$;
> Update decision variable $w$: $w_j = w_{j-1} - \lambda \cdot \nabla_w \mathbb{E}_{x \in S_j}[f(w_{j-1}, x)]$;

**return** $w$ uniformly selected from $[w_1, \cdots, w_I]$;

---

Next, we apply the policy optimization methods to transform the objective in equation 8 into a tractable form. Here, we give the transformed objectives based on the diffusion model formulations while deferring the objective reformulations based on VAE formulations to Appendix B.

Based on the policy gradient trick Sutton et al. (1999), we can transform the objective equation 8 into the following tractable form as detailed in Appendix A.1,

$$\max_{\theta} \hat{\mathbb{E}}_{P_\theta(x_{0:T})}[\ln P_\theta(x_{0:T}) \cdot f(w, x_0)] - \mu \cdot J_{\text{DM}}(\theta, P_0) \tag{9}$$

where $\hat{\mathbb{E}}_{P_\theta(x_{0:T})}$ is the empirical mean based on the $T-$step samples for the backward process of the diffusion model $P_\theta$, and $\ln P_\theta(x_{0:T}) = -\sum_{t=1}^{T}[x_{t-1} - \mu_\theta(x_t, t)]^2 + C_2$ where $C_2$ is a constant.

Proximal Policy Optimization (PPO) Schulman et al. (2017) is believed to have more stable performance which transforms the objective equation 8 into a differentiable form.

$$\max_{\theta} \hat{\mathbb{E}}_{P_{\theta_{\text{old}}}(x_{0:T})}[\min(r_\theta(x_{0:T})f(w, x_0)), \text{clip}(r_\theta(x_{0:T}), 1 - \kappa, 1 + \kappa) \cdot f(w, x_0))] - \mu \cdot J_{\text{DM}}(\theta, P_0) \tag{10}$$

where $P_{\theta_{\text{old}}}$ is a reference diffusion model which can be selected as a pre-trained diffusion model on the nominal distribution $P_0$, the probability ratio is $r_\theta(x_{0:T}) = \frac{P_\theta(x_{0:T})}{P_{\theta_{\text{old}}}(x_{0:T})} = \exp\{-\sum_{t=1}^{T}(\frac{\|x_{t-1} - \mu_\theta(x_t, t)\|^2}{2\sigma_t^2} - \frac{\|x_{t-1} - \mu_{\theta_{\text{old}}}(x_t, t)\|^2}{2\sigma_t^2})\}$, and $\kappa \in (0, 1)$ is the clipping parameter to avoid overly-large policy updates. To reduce the training complexity, instead of optimizing all the $T$ steps, we can only optimize the last $T'$ steps of the backward process by choosing $r_\theta(x_{0:T'}) = \exp\{-\sum_{t=1}^{T'}(\frac{\|x_{t-1} - \mu_\theta(x_t, t)\|^2}{2\sigma_t^2} - \frac{\|x_{t-1} - \mu_{\theta_{\text{old}}}(x_t, t)\|^2}{2\sigma_t^2})\}$ and only keep the loss terms of corresponding steps in $J(\theta, P_0)$. The performance of this trick to reduce computation complexity is verified by our experiments in Section 7. More derivation details for the diffusion-based objective can be found in Appendix A.2.

**Min-Max Solution**. With the inner maximization solution, we integrate the design methodology of gradient descent with max-oracle Jin et al. (2020) into the min-max solution in Algorithm 1. Specifically, in each iteration, we first run `InnerMax` to search for the worst-case generative model $P_{\theta^{(j)}}$ that maximizes the expected loss given the optimization variable $w$. Next, in order to provide diverse samples for the statistical optimization based on the worst-case probability model, we generate an adversarial dataset $S_j$ based on the diffusion model $P_{\theta^{(j)}}$ and use it to update $w$. The convergence of Algorithm 1 in proved in Theorem 1 and Theorem 2.

## 6 ANALYSIS

We provide the convergence analysis which holds for Algorithm 1 in this section. We first prove the convergence of the inner maximization, followed by the overall convergence of the DRO with generative ambiguity sets.

**Theorem 1** (Convergence of Inner Maximization). *Let $\theta^*$ be the optimal diffusion parameter that solves the inner maximization equation 7 given a variable $w$. If the reconstruction loss is bounded*

*as $J(\theta) \leq \bar{J}$ and the step size is chosen as $\eta \sim \mathcal{O}(\frac{1}{\sqrt{K}})$, the inner maximization error holds that*

$$\Delta' := \mathbb{E}_{x \sim P_{\theta^*}}[f(w, x)] - \mathbb{E}_k \mathbb{E}_{x \sim P_{\theta_k}}[f(w, x)] \leq \frac{1}{\sqrt{K}} \max\{\epsilon, \bar{J}\} \|\mu^{(1)}\| \tag{11}$$

*where the outer expectation is taken over the randomness of output selection. In addition, the inclusive KL-divergence with respect to the nominal distribution is bounded as*

$$\mathbb{E}_k[D_{\mathrm{KL}}(P_0 \| P_{\theta_k})] \leq \epsilon + \frac{\max\{\epsilon, \bar{J}\} |\mu_C - \mu^{(1)}|}{\sqrt{K}(\mu_C - \mu^*)} + R(\pi, P_0) + C_1 \tag{12}$$

*where $\mu_C > \mu^*$ with $\mu^*$ being the optimal dual variable for the constrained optimization in equation 7, $R(\pi, P_0)$ is the prior matching loss with $\pi$ being the prior distribution, and $C_1$ is a constant.*

Proofs of Theorem 1 are provided in Appendix C.2. The bound shows that when the inner iteration number $K$ is sufficiently large, the inner maximization error will be small enough, so GAS-DRO can find a worst-case distribution in the Diffusion Ambiguity Set. Moreover, the *inclusive* KL-divergence w.r.t. the nominal distribution is bounded by the budget $\epsilon$, the prior matching error $R(\pi, P_0)$ (e.g. $D_{\mathrm{KL}}(p_T \| \pi)$ in diffusion models), and the constant gap $C_1$ which relying on the negative entropy $-H(P_0)$. This substantiates that inner maximization of GAS-DRO can find a worst-case distribution that is consistent with the nominal distribution by adjusting $\epsilon$.

**Theorem 2** (Convergence of GAS-DRO). *Assume that objective $f(w, x)$ is $\beta-$smooth and $L-$Lipschitz with respect to $w$ and is upper bounded by $\bar{f}$. If each dataset $S_j$ sampled from the generative model contains $n$ examples and the step size is $\lambda \sim \mathcal{O}(\sqrt{\frac{1}{\beta L^2 H}})$, then with probability $1 - \delta, \delta \in (0, 1)$, the average norm of Moreau envelope of $\phi(w) := \max_\theta \mathbb{E}_{P_\theta}[f(w, x)]$ satisfies*

$$\mathbb{E}_{j,k}\left[\|\nabla \phi_{\frac{1}{2\beta}}(w)\|^2\right] \leq 4\beta \Delta' + \frac{V_1}{\sqrt{n}} + \frac{V_2}{\sqrt{H}} \tag{13}$$

*where the expectation is taken over the randomness of output selection, $\Delta'$ is the error of inner maximization bounded in Theorem 1, $V_1 = 8\beta \bar{f} \sqrt{\log(2/\delta)}$ and $V_2 = 4L\sqrt{(\phi_{\frac{1}{2\beta}}(w_1) - \min_w \phi(w))\beta}$.*

The proof of Theorem 2 is deferred to Appendix C.3. It shows that with sufficiently large iteration number $H$ and sampling size $n$, the average gradient norm of the Moreau envelope for the optimal inner maximization function $\phi(w)$ is bounded by the error $\Delta'$ of the maximization oracle. Since $\Delta'$ decreases as the inner iteration number $K$ becomes sufficiently large as proved by Theorem 1, the average gradient norm of the Moreau envelope can be small enough. The convergence of the Moreau envelope gradient norm indicates that the algorithm converges to an approximately stationary point as explained in Appendix C.3.1. The intuitive reason is that if the gradient norm of Moreau envelope is bounded by a small enough value $\Delta$ for a decision variable $w$, i.e. $\|\nabla \phi_{\frac{1}{2\beta}}(w)\| \leq \Delta$, there exists $\hat{w}$ which is close enough to $w$ ($\|\hat{w}_j - w_j\| \leq \frac{\Delta}{2\beta}$) and is an approximately stationary solution for minimizing the optimal inner maximization function $\phi$. Thus, GAS-DRO can converge to an approximately stable solution of equation 1 with enough maximization and minimization iterations.

**Impact of the expressive capacity of generative models.** While the generative models with sufficient expressive capacity can define an ambiguity set with diverse adversarial distributions, the real world generative models always have finite capacity. The expressive capacity of generative models determines whether a real worst-case distribution can be effectively identified by the inner maximization and further affects the robustness performance of DRO. To quantify the impact of the expressivity of generative models, we formally define the $\Gamma-$expressivity of the generative model.

**Definition 1** ($\Gamma-$expressivity of the generative model). *A generative model $P_\theta$ is $\Gamma-$expressive ($\Gamma > 0$) if for any testing distribution $P$ in the set of all possible testing distributions $\mathcal{P}$, there exists a $\theta$ such that $D_{\mathrm{W}}(P_\theta, P) \leq \Gamma$ where $D_{\mathrm{W}}$ is the Wasserstein metric.*

Next, we analyze the impact of the $\Gamma-$expressivity on the DRO performance. Given the set of all possible testing distributions $\mathcal{P}$, GAS-DRO has better robustness performance if it can identify the real worst-case distribution $\tilde{P} = \arg\max_{\mathcal{P}} \mathbb{E}_{x \sim P}[f(w, x)]$ with a smaller error. Thus, we bound the error of the inner maximization $\tilde{\Delta}' := \mathbb{E}_{x \sim \tilde{P}}[f(w, x)] - \mathbb{E}_k \mathbb{E}_{x \sim P_{\theta_k}}[f(w, x)]$ in Corollary 1 (in

Appendix C.4). We observe that $\Gamma-$expressivity affects the inner maximization performance by introducing an additional error $L_x\Gamma$ where $L_x$ is the Lipschitz continuity parameter of $f(w, x)$ with respect to $x$. When the expressivity parameter $\Gamma$ is smaller, the generative model has a stronger ability to express the possible testing distributions and GAS-DRO can identify adversarial distributions that are more consistent with the true worst-case testing distribution, as reflected by a smaller inner maximization gap. Consequently, by optimizing the parameter $w$ based on the learned adversarial distribution, GAS-DRO can achieve better robustness across the true testing distribution. In the ideal case when $\Gamma = 0$, the error bound of the inner maximization coincides with the inner maximization error bound in Theorem 1, which asymptotically converges to zero as the number of inner iterations $K$ increases.

# 7 EXPERIMENTS

This section reports the main experiment results on a representative ML task—time series forecasting based on the Electricity Maps (2025) —to verify the effectiveness of the proposed algorithm. Additionally, we evaluate the performance of GAS-DRO on an image classification task based on the MNIST dataset Lecun et al. (1998) and USPS dataset Hull (1994), with details in Appendix E.

**Task.** Accurate forecasting of renewable is critical for power grid optimization, energy-efficient computing, sustainable power management, and greenhouse gas emission reduction. We evaluate GAS-DRO on time-series prediction with the Electricity Maps (2025) datasets, using an $n$-step sliding window to predict emissions at step $(n + 1)$.

**Datasets.** We conduct experiments using datasets from the Electricity Maps (2025) platform, which provides high-resolution spatio-temporal data on electricity operations, including carbon intensity ($gCO_2$eq/kWh) and energy mix. We construct training and OOD testing datasets based on the hourly records from 2021–2024 across four regions: California (**BANC_21$\sim$24**), Texas (**ERCO_21$\sim$ 24**), Queensland (**QLD_21$\sim$24**), and the United Kingdom (**GB_21$\sim$24**). The training set is constructed by merging **BANC_23$\sim$24** (**BANC_2324**) into 438 sequence samples, while evaluation is performed on independent test sets of 312 samples each from other years and regions. To measure distributional shift between training and test sets, we compute the Wasserstein distance, reported in Table 1.

**Baselines.** The baselines which are compared with GAS-DRO in our experiments are introduced as below. **ML** is a method that trains the standard ML to minimize the time series forecasting error without DRO. **DML** represents an ML model fine-tuned with diffusion-generated augmented datasets. **W-DRO** represents the FWDRO algorithm proposed in Staib & Jegelka (2017), where the ambiguity set is characterized by the Wasserstein metric. **KL-DRO** represents the classic KL-divergence-based DRO framework, in which the ambiguity set is characterized by the KL- divergence Hu & Hong (2013). **DRAGEN** represents a Wasserstein DRO framework, in which the ambiguity set is constructed as a Wasserstein ball in the latent space of a learned generative model Ren & Majumdar (2022). **P-DRO** Michel et al. (2021) is a DRO framework that parameterizes the adversarial distributions using a neural generative model within a KL-divergence–based ambiguity set.

**Setups.** Detailed training setups of GAS-DRO and other baselines are provided in Appendix D.

**Main Results.** The main results are given in Table 1. We observe that all DRO methods improve testing performance on OOD datasets compared to ML by optimizing the worst-case expected loss, while DML improves upon ML by leveraging diffusion-generated augmented data. Using ML as the reference baseline, GAS-DRO achieves the highest performance gain of 63.7%, followed by DRAGEN (48.9%),P-DRO (42.5%), DML (39.7%), KL-DRO (36.1%), and W-DRO (24.0%). Although both DML and traditional DRO methods deliver noticeable improvements, GAS-DRO still surpasses them by an average margin of 25.4%, likely because it exploits the distribution learning capability of diffusion models to construct ambiguity sets, thereby generating adversarial distributions that are both strong and realistic and ultimately achieving superior OOD generalization.

In addition, our method outperforms the DRO baselines with generative models. Although DRAGEN also employs a generative model to construct its ambiguity set, it still relies on relaxing Wasserstein DRO framework which is limited in overcoming the inherent limitations of W-DRO. Moreover, P-DRO, as a DRO algorithm that uses generative models to model the adversarial distributions, remains restricted by the KL-divergence-based ambiguity set, preventing it from modeling diverse worst-case distributions. Also, the crude approximations in P-DRO may bring harmful effects. By

contrast, `GAS-DRO` directly defines the ambiguity set on the parameterized space of generative models, enabling it to generate worst-case distributions that are both diverse and consistent with the nominal distribution, thereby offering a principled and tractable DRO algorithm.

In Appendix D.4, we provide additional results and more detailed analysis. Ablation studies are conducted to evaluate the impacts of the adversarial budgets and the distributional discrepancies caused by different noise types and strengths. Furthermore, we empirically validate the effect of the expressive capacity by evaluating `GAS-DRO` with diffusion models with different architectures and a VAE model in Appendix D.4.5. Moreover, we measure and discuss the computational overhead of `GAS-DRO` and other baselines in Appendix D.4.6.

Table 1: Test Result on Different Datasets.

| Datasets (Wasserstein Distance) | MSE | | | | | | |
|---|---|---|---|---|---|---|---|
| | GAS−DRO | DRAGEN | P−DRO | KL−DRO | W−DRO | DML | ML |
| **BANC_22** (0.0240) | **0.0047** | 0.0069 | 0.0079 | 0.0086 | 0.0073 | 0.0078 | 0.0183 |
| **BANC_21** (0.1213) | **0.0054** | 0.0087 | 0.0074 | 0.0112 | 0.0121 | 0.0093 | 0.0238 |
| **QLD_24** (0.2171) | **0.0450** | 0.0618 | 0.0752 | 0.0754 | 0.0823 | 0.0766 | 0.0887 |
| **QLD_23** (0.2033) | **0.0509** | 0.0681 | 0.0820 | 0.0831 | 0.0879 | 0.0834 | 0.0946 |
| **QLD_22** (0.2782) | **0.0192** | 0.0298 | 0.0331 | 0.0379 | 0.0557 | 0.0352 | 0.0667 |
| **QLD_21** (0.3054) | **0.0186** | 0.0288 | 0.0322 | 0.0377 | 0.0574 | 0.0339 | 0.0696 |
| **GB_24** (0.0419) | **0.0119** | 0.0146 | 0.0191 | 0.0178 | 0.0176 | 0.0172 | 0.0285 |
| **GB_23** (0.0666) | **0.0100** | 0.0141 | 0.0154 | 0.0178 | 0.0200 | 0.0164 | 0.0311 |
| **GB_22** (0.1255) | **0.0105** | 0.0153 | 0.0158 | 0.0197 | 0.0245 | 0.0172 | 0.0360 |
| **GB_21** (0.1359) | **0.0094** | 0.0140 | 0.0137 | 0.0181 | 0.0229 | 0.0158 | 0.0340 |
| **ERCO_24** (0.1206) | **0.0158** | 0.0201 | 0.0211 | 0.0241 | 0.0266 | 0.0224 | 0.0379 |
| **ERCO_23** (0.1207) | **0.0106** | 0.0146 | 0.0148 | 0.0179 | 0.0196 | 0.0162 | 0.0319 |
| **ERCO_22** (0.1581) | **0.0076** | 0.0115 | 0.0103 | 0.0146 | 0.0187 | 0.0123 | 0.0312 |
| **ERCO_21** (0.1417) | **0.0093** | 0.0141 | 0.0143 | 0.0189 | 0.0263 | 0.0160 | 0.0382 |
| **Average** | **0.0163** | 0.0230 | 0.0259 | 0.0288 | 0.0342 | 0.0271 | 0.0450 |
| **Worst** | **0.0509** | 0.0681 | 0.0820 | 0.0831 | 0.0879 | 0.0834 | 0.0946 |

## 8 CONCLUSION

In this paper, we introduce a novel generative-model–based ambiguity modeling framework for DRO and propose `GAS-DRO` to solve DRO with generative ambiguity sets. While the method is developed based on the formulations of diffusion models or VAEs, the proposed framework is general and can be adapted to other likelihood-based generative models. By exploiting the expressive distributional modeling capacity of generative models, `GAS-DRO` can effectively identify adversarial distributions that induce worst-case performance while remaining consistent with the nominal distribution, thereby yielding tractable DRO solutions with strong generalization performance . We establish the stationary convergence guarantees of `GAS-DRO` and demonstrate through experiments its superior OOD generalization performance by comparing with the state-of-the-art baselines.

**Future Directions**. It would be interesting to formally establish consistency bounds for the training losses of other generative models and to exploit them for constructing ambiguity sets in DRO. In addition, improving the training efficiency of generative models would directly improve the computational efficiency of `GAS-DRO`. Moreover, generative ambiguity sets provide a flexible interface for incorporating prior knowledge about the testing environment into the generative model, which can further enhance robustness and generalization under distributional shifts.

**Broader Impacts**. The proposed algorithm provides an effective generative model–based approach to ambiguity modeling that addresses fundamental challenges in DRO. It has the potential to bridge the gap between imperfect training environments and real-world deployment, thereby enhancing the safety and robustness of machine learning models across a broad range of critical real-world tasks.

## ACKNOWLEDGMENT

This work were supported in part by University of Houston Start-up Funds 74825 [R0512039] and 74833 [R0512042].

We thank the ICLR reviewers for the constructive comments which helped us to improve the draft.

ETHICS STATEMENT

This work adheres to the ICLR Code of Ethics. Our research does not involve human subjects, personally identifiable information, or sensitive personal data, and thus no IRB approval was required. All datasets used in this study are publicly available, widely recognized in the research community, and utilized strictly in accordance with their licenses. For any preprocessing, we provide clear documentation to ensure transparency. The proposed methods are designed to advance the theoretical and practical understanding of distributionally robust optimization and generative modeling. These contributions are primarily academic in nature and intended to support future work in reliable machine learning. We do not anticipate direct harmful applications.

REPRODUCIBILITY STATEMENT

We have taken extensive measures to ensure the reproducibility of our work. A complete description of the proposed model architecture, training objectives, and optimization procedure is provided in Section 5. The datasets used in our experiments are all publicly available and widely adopted in the research community. We provide a detailed account of the data preprocessing pipeline, including normalization and sampling strategies, in Appendix D.

To further enhance reproducibility, all hyperparameters, model configurations, and training schedules are reported explicitly in Section 7 and Appendix D, along with the evaluation metrics and criteria employed for performance comparison. For theoretical contributions, we clearly state all assumptions and provide complete, rigorous proofs of our claims in Appendix C.

We also include ablation studies and sensitivity analyses to illustrate the robustness of our results with respect to key hyperparameters, which may assist future researchers in replicating or extending our work. Finally, we provide our source code in the link https://github.com/CIGLAB-Houston/GAS-DRO.

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

## A  DETAILS OF GAS-DRO (ALGORITHM 1) WITH DIFFUSION MODELS

In this section, we provide details to transform the objective of inner maximization in Algorithm 1 based on the formilations of diffusion models.

### A.1  OBJECTIVE DERIVATION BY POLICY GRADIENT

Let $x_{0:T}$ be the output vector of each step in the backward process of the diffusion model. Denote $P_{0:T,\theta}$ as the joint distribution of $x_{0:T}$. Since $x_0 \sim P_\theta$, we can express the first term of the Lagrangian-relaxed objective as

$$\mathbb{E}_{x \sim P_\theta}[f(w, x)] = \mathbb{E}_{x_{0:T} \sim P_{0:T,\theta}}[f(w, x_0)] \tag{14}$$

Then, the gradient of Lagrangian-relaxed objective can be expressed as

$$
\begin{aligned}
&\nabla_\theta \left( \mathbb{E}_{x \sim P_\theta}[f(w, x)] - \mu J(\theta, P_0) \right) \\
=&\nabla_\theta \left( \mathbb{E}_{x_{0:T} \sim P_{0:T,\theta}}[f(w, x_0)] - \mu J(\theta, P_0) \right) \\
=&\int_{x_{0:T}} f(w, x_0) \nabla_\theta P_{0:T,\theta}(x_{0:T}) \mathrm{d}x_{0:T} - \mu \nabla_\theta J(\theta, P_0) \\
=&\int_{x_{0:T}} P_{0:T,\theta}(x_{0:T}) f(w, x_0) \nabla_\theta \ln P_{0:T,\theta}(x_{0:T}) \mathrm{d}x_{0:T} - \mu \nabla_\theta J(\theta, P_0) \\
=&\mathbb{E}_{x_{0:T} \sim P_{0:T,\theta}} \left[ f(w, x_0) \nabla_\theta \ln P_{0:T,\theta}(x_{0:T}) \right] - \mu \nabla_\theta J(\theta, P_0)
\end{aligned}
\tag{15}
$$

The first term $\mathbb{E}_{x_{0:T} \sim P_{0:T,\theta}}$ can be calculated by empirical mean $\hat{\mathbb{E}}_{x_{0:T} \sim P_{0:T,\theta}}$ based on the examples sampled from $P_{0:T,\theta}$. Thus, we can equivalently implement the gradient ascent by optimizing the objective in equation 9:

$$\max_\theta \hat{\mathbb{E}}_{x_{0:T} \in P_{0:T,\theta}}[\ln P_{0:T,\theta}(x_{0:T}) \cdot f(w, x_0)] - \mu \cdot J(\theta, P_0) \tag{16}$$

Next, we derive the expression for $\ln P_{0:T,\theta}(x_{0:T})$. Consider a discrete-time diffusion backward process as

$$x_{t-1} = \mu_\theta(x_t, t) + \sigma_t w_t \tag{17}$$

where $w_t$ is a standard multi-dimensional Gaussian variable. Given $\theta$, the conditional probability at each step $t$ is

$$P_{t-1,\theta}(x_{t-1} \mid x_t) = \mathcal{N}(x_{t-1}, \mu_\theta(x_t, t), \sigma_t^2 \mathbf{I}) \tag{18}$$

We can explicitly express the joint distribution $P_{0:T,\theta}$ of $x_{0:T}$ for all $T$ backward steps:

$$P_{0:T,\theta}(x_{0:T}) = P(x_T) \prod_{t=1}^{T} P_{t-1,\theta}(x_{t-1} \mid x_t) = \frac{1}{\sqrt{2\pi}} e^{-\frac{\|x_T\|^2}{2}} \cdot \frac{1}{\sqrt{2\pi}\sigma_t} e^{-\sum_{t=1}^{T} \frac{\|x_{t-1} - \mu_\theta(x_t, t)\|^2}{2\sigma_t^2}} \tag{19}$$

Thus, we can get the expression of $\ln P_{0:T,\theta}(x_{0:T})$ as

$$\ln P_{0:T,\theta}(x_{0:T}) = -\sum_{t=1}^{T} \|x_{t-1} - \mu_\theta(x_t, t)\|^2 / (2\sigma_t^2) + C_2 \tag{20}$$

where $C_2 = -\ln(2\pi\sigma_t) - \frac{\|x_T\|^2}{2}$.

### A.2  OBJECTIVE DERIVATION BY PROXIMAL POLICY OPTIMIZATION

In this PPO method, we convert the first term of equation 8 as a PPO-like objective given a reference diffusion model $P_{\theta_0}$, i.e.

$$
\begin{aligned}
\mathbb{E}_{x \sim P_\theta}[f(w, x)] &= \mathbb{E}_{x_{0:T} \sim P_{0:T,\theta}}[f(w, x_0)] \\
&= \mathbb{E}_{x_{0:T} \sim P_{0:T,\theta_0}} \left[ \frac{P_{0:T,\theta}(x_{0:T})}{P_{0:T,\theta_0}(x_{0:T})} f(w, x_0) \right]
\end{aligned}
\tag{21}
$$

where the probability ratio is

$$r_\theta(x_{0:T}) = \frac{P_{0:T,\theta}(x_{0:T})}{P_{0:T,\theta_0}(x_{0:T})} = \exp\left\{-\sum_{t=1}^{T}\left(\frac{\|x_{t-1} - \mu_\theta(x_t,t)\|^2}{2\sigma_t^2} - \frac{\|x_{t-1} - \mu_{\theta_0}(x_t,t)\|^2}{2\sigma_t^2}\right)\right\}$$

given the joint probability expression. The expectation can be approximated by empirical mean based on the examples sampled by $P_{0:T,\theta_0}$. Like PPO, we apply clipping on the ratio to avoid overly-large updates, and we can get the objective in Eqn. equation 10:

$$\max_\theta \hat{\mathbb{E}}_{P_{0:T,\theta_0}(x_{0:T})}[\min(r_\theta(x_{0:T})f(w,x_0)), \text{clip}(r_\theta(x_{0:T}), 1-\kappa, 1+\kappa)\cdot f(w,x_0))] - \mu\cdot J(\theta, P_0) \tag{22}$$

where $\kappa \in (0,1)$ is the clipping parameter.

## B    DETAILS OF GAS-DRO (ALGORITHM 1) WITH VAEs

### B.1    AMBIGUITY SET FOR VAE

Given the formulations of VAE in Section 4.1, as is proved in Appendix C.1, Lemma 1 can be restated as below.

**Lemma 1** (Restated for VAEs).    *The inclusive KL-divergence between the empirical distribution $P_0(x)$ and the VAE distribution $P_\theta(x)$ is bounded as*

$$D_{\text{KL}}\big(q_0(x) \,\|\, p_\theta(x)\big) \leq J_{VAE}(\varphi, \theta, P_0) + \mathbb{E}_{P_0(x)}\left[D_{\text{KL}}\big(q_\varphi(z \mid x) \,\|\, p'(z)\big)\right] - H(P_0(x))$$

*where $J_{VAE}(\varphi, \theta, P_0) = \mathbb{E}_{P_0(x)}\mathbb{E}_{q_\varphi(z|x)}\big[-\log p_\theta(x \mid z)\big]$ is the reconstruction loss of VAE, $q_\varphi(z \mid x)$ denotes the encoder with parameter $\varphi$ (approximate posterior), $p_\theta(x \mid z)$ denotes the decoder model with parameter $\theta$, and $p'(z)$ is the prior distribution.*

To construct a VAE-based ambiguity set for DRO, we first train the a VAE model based on the nominal distribution $P_0$:

$$(\varphi^*, \theta^*) = \min_{\varphi,\theta}[-\mathbb{E}_{x\sim P_0}\mathbb{E}_{z\sim q_\varphi(z|x)}[\log p_\theta(x|z)] + \mathbb{E}_{x\sim P_0}[D_{\text{KL}}\big(q_\varphi(z|x) \,\|\, p(z)\big)]] \tag{23}$$

Then, we keep $\varphi = \varphi^*$ and change $\theta$ in the set $\{\theta \mid J_{\text{VAE}}(\varphi^*, \theta, P_0) \leq \epsilon\}$. In this way, we get DRO with VAE-based generative ambiguity set as

$$\min_{w\in\mathcal{W}} \max_{\theta\in\Theta} \mathbb{E}_{x\sim P_\theta}[f(w,x)], \quad \text{s.t.} \quad J_{\text{VAE}}(\varphi^*, \theta, P_0) \leq \epsilon \tag{24}$$

where $J_{\text{VAE}}(\varphi, \theta, P_0) = \mathbb{E}_{P_0(x)}\mathbb{E}_{q_\varphi^*(z|x)}\big[-\log p_\theta(x \mid z)\big]$.

### B.2    ALGORITHM DETAILS OF GAS-DRO WITH VAEs

As in Algorithm 1, to solve the constrained maximization problem, we adopt a dual learning approach: introducing a Lagrangian dual $\mu > 0$ to reformulate it as the unconstrained problem and updating $\mu$ via dual gradient descent:

$$\max_\theta \ \mathbb{E}_{x\sim P_\theta}[f(w,x)] - \mu J_{\text{VAE}}(\varphi^*, \theta, q_0(x)) \tag{25}$$

In order to get $J_{\text{VAE}}(\varphi^*, \theta, q_0(x)) = \mathbb{E}_{P_0(x)}\mathbb{E}_{q_\varphi^*(z|x)}\big[-\log p_\theta(x \mid z)\big]$, we first encode the initial dataset $S_0 \sim P_0$ into latent variables $Z_0$ using the pretrained encoder $\varphi^*$. Then, we can empirically compute the reconstruction loss as $J_{\text{VAE}}(\varphi^*, \theta, q_0(x)) = \hat{\mathbb{E}}_{x\sim P_0, z\sim Z_0}\big[-\log p_\theta(x \mid z)\big]$.

Next, to solve the inner maximization in Algorithm 1, we apply the policy optimization methods to transform the objective in equation 25 into a tractable form. As shown in Appendix A.1, based on policy gradient Sutton et al. (1999), we can transform the objective equation 25 into

$$\max_\theta \hat{\mathbb{E}}_{p_\theta(x,z)}[\ln p_\theta(x,z)\cdot f(w,x)] - \mu\cdot J_{\text{VAE}}(\varphi^*, \theta, q_0(x)) \tag{26}$$

where $\hat{\mathbb{E}}_{p_\theta(x,z)}$ is the empirical mean based on the decoding process of the VAE model.

Similar to diffusion models, PPO Schulman et al. (2017) provides enhanced stability by transforming the objective equation 25 into a differentiable objective function.

$$\max_{\theta} \hat{\mathbb{E}}_{p_{\theta_{\text{old}}}(x,z)}[\min(r_\theta(x,z)f(w,x)), \text{clip}(r_\theta(x,z), 1-\kappa, 1+\kappa) \cdot f(w,x))] - \mu \cdot J_{\text{VAE}}(\varphi, \theta, q_0(x)) \tag{27}$$

where $p_{\theta_{\text{old}}}(x,z)$ is a reference VAE decoder which can be selected as a pre-trained VAE decoder on the nominal distribution $P_0(x)$, the probability ratio is $r_\theta(x,z) = \frac{P_\theta(x,z)}{p_{\theta_{\text{old}}}(x,z)}$, and $\kappa \in (0,1)$ is the clipping parameter to avoid overly-large policy updates.

## C  THEOREM PROOFS

### C.1  PROOF OF LEMMA 1

**Lemma 1**. *The inclusive KL-divergence between the nominal distribution $P_0$ and the sampling distribution $P_\theta$ of a likelihood-based generative model can be bounded as*

$$D_{\text{KL}}(P_0\|P_\theta) \leq J(\theta, P_0) + R(p', P_0) + C_1$$

*where $J(\theta, P_0)$ is the reconstruction loss which can be $J_{\text{DM}}(\theta, P_0)$ in equation 4 or $J_{\text{VAE}}(\theta, P_0)$ in the first term of equation 6, $R(p', P_0)$ is a prior matching loss relying on a prior distribution $\pi$ but not relying on $\theta$, and $C_1$ is a constant that does not rely on $p'$ or $\theta$.*

*Proof.* Likelihood-based generative models are typically trained to maximize the lower bounds of their expected log-likelihood given their sampling distribution $P_\theta$ and a nominal distribution $P_0$, i.e. $\mathbb{E}_{P_0(x)}\left[\log P_\theta(x)\right]$. Applying the relationship between KL divergence and cross-entropy, we have

$$D_{\text{KL}}(P_0(x_0)\|P_\theta(x_0)) = -\mathbb{E}_{P_0(x)}\left[\log P_\theta(x)\right] + \mathbb{E}_{P_0(x)}\left[\log P_0(x)\right], \tag{28}$$

where the first term can be bounded by the training loss $J(\theta, P_0)$ of likelihood-based generative models plus an average prior matching error $R(p', P_0)$, and the second term is the entropy of $P_0$. Thus, we can get a general bound of the KL divergence in Lemma 1.

Next, we give the concrete bounds for variational diffusion models and VAEs. Also, the bound for score-matching-based diffusion models is given in Lemma 3 based on the SDE formulations of diffusion models.

We first give the bound based on the variational diffusion models. According to Luo (2022), it holds that

$$\mathbb{E}_{P_0(x)}\left[\log P_\theta(x)\right] \geq \mathbb{E}_{P_0(x)}\left[\mathbb{E}_{q(x_1|x_0)}[\log p_\theta(x_0 \mid x_1)]\right] - \mathbb{E}_{P_0(x)}\left[D_{\text{KL}}\left[q(x_T|x_0)\|p(x_T)\right]\right]$$

$$- \mathbb{E}_{P_0(x)}\left[\sum_{t=2}^{T} \mathbb{E}_{q(x_t|x_0)}\left[D_{\text{KL}}(q(x_{t-1} \mid x_t, x_0)\|p_\theta(x_{t-1} \mid x_t))\right]\right]$$

$$= -J_{\text{DM}}(\theta, P_0) - d\log(\sqrt{2\pi}\sigma_1) - \mathbb{E}_{P_0(x)}\left[D_{\text{KL}}\left[q(x_T|x_0)\|p(x_T)\right]\right] \tag{29}$$

where $J_{\text{DM}}(\theta, P_0)$ is the loss in equation 4 and $\sigma_1$ is the variance for the last reverse step in equation 3. Based on the relationship between KL divergence and cross-entropy, we have

$$D_{\text{KL}}(P_0(x_0)\|P_\theta(x_0)) = -\mathbb{E}_{P_0(x)}\left[\log p_\theta(x)\right] + \mathbb{E}_{P_0(x)}\left[\log P_0(x)\right]$$

$$\leq J_{\text{DM}}(\theta, P_0) + \mathbb{E}_{P_0(x)}\left[D_{\text{KL}}\left[q(x_T|x_0)\|p(x_T)\right]\right] + C_1 \tag{30}$$

where $C_1 = d\log(\sqrt{2\pi}\sigma_1) - H(P_0)$. Therefore, we get the bound in Lemma 1 with $R(p', P_0) = \mathbb{E}_{P_0(x)}\left[D_{\text{KL}}\left[q(x_T|x_0)\|p(x_T)\right]\right]$.

Next, we prove Lemma 1 for VAEs. According to Kingma et al. (2019), we can bound the the log-likelihood $\log p_\theta(x)$ by its EVIDENCE LOWER BOUND (ELBO):

$$\log p_\theta(x) \geq D_{\mathrm{KL}}\big(q_\varphi(z \mid x) \,\|\, p(z)\big) - \mathbb{E}_{q_\varphi(z|x)}\big[\log p_\theta(x \mid z)\big] \tag{31}$$

Applying the relationship between KL divergence and cross-entropy, we have

$$D_{\mathrm{KL}}\big(P_0(x) \,\|\, P_\theta(x)\big) \leq \mathbb{E}_{P_0(x)}\mathbb{E}_{q_\varphi(z|x)}\big[-\log p_\theta(x \mid z)\big] + \mathbb{E}_{P_0(x)}D_{\mathrm{KL}}\big(q_\varphi(z \mid x) \,\|\, p'(z)\big) - H(P_0(x))$$

$$\tag{32}$$

Thus, we get the inclusive KL-divergence bound in Lemma 1 for VAEs with $J(\theta, P_0) = J_{\mathrm{VAE}}(\theta, P_0) = \mathbb{E}_{P_0(x)}\mathbb{E}_{q_\varphi(z|x)}\big[-\log p_\theta(x \mid z)\big]$, $R(p', P_0) = \mathbb{E}_{P_0(x)}D_{\mathrm{KL}}\big(q_\varphi(z \mid x) \,\|\, p'(z)\big)$, and $C_1 = -H(P_0(x))$.

$\square$

## C.2 PROOF OF THEOREM 1

**Theorem 1.** *Let $\theta^*$ be the optimal diffusion parameter that solves the inner maximization equation 7 given a variable $w$. If the expected score-matching loss is bounded as $J(\theta) \leq \bar{J}$ and the step size is chosen as $\eta \sim \mathcal{O}(\frac{1}{\sqrt{K}})$, the inner maximization error holds that*

$$\Delta' := \mathbb{E}_{x \sim P_{\theta^*}}[f(w, x)] - \mathbb{E}_k \mathbb{E}_{x \sim P_{\theta_k}}[f(w, x)] \leq \frac{1}{\sqrt{K}}\max\{\epsilon, \bar{J}\}\|\mu^{(1)}\| \tag{33}$$

*where the outer expectation is taken over the randomness of output selection. In addition, the inclusive KL-divergence with respect to the nominal distribution is bounded as $\mathbb{E}_k[D_{\mathrm{KL}}(P_0\|P_{\theta_k})] \leq \epsilon + \frac{\max\{\epsilon, \bar{J}\}|\mu_C - \mu^{(1)}|}{\sqrt{K}(\mu_C - \mu^*)} + D_{\mathrm{KL}}(P_T\|\pi) + C_1$, where $\mu_C > \mu^*$ and $C_1$ are constants, $P_T$ is the output distribution of the forward process, and $\pi$ is the initial distribution of the reverse process.*

### C.2.1 CONVERGENCE OF DUAL GRADIENT DESCENT

To prove Theorem 1, we need the convergence analysis of a general dual gradient descent in the following lemma.

**Lemma 2.** *Assume that the dual variable is updated following*

$$\mu_{k+1} = \max\{\mu_k - \eta \cdot b_k, 0\}$$

*where $\eta > 0$ and $b_k$ has the same dimension as $\mu$ and $\max\{\cdot, 0\}$ is an element-wise non-negativity operation. With $\eta$ as the step size in dual gradient descent, given any $\mu > 0$, we have*

$$\frac{1}{K}\sum_{k=1}^{K}\langle \mu_k - \mu, b_k\rangle \leq \eta\frac{1}{K}\sum_{k=1}^{K}\|b_k\|^2 + \frac{1}{2K\eta}\|\mu - \mu^{(1)}\|^2 \tag{34}$$

*Proof.* For any dimension $j$ such that $\mu_{k,j} \geq \eta b_{k,j}$, we have $\mu_{k+1,j} = \mu_{k,j} - \eta \cdot b_{k,j}$, so it holds for any $\mu > 0$ that

$$\left(b_{k,j} + \frac{1}{\eta}(\mu_{k+1,j} - \mu_{k,j})\right)(\mu_j - \mu_{k+1,j}) = 0 \tag{35}$$

For any dimension $j$ such that $\mu_{k,j} < \eta b_{k,j}$, we have $\mu_{k+1,j} = 0$, and it holds for any $\mu > 0$ that

$$\left(b_{k,j} + \frac{1}{\eta}(\mu_{k+1,j} - \mu_{k,j})\right)(\mu_j - \mu_{k+1,j}) = (b_{k,j} - \frac{\mu_{k,j}}{\eta})\mu_j \geq (b_{k,j} - \frac{\eta b_{k,j}}{\eta})\mu_j = 0 \tag{36}$$

Combing equation 35 and equation 36 and we have for any $\mu > 0$ that

$$\left(b_k + \frac{1}{\eta}(\mu_{k+1} - \mu_k)\right)^\top (\mu - \mu_{k+1}) \geq 0 \tag{37}$$

Therefore, it holds for any $k \in [K]$ and $\mu > 0$ that

$$
\begin{aligned}
(\mu_k - \mu)^\top b_k &= (\mu_k - \mu_{k+1})^\top b_k + (\mu_{k+1} - \mu)^\top b_k \\
&\leq (\mu_k - \mu_{k+1})^\top b_k + \frac{1}{\eta}(\mu_{k+1} - \mu_k)^\top(\mu - \mu_{k+1}) \\
&= (\mu_k - \mu_{k+1})^\top b_k + \frac{1}{2\eta}\left(\|\mu - \mu_k\|^2 - \|\mu - \mu_{k+1}\|^2 - \|\mu_{k+1} - \mu_k\|^2\right) \\
&\leq \eta\|b_k\|^2 + \frac{1}{2\eta}\left(\|\mu - \mu_k\|^2 - \|\mu - \mu_{k+1}\|^2\right)
\end{aligned}
\tag{38}
$$

where the first inequality holds by equation 37, the second equality holds by three-point property, and the last inequality holds because $\|a\|^2 + \|b\|^2 \geq 2a^\top b$.

Taking the sum over $k \in [K]$, we have

$$
\sum_{k=1}^{K} \langle \mu_k - \mu, b_k \rangle \leq \eta \sum_{k=1}^{K} \|b_k\|^2 + \frac{1}{2\eta}\left(\|\mu - \mu_1\|^2\right)
\tag{39}
$$

which proves Lemma 2. $\qquad\square$

### C.2.2 BOUND OF EXPECTED OBJECTIVE IN THEOREM 1

*Proof.* Denote $F(\theta) = \mathbb{E}_{x \sim P_\theta}[f(w, x)]$ and the optimal parameter to solve the inner maximization of equation 7 is $\theta^*$. Define the dual problem of the inner maximization of equation 7 as

$$
Q(\mu) = \max_\theta F(\theta) + \mu(\epsilon - J(\theta, S_0))
\tag{40}
$$

Given any $\mu > 0$, it holds be weak duality that

$$
F(\theta^*) \leq F(\theta^*) + \mu(\epsilon - J(\theta^*, S_0)) \leq Q(\mu)
\tag{41}
$$

where the second inequality holds because $Q(\mu)$ maximizes $F(\theta) + \mu(\epsilon - J(\theta, S_0))$ over $\theta$. Thus, the average gap between the expected loss of $\theta^*$ and $\theta_k$ is

$$
\begin{aligned}
\frac{1}{K}\sum_{k=1}^{K}(F(\theta^*) - F(\theta_k)) &\leq \frac{1}{T}\sum_{t=1}^{T}(Q(\mu_k) - F(\theta_k)) \\
&= \frac{1}{K}\sum_{k=1}^{K}\mu_k(\epsilon - J(\theta_k, S_0)) \\
&\leq \eta\frac{1}{K}\sum_{k=1}^{K}(\epsilon - J(\theta_k, S_0))^2 + \frac{1}{2\eta}\frac{1}{K}\|\mu^{(1)}\|^2 \\
&\leq \eta(\max\{\epsilon, \bar{J}\})^2 + \frac{1}{\eta}\frac{1}{K}\|\mu^{(1)}\|^2
\end{aligned}
\tag{42}
$$

where the equality holds because $\theta_k = \arg\max_\theta F(\theta) + \mu_k(\epsilon - J(\theta, S_0))$ by equation 8 and so $Q(\mu_k) = F(\theta_k) + \mu_k(\epsilon - J(\theta_k, S_0))$, the second inequality holds by Lemma 2 with the choice of $\mu = 0$, and the last inequality holds by the assumption $J(\theta_k, S_0) \leq \bar{J}$.

Choosing $\eta = \frac{\mu^{(1)}}{\sqrt{K}\max\{\epsilon, \bar{J}\}}$, it holds that

$$
\frac{1}{K}\sum_{k=1}^{K}(F(\theta^*) - F(\theta_k)) \leq \frac{1}{\sqrt{K}}\max\{\epsilon, \bar{J}\}\mu^{(1)}
\tag{43}
$$

which proves the bound of the expected loss given the uniformly selected $k \in [K]$.

### C.2.3 BOUND OF KL DIVERGENCE IN THEOREM 1

For iteration $k$, denote $b(\theta) = \epsilon - J(\theta, S_0)$ and $b_k = \epsilon - J(\theta_k, S_0)$. Denote the constraint violation on the score matching loss at round $k$ as $v_k = J(\theta_k, S_0) - \epsilon$. Denote the optimal dual variable as $\mu^* = \arg\min_\mu Q(\mu)$. Choose a dual variable $\mu_C > \mu^*$, we have the following decomposition.

$$\sum_{k=1}^{K}(\mu_k - \mu_C)b_k = \sum_{k=1}^{K}\mu_k \cdot b_k + \sum_{k=1}^{K}(-\mu_C) \cdot b_k \tag{44}$$

For the first term, we have

$$\sum_{k=1}^{K}\mu_k \cdot b_k = \sum_{k=1}^{K}F(\theta_k) + \mu_k \cdot b_k - F(\theta_k) = \sum_{k=1}^{K}Q(\mu_k) - F(\theta_k)$$

$$\geq KQ(\mu^*) - \sum_{k=1}^{K}F(\theta_k)$$

$$\geq KQ(\mu^*) - \sum_{k=1}^{K}\max_{\theta \in \{\theta | J(\theta, S_0) \leq \epsilon + v_k\}} F(\theta) \tag{45}$$

$$\geq KQ(\mu^*) - \sum_{k=1}^{K}\max_\theta (F(\theta) + \mu^*(\epsilon + v_k - J(\theta, S_0)))$$

$$= KQ(\mu^*) - KQ(\mu^*) - \mu^*\sum_{k=1}^{K}v_k = -\mu^*\sum_{k=1}^{K}v_k$$

where the first inequality is because $\mu^*$ minimizes $Q(\mu_k)$, the second inequality holds because $\theta_k \in \{\theta \mid J(\theta, S_0) \leq \epsilon + v_k\}$, the third inequality holds by weak duality for $\mu^*$. Continuing with equation 44, we have

$$\sum_{k=1}^{K}(\mu_k - \mu_C)b_k \geq \sum_{k=1}^{K}(-\mu^* v_k + \mu_C(J(\theta_k, S_0) - \epsilon)) = (-\mu^* + \mu_C)\sum_{k=1}^{K}v_k \tag{46}$$

where the equality holds because $v_k = J(\theta_k, S_0) - \epsilon$.

By Lemma 2 with the choice of $\mu = \mu_C$, we have

$$\frac{1}{K}\sum_{k=1}^{K}(\mu_k - \mu_C)b_k \leq \eta\frac{1}{K}\sum_{k=1}^{K}\|b_k\|^2 + \frac{1}{2K\eta}(\mu_C - \mu^{(1)})^2 \leq \eta(\max\{\epsilon, \bar{J}\})^2 + \frac{1}{2K\eta}(\mu_C - \mu^{(1)})^2 \tag{47}$$

If the step size is chosen as $\eta = \frac{\mu^{(1)}}{\sqrt{K}\max\{\epsilon, \bar{J}\}}$, it holds that

$$\frac{1}{K}\sum_{k=1}^{K}(\mu_k - \mu_C)b_k \leq \frac{1}{\sqrt{K}}\max\{\epsilon, \bar{J}\}\left(\mu^{(1)} + \frac{|\mu_C - \mu^{(1)}|^2}{2\mu^{(1)}}\right) \tag{48}$$

Since $\mu_C$ is larger than $\mu^*$, by equation 46, we have

$$\frac{1}{K}\sum_{k=1}^{K}v_k \leq \frac{1}{K(\mu_C - \mu^*)}\sum_{k=1}^{K}(\mu_k - \mu_C)b_k \leq \frac{\max\{\epsilon, \bar{J}\}\left(\mu^{(1)} + \frac{|\mu_C - \mu^{(1)}|^2}{2\mu^{(1)}}\right)}{\sqrt{K}(\mu_C - \mu^*)} = \frac{C_4}{\sqrt{K}} \tag{49}$$

which means $\frac{1}{K}\sum_{k=1}^{K}J(\theta_k, S_0) \leq \epsilon + \frac{C_4}{\sqrt{K}}$.

Since $J(\theta_k, S_0)$ is the denoising score matching loss $J_{DSM}(\theta, r(\cdot)^2)$, by Lemma 1, we complete the proof by

$$\frac{1}{K}\sum_{k=1}^{K}D_{\mathrm{KL}}(P_0, P_{\theta_k}) \leq \frac{1}{K}\sum_{k=1}^{K}J(\theta_k, S_0) + D_{\mathrm{KL}}(P_T\|\pi) + C_1$$

$$\leq \epsilon + \frac{C_4}{\sqrt{K}} + D_{\mathrm{KL}}(P_T\|\pi) + C_1 \tag{50}$$

where $C_1 = \int_{t=0}^{T} r(t)^2 \left( \mathbb{E}_{P_{0,t}(x_0,x_t)} \left[ \frac{1}{2} \|\nabla_{x_t} \log P_t(x_t)\|_2^2 - \frac{1}{2} \|\nabla_{x_t} \log P_{t|0}(x_t \mid x_0)\|_2^2 \right] \right) dt$, $C_4 = \frac{\max\{\epsilon, \bar{J}\} \left( \mu^{(1)} + \frac{|\mu_C - \mu^{(1)}|^2}{2\mu^{(1)}} \right)}{(\mu_C - \mu^*)}$ with $\mu_C > \mu^*$.   $\square$

## C.3 PROOF OF THEOREM 2

Theorem 1 relies on the following assumptions.

**Assumption 1.** *The objective function $f(w,x)$ is $L-$Lipschitz with respect to $w$, i.e. for any $x \in \mathcal{X}$, $w, w' \in \mathcal{W}$, it holds that*
$$|f(w,x) - f(w',x)| \leq L \cdot \|w - w'\|_2.$$
*Moreover, $f(w,x)$ is bounded by $\bar{f}$, i.e. for any $x \in \mathcal{X}$, $w \in \mathcal{W}$, $\|f(w,x)\| \leq \bar{f}$.*

**Assumption 2.** *The objective function $f(w,x)$ is $\beta-$smooth with respect to $w$, i.e. for any $x \in \mathcal{X}$, $w, w' \in \mathcal{W}$, it holds that*
$$\|\nabla_w f(w,x) - \nabla_w f(w',x)\|_2 \leq \beta \cdot \|w - w'\|_2.$$

**Theorem 2.** *Assume that the DRO objective $f(w,x)$ is $\beta-$smooth and $L-$Lipschitz with respect to $w$ and is upper bounded by $\bar{f}$. If each sampled dataset $S_j$ from diffusion model has $n$ samples and the step size for minimization is chosen as $\lambda \sim \mathcal{O}(\sqrt{\frac{1}{\beta L^2 H}})$, then with probability $1 - \delta, \delta \in (0,1)$, the average Moreau envelope of the optimal inner maximization function $\phi(w) := \max_\theta \mathbb{E}_{P_\theta}[f(w,x)]$ satisfies*

$$\mathbb{E}\left[ \|\nabla \phi_{\frac{1}{2\beta}}(w)\|^2 \right] \leq 4\beta\Delta' + \frac{V_1}{\sqrt{n}} + \frac{V_2}{\sqrt{H}} \tag{51}$$

*where $w$ is the output ML parameter by Algorithm 1, $\phi_{\frac{1}{2\beta}}(w) := \min_{w'} \phi(w') + \beta\|w - w'\|^2$ is the Moreau envelope of $\phi$, $\Delta'$ is the error of inner maximization bounded in Theorem 1, $V_1 = 8\beta\bar{f}\sqrt{\log(2/\delta)}$ and $V_2 = 4L\sqrt{(\phi_{\frac{1}{2\beta}}(w_1) - \min_w \phi(w))\beta}$.*

*Proof.* The proof of Theorem 2 follows the techniques of Jin et al. (2020) with the difference that the maximization oracle is an empirical approximation. The optimization variable $w$ is updated as $w_j = w_{j-1} - \lambda \cdot \nabla_w \mathbb{E}_{x \in S_j}[f(w_{j-1},x)]$. Define $\phi(w) := \max_\theta \mathbb{E}_{P_\theta}[f(w,x)]$ as the inner maximization function and define $\phi_\rho(w) := \min_{w'} \phi(w') + \frac{1}{2\rho}\|w - w'\|^2$ as the Moreau envelope of $\phi$.

Denote $\hat{w}_j = \arg\min_w \phi(w) + \beta\|w - w_j\|^2$ as the proximal point of the Moreau envelope $\phi_{\frac{1}{2\beta}}(w)$. Since $f$ is $\beta-$smooth, we have

$$f(\hat{w}_j, x) \geq f(w_j, x) + \langle \nabla_w f(w_j, x), \hat{w}_j - w_j \rangle - \frac{\beta}{2}\|\hat{w}_j - w_j\|^2 \tag{52}$$

Thus, it holds with probability at least $1 - \delta, \delta \in (0,1)$ that

$$\begin{aligned}
\phi(\hat{w}_j) \geq \mathbb{E}_{P_{\theta_j}}[f(\hat{w}_j, x)] &\geq \mathbb{E}_{S_j}[f(\hat{w}_j, x)] - \frac{\bar{f}\sqrt{\log(2/\delta)}}{\sqrt{n}} \\
&\geq \mathbb{E}_{S_j}[f(w_j, x)] + \langle g(w_j, S_j), \hat{w}_j - w_j \rangle - \frac{\beta}{2}\|\hat{w}_j - w_j\|^2 - \frac{\bar{f}\sqrt{\log(2/\delta)}}{\sqrt{n}} \\
&\geq \mathbb{E}_{P_{\theta_j}}[f(w_j, x)] + \langle g(w_j, S_j), \hat{w}_j - w_j \rangle - \frac{\beta}{2}\|\hat{w}_j - w_j\|^2 - \frac{2\bar{f}\sqrt{\log(2/\delta)}}{\sqrt{n}} \\
&\geq \phi(w_j) - \Delta'_j + \langle g(w_j, S_j), \hat{w}_j - w_j \rangle - \frac{\beta}{2}\|\hat{w}_j - w_j\|^2 - \frac{2\bar{f}\sqrt{\log(2/\delta)}}{\sqrt{n}}
\end{aligned} \tag{53}$$

where the second inequality holds by applying McDiarmid's inequality on $\mathbb{E}_{P_{\theta_j}}[f(\hat{w}_j, x)]$ and $\bar{f}$ is the upper bound of $f$, the third inequality holds by equation 52 and $g(w_j, S_j) = \mathbb{E}_{S_j}[\nabla_w f(w_j, x)]$,

the forth inequality holds by applying McDiarmid's inequality on $\mathbb{E}_{S_j}[f(w_j, x)]$, and the last inequality holds by Theorem 1.

Now, it holds that

$$
\begin{aligned}
& \phi_{\frac{1}{2\beta}}(w_{j+1}) \\
&= \min_{w'} \phi(w') + \beta\|w_{j+1} - w'\|^2 \\
&\leq \phi(\hat{w}_j) + \beta\|w_j - \lambda g(w_j, S_j) - \hat{w}_j\|^2 \\
&= \phi(\hat{w}_j) + \beta\|w_j - \hat{w}_j\|^2 + 2\beta\lambda < g(w_j, S_j), \hat{w}_j - w_j > + \lambda^2\beta\|g(w_j, S_j)\|^2 \\
&\leq \phi_{\frac{1}{2\beta}}(w_j) + 2\beta\lambda < g(w_j, S_j), \hat{w}_j - w_j > + \lambda^2\beta\|g(w_j, S_j)\|^2 \\
&\leq \phi_{\frac{1}{2\beta}}(w_j) + 2\beta\lambda\left(\phi(\hat{w}_j) - \phi(w_j) + \Delta' + \frac{\beta}{2}\|\hat{w}_j - w_j\|^2 + \frac{2\bar{f}\sqrt{\log(2/\delta)}}{\sqrt{n}}\right) + \lambda^2\beta L^2
\end{aligned}
\tag{54}
$$

where the last inequality holds by equation 53.

Summing up inequality equation 54 from $j = 1$ to $j = H$, we have

$$
\begin{aligned}
& \phi_{\frac{1}{2\beta}}(w_{j+1}) \\
&\leq \phi_{\frac{1}{2\beta}}(w_1) + 2\beta\lambda\sum_{j=1}^{H}\left(\phi(\hat{w}_j) - \phi(w_j) + \Delta'_j + \frac{\beta}{2}\|\hat{w}_j - w_j\|^2 + \frac{2\bar{f}\sqrt{\log(2/\delta)}}{\sqrt{n}}\right) + \lambda^2\beta L^2 H
\end{aligned}
\tag{55}
$$

and we further have

$$
\begin{aligned}
& \frac{1}{H}\sum_{j=1}^{H}\left(\phi(w_j) - \phi(\hat{w}_j) - \frac{\beta}{2}\|\hat{w}_j - w_j\|^2\right) \\
&\leq \Delta'_j + \frac{2\bar{f}\sqrt{\log(2/\delta)}}{\sqrt{n}} + \frac{\phi_{\frac{1}{2\beta}}(w_1) - \min_w \phi(w)}{2\beta\lambda H} + \frac{\lambda L^2}{2}
\end{aligned}
\tag{56}
$$

Also, it holds that

$$
\begin{aligned}
& \phi(w_j) - \phi(\hat{w}_j) - \frac{\beta}{2}\|\hat{w}_j - w_j\|^2 \\
&= \phi(w_j) + \beta\|w_j - w_j\|^2 - \phi(\hat{w}_j) - \beta\|\hat{w}_j - w_j\|^2 + \frac{\beta}{2}\|\hat{w}_j - w_j\|^2 \\
&\geq \frac{\beta}{2}\|\hat{w}_j - w_j\|^2 = \frac{1}{4\beta}\|\nabla\phi_{\frac{1}{2\beta}}(w_j)\|^2
\end{aligned}
\tag{57}
$$

where the inequality holds by the definition of $\hat{w}_j$, and the last equality holds by the property of Moreau envelope such that $\frac{1}{2\beta}\nabla\phi_{\frac{1}{2\beta}}(w) = w - \hat{w}_j$ for any $w \in \mathcal{W}$. Therefore, we have

$$
\begin{aligned}
& \frac{1}{H}\sum_{j=1}^{H}\|\nabla\phi_{\frac{1}{2\beta}}(w_j)\|^2 \\
&\leq 4\beta\frac{1}{H}\sum_{j=1}^{H}\Delta'_j + \frac{8\beta\bar{f}\sqrt{\log(2/\delta)}}{\sqrt{n}} + \frac{2(\phi_{\frac{1}{2\beta}}(w_1) - \min_w \phi(w))}{\lambda H} + 2\beta\lambda L^2
\end{aligned}
\tag{58}
$$

By optimally choosing $\lambda = \sqrt{\frac{\phi_{\frac{1}{2\beta}}(w_1) - \min_w \phi(w)}{\beta L^2 H}}$, we have

$$
\begin{aligned}
& \mathbb{E}\left[\|\nabla\phi_{\frac{1}{2\beta}}(w_j)\|^2\right] \\
&\leq 4\beta\Delta' + \frac{8\beta\bar{f}\sqrt{\log(2/\delta)}}{\sqrt{n}} + 4L\sqrt{\frac{(\phi_{\frac{1}{2\beta}}(w_1) - \min_w \phi(w))\beta}{H}}
\end{aligned}
\tag{59}
$$

$\square$

### C.3.1 EXPLANATION OF CONVERGENCE BY MOREAU ENVELOPE

The gradient bound of Moreau envelope indicates that the algorithm converges to an approximately stationary point, which is explained as below. The Moreau envelope $\nabla\phi_{\frac{1}{2\beta}}(w_j)$ satisfies $\nabla\phi_{\frac{1}{2\beta}}(w_j) = 2\beta \cdot (w_j - \hat{w}_j)$. Since the proximal point is $\hat{w}_j = \arg\min_w \phi(w) + \beta\|w - w_j\|^2$, we have $\nabla\phi(\hat{w}_j) + 2\beta(\hat{w}_j - w_j) = 0$. Thus, it holds that $\nabla\phi_{\frac{1}{2\beta}}(w_j) = 2\beta \cdot (w_j - \hat{w}_j) = \nabla\phi(\hat{w}_j)$ and $\|\hat{w}_j - w_j\| = \|\frac{1}{2\beta}\nabla\phi(\hat{w}_j)\| = \frac{1}{2\beta}\|\nabla\phi_{\frac{1}{2\beta}}(w_j)\|$. Therefore, if the gradient of Moreau envelope is bounded for the decision variable $w$, i.e. $\|\nabla\phi_{\frac{1}{2\beta}}(w)\| \leq \Delta$, its proximal point $\hat{w}$ is an approximately stationary point for the optimal inner maximization function $\phi$ (bounded gradient of the inner maximization function $\|\nabla\phi(\hat{w})\| \leq \Delta$), and the distance between $w$ and $\hat{w}$ is close enough: $\|\hat{w}_j - w_j\| = \frac{1}{2\beta}\|\nabla\phi_{\frac{1}{2\beta}}(w_j)\| \leq \frac{\Delta}{2\beta}$. Thus, Algorithm 1 approximately converges.

### C.4 ANALYSIS ON THE EFFECTS OF THE CAPACITY OF GENERATIVE MODELS

**Corollary 1.** *Let $\tilde{P} = \arg\max_{\mathcal{P}} \mathbb{E}_{x\sim P}[f(w,x)]$ be the worst-case distribution from the set of possible testing distributions $\mathcal{P}$ given a variable $w$. Assume that the generative model $P_\theta$ is $\Gamma-$expressive (Definition 1), and $\epsilon$ is large enough such that there exists a $\tilde{\theta}$ which satisfies $J(\tilde{\theta}, P_0) \leq \epsilon$ in equation 7 and $D_W(P_{\tilde{\theta}}, \tilde{P}) \leq \Gamma$. If the objective function $f(w,x)$ is $L_x-$Lipschitz continuous with respect to $x$, the expected score-matching loss is bounded as $J(\theta) \leq \bar{J}$ and the step size is chosen as $\eta \sim \mathcal{O}(\frac{1}{\sqrt{K}})$, the overall error of inner maximization is*

$$\tilde{\Delta}' := \mathbb{E}_{x\sim\tilde{P}}[f(w,x)] - \mathbb{E}_k\mathbb{E}_{x\sim P_{\theta_k}}[f(w,x)] \leq \frac{1}{\sqrt{K}}\max\{\epsilon, \bar{J}\}\|\mu^{(1)}\| + L_x\Gamma \tag{60}$$

*where $L_x$ is the Lipschitz continuity parameter of $f(w,x)$ with respect to $x$. In addition, the inclusive KL-divergence with respect to the nominal distribution is bounded as*

$$\mathbb{E}_k[D_{KL}(P_0\|P_{\theta_k})] \leq \epsilon + \frac{\max\{\epsilon, \bar{J}\}|\mu_C - \mu^{(1)}|}{\sqrt{K}(\mu_C - \mu^*)} + D_{KL}(P_T\|\pi) + C_1, \tag{61}$$

*where $\mu^{(1)}$ is initialized dual variable, $\mu_C > \mu^*$ and $C_1$ are constants, $P_T$ is the output distribution of the forward process, and $\pi$ is the initial distribution of the reverse process.*

*Proof.* Let $\theta^*$ be the optimal diffusion parameter that solves the inner maximization equation 7 given a variable $w$. By Theorem 1, the inner maximization error holds that

$$\Delta' := \mathbb{E}_{x\sim P_{\theta^*}}[f(w,x)] - \mathbb{E}_k\mathbb{E}_{x\sim P_{\theta_k}}[f(w,x)] \leq \frac{1}{\sqrt{K}}\max\{\epsilon, \bar{J}\}\|\mu^{(1)}\| \tag{62}$$

where the outer expectation is taken over the randomness of output selection. In addition, the *inclusive* KL-divergence with respect to the nominal distribution is bounded as

$$\mathbb{E}_k[D_{KL}(P_0\|P_{\theta_k})] \leq \epsilon + \frac{\max\{\epsilon, \bar{J}\}|\mu_C - \mu^{(1)}|}{\sqrt{K}(\mu_C - \mu^*)} + R(\pi, P_0) + C_1. \tag{63}$$

That means the KL divergence with respect to the nominal distribution $P_0$ is bounded in the same way as Theorem 1.

It remains to bound the overall error in equation 60. Since $\tilde{\theta}$ satisfies $J(\tilde{\theta}, P_0) \leq \epsilon$, it is within the ambiguity set defined in equation 7, so it holds that

$$\mathbb{E}_{x\sim P_{\theta^*}}[f(w,x)] \geq \mathbb{E}_{x\sim P_{\tilde{\theta}}}[f(w,x)]. \tag{64}$$

Moreover, since $\tilde{\theta}$ satisfies $D_W(P_{\tilde{\theta}}, \tilde{P}) \leq \Gamma$, it holds by Kantorovich-Rubinstei Lemma (Kantorovich & Rubinshtein (1958), Theorem 3.2 in Mohajerin Esfahani & Kuhn (2018)) that

$$\mathbb{E}_{x\sim\tilde{P}}[f(w,x)] - \mathbb{E}_{x\sim P_{\tilde{\theta}}}[f(w,x)] \leq L_x\Gamma. \tag{65}$$

Combing the above two inequalities with the inner maximization error in equation 62, we can bound the overall maximization error $\tilde{\Delta}'$ in equation 60. $\qquad\square$

**Remark 1.** Corollary 1 shows the effect of the expressive capacity of the generative model on the performance of GAS-DRO. When the expressivity parameter $\Gamma$ is smaller, the generative model has a stronger ability to approximate the possible testing distributions. In this case, the inner maximization in GAS-DRO can identify adversarial distributions whose objective value is closer to that of the true worst-case distribution $\tilde{P}$. Consequently, by optimizing the objective parameter $w$ with respect to the parameterized adversarial distribution, GAS-DRO can achieve better robustness under the true worst-case distribution in the testing distribution set $\mathcal{P}$. If the expressivity parameter of the generative model is $\Gamma = 0$, the generative model can approximate the possible testing distributions arbitrarily well. In this ideal case, the overall error bound of the inner maximization coincides with the inner maximization error bound in Theorem 1, which asymptotically converges to zero as the number of inner iterations $K$ increases.

# D    EXPERIMENTS ON TIME SERIES PREDICTION

## D.1    DATASETS

The experiments are conducted based on Electricity Maps (2025), a widely utilized global platform that provides high-resolution spatio-temporal data on electricity system operations, including carbon intensity ($gCO_2$eq/kWh) and energy mix, and is actively employed for carbon-aware scheduling and carbon footprint estimation in real systems such as data centers.

We utilize datasets from Electricity Maps that record hourly electricity carbon intensity over the period 2021~2024 across four representative regions: California, United States (**BANC_21~24**), Texas, United States (**ERCO_21~24**), Queensland, Australia (**QLD_21~24**), and the United Kingdom (**GB_21~24**). These datasets capture fine-grained temporal variations in carbon intensity (measured in $gCO_2$eq/kWh) arising from different energy mixes in diverse geographical and regulatory contexts, thereby providing a comprehensive benchmark for evaluating carbon-aware forecasting models. For model training, we construct a dataset by merging **BANC_23** and **BANC_24**, resulting in 438 sequence samples. Model evaluation is then performed on multiple independent test sets, each consisting of 312 sequence samples drawn from other years and regions to ensure heterogeneous and challenging testing scenarios. To quantify the distributional shift between the training and test sets, we compute the Wasserstein distance, which provides a principled measure of discrepancy between probability distributions. The calculated distances are reported alongside the dataset names in Table 1.

## D.2    BASELINES

The baselines which are compared with GAS-DRO in our experiments are introduced as below.

**ML**: This method trains the standard ML to minimize the time series forecasting error without DRO.

**DML**: DML represents an ML model fine-tuned with diffusion-generated augmented datasets. Compared to GAS-DRO, DML performs standard training based on the augmented datasets rather than a distributionally robust training.

**W-DRO**: W-DRO represents a Wasserstein-based DRO framework, in which the ambiguity set is characterized by the Wasserstein metric. For our experiments, we employ the FWDRO algorithm proposed in Staib & Jegelka (2017), which transforms the inner maximization of W-DRO into an adversarial optimization with a mixed norm ball and then alternatively solves the adversarial examples and the ML weights.

**KL-DRO**: KL-DRO represents a KL-divergence-based DRO framework, in which the ambiguity set is characterized by the KL- divergence. We employ the standard KL-DRO solution derived in Hu & Hong (2013), which is commonly adopted in practice.

**DRAGEN**: DRAGEN represents a W-DRO framework, in which the ambiguity set is constructed as a Wasserstein ball in the latent space of a learned generative environment model(VAE) Ren & Majumdar (2022).

**P-DRO**: P-DRO represents a DRO framework that parameterizes the adversarial distributions using a neural generative model within a KL-divergence–based ambiguity set. Michel et al. (2021)

### D.3 TRAINING SETUPS

**Predictor**: The predictors in `GAS-DRO` and all the baselines share the same two-layer stacked LSTM architecture with 128 and 64 hidden neurons.

**Diffusion Model**: The diffusion models in `GAS-DRO` and `P-DRO` are both DDPM Ho et al. (2020) which has $T = 500$ steps in a forward or a backward process. The U-Net architecture that the DDPM used consists of four downsampling layers with channel sizes of 128, 256, 256, and 256, respectively.

**VAE**: The VAE model used for `DRAGEN` is built from an encoder–decoder architecture in which both components contain two convolutional layers with 16 channels followed by two MLP layers. The encoder progressively downsamples the input image through stacked convolutions and outputs the mean and log-variance of a 2-dimensional latent distribution, after which a reparameterization step is applied. The decoder then maps the latent variables back to the image space through an MLP projection followed by successive upsampling and convolutional layers.

**Training**: For `GAS-DRO`, we adopt the PPO-based reformulation in equation 10 for inner maximization. We train the reference DDPM $\theta_0$ in equation 10 based on the original training dataset **BANC_2324** ( **BANC_23** & **BANC_24**) and use it to generate an initial dataset $z_0$ to calculate $r_\theta$ in equation 10. The sampling variance of DDPM is chosen from a range $[0.1, 0.5]$. To improve training efficiency, only the last $T' = 15$ backward steps of the DDPM model are fine-tuned by equation 10. We choose a slightly higher clipping parameter $\kappa = 0.4$ in equation 10 to encourage the maximization while maintaining stability. We choose $\epsilon = 0.015$ as `InnerMax`'s adversarial budget which gives the best average performance over all validation datasets. We choose $\eta = 0.01$ as the rate to update the Lagrangian parameter $\mu$ in Algorithm 1. We use the Adam optimizer with a learning rate of $1 \times 10^{-5}$ for both the diffusion training in the maximization and the predictor update in minimization. The diffusion model is trained for 10 inner epochs with a batch size of 64. The predictor is trained for 15 epochs with a batch size of 64.

For the baseline methods, we choose the same neural network architecture as `GAS-DRO`. We carefully tuned the hyperparameters of the baseline algorithms to achieve optimal average performance over all validation datasets. For `W-DRO`, we consider the Wasserstein distance with respect to $l_2$−norm and set the adversarial budget as $\epsilon = 0.3$ with a learning rate of $5 \times 10^{-4}$. For `KL-DRO`, we choose the adversarial budget $\epsilon = 4$ with a learning rate of $5 \times 10^{-4}$. For `DRAGEN`, we adopt the Wasserstein distance with respect to $l_2$−norm and set the adversarial budget as $\epsilon = 2$ with a learning rate of $5 \times 10^{-5}$. It is worth noting that, we pre-trained a well-performing VAE for `DRAGEN` with a learning rate of $1 \times 10^{-2}$, which achieved an average Wasserstein distance of 0.0103 on data reconstruction over the training set. For `P-DRO`, its generative model uses the exact same DDPM architecture as `GAS-DRO`. We set the temperature as $\tau = 0.1$ and set the learning rate as $1 \times 10^{-5}$ for the inner maximization and the outer minimization. In addition, `P-DRO` maintains a window of consecutive minibatches for updates of the Normalizer, and the window length is set as 5. Finally, same as `GAS-DRO`, the diffusion model in `P-DRO` is trained for 10 inner epochs with a batch size of 64 while the predictor is trained for 15 epochs with a batch size of 64.

### D.4 EXPERIMENTAL RESULTS ON TIME SERIES PREDICTION

#### D.4.1 MIAN RESULTS

We tune the hyperparameters of all methods based on validation datasets and present their best performance in Table 1 where all the datasets are OOD testing datasets with different discrepancies.

We can find that all DRO methods improve the testing performance for OOD datasets comparing to `ML` by optimizing the worst-case expected loss while `DML` improves upon `ML` by leveraging diffusion-generated augmented datasets. Using `ML` as the reference baseline, `GAS-DRO` achieves the highest performance gain of 63.7%, followed by `DRAGEN`, `P-DRO` and `DML` with improvements of 48.9%, 42.5% and 39.7%, respectively. In contrast, `KL-DRO` and `W-DRO` yield improvements of 36.1% and 24.0%, respectively.

Notably, `DML` surpasses `KL-DRO` by 3.6% only after augmentation training with diffusion-generated datasets, highlighting the significant positive impact of diffusion model. Moreover, `DML` differs from `GAS-DRO` only in that it disables the DRO component, while their training procedures remain identi-

cal. Thus, `GAS-DRO`, `DML`, and `ML` together form two ablation studies: the performance gap between `GAS-DRO` and `DML` confirms that DRO contributes a 39.7% performance gain to `GAS-DRO`, while the gap between `DML` and `ML` verifies that the diffusion model also provides a 39.7% performance gain.

Although both `DML` and the other four DRO methods deliver noticeable performance gains, `GAS-DRO` still outperforms them by an average margin of 25.4% and by 27.8% in the worst case. The underlying reason may lie in the fact that `GAS-DRO` leverages the distribution learning capability of the diffusion model when constructing the ambiguity set, enabling the generation of adversarial distributions that are both strong and realistic, thereby achieving superior OOD generalization.

In contrast, although `DRAGEN` also employs a generative model to construct its ambiguity set, it still relies on relaxing Wasserstein DRO framework which is limited in overcoming the inherent limitations of `W-DRO`. In addition, `P-DRO`, as a DRO algorithm that uses generative models to model the adversarial distributions, remains restricted by the KL-divergence-based ambiguity set, preventing it from modeling diverse worst-case distributions. By contrast, `GAS-DRO` directly defines the ambiguity set on the parameterized space of generative models, enabling it to generate worst-case distributions that are both diverse and consistent with the nominal distribution, thereby offering a principled and tractable DRO algorithm.

Furthermore, the KL-divergence used in `KL-DRO` requires all distributions to be absolutely continuous with respect to the training distribution $P_0$, which constrains the support space of the ambiguity set and limits the search for strong adversaries. Among the baselines, `W-DRO` performs the worst, likely due to the fact that solving Wasserstein-based DRO usually involves optimal transport problems or their relaxations, which are computationally more demanding and often rely on approximate methods such as dual reformulation or adversarial training, resulting in suboptimal solutions. Moreover, since `W-DRO` allows adversarial distributions with supports different from the training distribution, it is theoretically more flexible and closer to real distribution shifts, but this flexibility can produce overly extreme adversaries and thus lead to overly conservative model training.

### D.4.2 EFFECT OF NOISY TYPES

In this test, we add a certain amount of different types of noise into each test set to compare the noise robustness of different algorithms. The noise types include Gaussian Noise, Perlin Noise, and Cutout Noise.

**Gaussian Noise**: In the Gaussian Noise test, we add Gaussian Noise with $\sigma = 0.1$ to each test set. As shown in Table 2, all algorithms exhibit performance degradation compared to the noise-free setting. Nevertheless, `GAS-DRO` still significantly outperforms the others: taking `ML` as the reference, our method achieves a 48.3% improvement, followed by `DRAGEN` and `DML` with gains of 38.1% and 31.7%, respectively. Among the remaining baselines, `KL-DRO` improves performance by 29.1%, while `W-DRO` is the weakest, surpassing `ML` by only 20%. In

Table 2: Gaussian-Corrupted Test.

| Dataset | MSE | | | | | |
|---|---|---|---|---|---|---|
| | GAS-DRO | DRAGEN | KL-DRO | W-DRO | DML | ML |
| BANC_22 | **0.0170** | 0.0188 | 0.0197 | 0.0183 | 0.0189 | 0.0291 |
| BANC_21 | **0.0177** | 0.0199 | 0.0214 | 0.0216 | 0.0199 | 0.0337 |
| QLD_24 | **0.0560** | 0.0716 | 0.0839 | 0.0911 | 0.0847 | 0.0990 |
| QLD_23 | **0.0615** | 0.0766 | 0.0925 | 0.0960 | 0.0934 | 0.1034 |
| QLD_22 | **0.0307** | 0.0402 | 0.0476 | 0.0639 | 0.0445 | 0.0766 |
| QLD_21 | **0.0297** | 0.0394 | 0.0475 | 0.0649 | 0.0431 | 0.0772 |
| GB_24 | **0.0238** | 0.0253 | 0.0280 | 0.0276 | 0.0279 | 0.0381 |
| GB_23 | **0.0221** | 0.0258 | 0.0260 | 0.0305 | 0.0269 | 0.0414 |
| GB_22 | **0.0227** | 0.0263 | 0.0302 | 0.0343 | 0.0279 | 0.0458 |
| GB_21 | **0.0211** | 0.0256 | 0.0279 | 0.0334 | 0.0256 | 0.0438 |
| ERCO_24 | **0.0281** | 0.0317 | 0.0350 | 0.0369 | 0.0340 | 0.0471 |
| ERCO_23 | **0.0228** | 0.0251 | 0.0281 | 0.0294 | 0.0267 | 0.0418 |
| ERCO_22 | **0.0206** | 0.0231 | 0.0256 | 0.0293 | 0.0230 | 0.0416 |
| ERCO_21 | **0.0217** | 0.0244 | 0.0295 | 0.0357 | 0.0264 | 0.0475 |
| Average | **0.0282** | 0.0338 | 0.0388 | 0.0438 | 0.0373 | 0.0547 |
| Worst | **0.0615** | 0.0766 | 0.0925 | 0.0960 | 0.0934 | 0.1034 |

proves performance by 29.1%, while `W-DRO` is the weakest, surpassing `ML` by only 20%. In fact, `W-DRO` is relatively adept at handling Gaussian Noise compared to other noise types, and thus maintains a noticeable advantage even under $\sigma = 0.1$ Gaussian Noise, a trend further confirmed in subsequent experiments. On the other hand, both `GAS-DRO` and `DML` are trained with diffusion-generated augmented datasets, which naturally incorporate Gaussian-based noise during the diffusion process. This exposure enables them to remain robust under Gaussian noise perturbations. In contrast, `DRAGEN` 's performance, which falls between `GAS-DRO` and `DML`, can be attributed to the property of the VAE: although Gaussian noise in the input space appears random, it is often compressed and filtered when mapped into the latent space, thereby preventing severe disruption of latent representations. Consequently, `DRAGEN` also performs reasonably well under Gaussian noise, though not as strongly as `GAS-DRO`.

**Perlin Noise**: Perlin Noise is a smooth pseudo-random gradient noise commonly used to simulate natural textures such as clouds, terrains, and wood grains. By combining multiple Perlin Noise components with different frequencies and amplitudes (known as octaves), more complex fractal noise can be produced. In this experiment, we superimpose 8 layers of Perlin Noise, with the noise amplitude normalized to the range [-1, 1].

As shown in Table 3, taking `ML` as the reference, our method outperforms `ML` by 76.4%, while `DRAGEN` achieves a 66.3% improvement, and both perform better than in the Gaussian Noise test. `DML` consistently ranks third in performance, trailing `DRAGEN` by approximately 6.2%, a gap that is nearly identical to that observed in the Gaussian Noise test. Although `KL-DRO` also shows a larger gain relative to `ML`, its MSE remains roughly the same as in the Gaussian Noise test; the apparent improvement is primarily due to the substantial performance drop of `ML` in this experiment. In contrast, `W-DRO` outperforms `ML` by only 9.6%, a significant decline compared to its performance under Gaussian Noise, with the MSE reduced by as much as 50%. This indicates that `W-DRO`

Table 3: Perlin-Corrupted Test.

| Dataset | MSE | | | | | |
|---|---|---|---|---|---|---|
| | GAS-DRO | DRAGEN | KL-DRO | W-DRO | DML | ML |
| **BANC_22** | **0.0117** | 0.0111 | 0.0296 | 0.0620 | 0.0227 | 0.0663 |
| **BANC_21** | **0.0110** | 0.0156 | 0.0288 | 0.0591 | 0.0211 | 0.0662 |
| **QLD_24** | **0.0355** | 0.0506 | 0.0636 | 0.0862 | 0.0588 | 0.0928 |
| **QLD_23** | **0.0406** | 0.0490 | 0.0685 | 0.0787 | 0.0669 | 0.0860 |
| **QLD_22** | **0.0186** | 0.0321 | 0.0476 | 0.0852 | 0.0341 | 0.0904 |
| **QLD_21** | **0.0192** | 0.0288 | 0.0475 | 0.0865 | 0.0355 | 0.0940 |
| **GB_24** | **0.0147** | 0.0152 | 0.0272 | 0.0499 | 0.0252 | 0.0563 |
| **GB_23** | **0.0133** | 0.0184 | 0.0295 | 0.0560 | 0.0239 | 0.0628 |
| **GB_22** | **0.0132** | 0.0223 | 0.0311 | 0.0611 | 0.0247 | 0.0666 |
| **GB_21** | **0.0114** | 0.0167 | 0.0311 | 0.0586 | 0.0224 | 0.0677 |
| **ERCO_24** | **0.0154** | 0.0224 | 0.0267 | 0.0436 | 0.0238 | 0.0525 |
| **ERCO_23** | **0.0142** | 0.0200 | 0.0382 | 0.0767 | 0.0262 | 0.0829 |
| **ERCO_22** | **0.0109** | 0.0195 | 0.0324 | 0.0655 | 0.0223 | 0.0732 |
| **ERCO_21** | **0.0104** | 0.0211 | 0.0283 | 0.0506 | 0.0194 | 0.0605 |
| **Average** | **0.0171** | 0.0245 | 0.0379 | 0.0657 | 0.0305 | 0.0727 |
| **Worst** | **0.0406** | 0.0506 | 0.0685 | 0.0865 | 0.0669 | 0.0940 |

is not well-suited for handling Perlin Noise, likely because the Wasserstein distance measures global distributional transport cost and is more effective in capturing smooth, small perturbations (e.g., Gaussian Noise), but fails to adequately model the long-range correlated patterns of Perlin Noise within the Wasserstein ball.

**Cutout Noise**: Cutout Noise is a commonly used perturbation method that randomly selects a region of the input data and sets its values to a constant, thereby simulating partial information loss. In our experiment, we randomly mask 30% of the sequence and set the masked values to a constant of 1. As shown in Table 4, the performance of all algorithms is very close to their performance under Perlin Noise. We attribute this to the fact that, although Perlin and Cutout Noises differ in form, both represent structured local perturbations that disrupt the continuity of the input patterns, thereby posing similar challenges to all algorithms and resulting in comparable performance under these two types of noise at a given perturbation level. This is further confirmed in the subsequent gradient-perturbation tests.

Table 4: Cutout-Corrupted Test.

| Dataset | MSE | | | | | |
|---|---|---|---|---|---|---|
| | GAS-DRO | DRAGEN | KL-DRO | W-DRO | DML | ML |
| **BANC_22** | **0.0063** | 0.0164 | 0.0181 | 0.0297 | 0.0125 | 0.0385 |
| **BANC_21** | **0.0095** | 0.0177 | 0.0302 | 0.0547 | 0.0192 | 0.0637 |
| **QLD_24** | **0.0404** | 0.0604 | 0.0798 | 0.1096 | 0.0697 | 0.1148 |
| **QLD_23** | **0.0426** | 0.0644 | 0.0802 | 0.1028 | 0.0743 | 0.1092 |
| **QLD_22** | **0.0196** | 0.0354 | 0.0507 | 0.0881 | 0.0384 | 0.0971 |
| **QLD_21** | **0.0208** | 0.0294 | 0.0534 | 0.0916 | 0.0400 | 0.1005 |
| **GB_24** | **0.0145** | 0.0165 | 0.0351 | 0.0598 | 0.0252 | 0.0688 |
| **GB_23** | **0.0122** | 0.0215 | 0.0343 | 0.0603 | 0.0230 | 0.0681 |
| **GB_22** | **0.0129** | 0.0228 | 0.0361 | 0.0639 | 0.0245 | 0.0720 |
| **GB_21** | **0.0127** | 0.0225 | 0.0356 | 0.0651 | 0.0245 | 0.0740 |
| **ERCO_24** | **0.0157** | 0.0214 | 0.0375 | 0.0643 | 0.0275 | 0.0732 |
| **ERCO_23** | **0.0134** | 0.0225 | 0.0349 | 0.0599 | 0.0241 | 0.0678 |
| **ERCO_22** | **0.0134** | 0.0162 | 0.0349 | 0.0599 | 0.0241 | 0.0678 |
| **ERCO_21** | **0.0116** | 0.0227 | 0.0354 | 0.0671 | 0.0225 | 0.0758 |
| **Average** | **0.0174** | 0.0278 | 0.0425 | 0.0699 | 0.0319 | 0.0782 |
| **Worst** | **0.0426** | 0.0644 | 0.0802 | 0.1096 | 0.0743 | 0.1148 |

### D.4.3 EFFECT OF NOISY LEVELS

In this experiment, we progressively increased the intensity of three types of noise. For Gaussian Noise, the perturbation range is set to $\sigma \in [0.05, 0.2]$; for Perlin Noise, the amplitude is controlled within the range $[0.05, 1]$; and for Cutout Noise, the Cutout Mask Ratio is adjusted between $[10\%, 40\%]$. As shown in Figures 2, 3, 4, `GAS-DRO` consistently outperforms all baseline methods across different noise types and intensity levels. With increasing noise strength, we observe that variations in Gaussian Noise have a more pronounced impact on all methods compared to Perlin and Cutout Noise. In contrast, the impact of stronger Perlin and Cutout Noise on `GAS-DRO` remains limited, and even at higher noise levels, `GAS-DRO` maintains stable and superior performance, highlighting its strong robustness. For `W-DRO`, however, the performance degradation under Perlin and Cutout Noise is much greater, with trends almost identical to `ML`, indicating that `W-DRO` is not effective in handling Perlin and Cutout Noise but is relatively better at coping with Gaus-

sian Noise. Interestingly, when the Cutout Mask Ratio is 30%, the performance of all algorithms is nearly identical to their performance under Perlin Noise with amplitude 1, suggesting that at specific noise levels, Perlin and Cutout—though different in form—both represent structured local perturbations that disrupt input continuity to a similar degree, thereby producing comparable impacts on the algorithms.

### D.4.4 EFFECTS OF ADVERSARIAL BUDGET

In this adversarial budget experiment, we examine the impact of the budget parameter $\epsilon$ in equation 7 on the performance of GAS-DRO. As illustrated in Fig. 5, the loss–$\epsilon$ curves across all datasets display a concave trend, with the best average performance achieved around $\epsilon = 0.015$. When $\epsilon$ is smaller than this threshold, the diffusion-modeled distributions are overly restricted to the training data, thereby hindering the ability of GAS-DRO to generalize to OOD datasets. In contrast, when $\epsilon$ becomes excessively large, the enlarged ambiguity set causes GAS-DRO to conservatively optimize against irrelevant distributions, which degrades its performance on real OOD datasets. Hence, selecting an appropriate value of $\epsilon$ is essential for constructing effective adversarial distributions, ensuring a proper balance between average-case and worst-case performance.

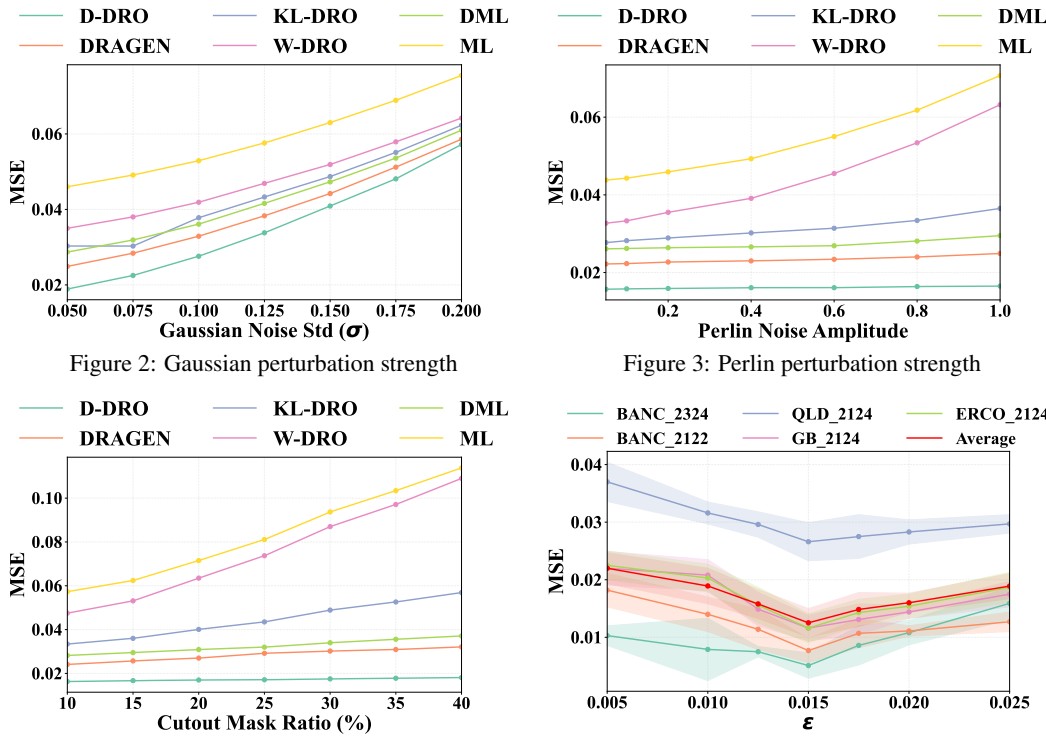

Figure 2: Gaussian perturbation strength

Figure 3: Perlin perturbation strength

Figure 4: Cutout perturbation strength

Figure 5: Effect of budget $\epsilon$ in GAS-DRO

### D.4.5 EFFECT OF THE CAPACITY OF GENERATIVE MODELS

To investigate the influence of diffusion model capacity on the performance of GAS-DRO, we vary the width of the UNet by modifying the channel dimensions at each level. Specifically, we denote the original four-level UNet with channels [128, 256, 256, 256] as **DFU-Full**. We then scale the model down to [96, 192, 192, 192], denoted as **DFU-$\frac{3}{4}$**, and further to [64, 128, 128, 128], denoted as **DFU-$\frac{1}{2}$**. In addition, we include a VAE-based variant of GAS-DRO for comparison, and the VAE architecture is given in Section D.3.

Table 5: Comparison of GAS-DRO with Different Generative Models.

| Model | Avg MSE | Worst MSE |
|---|---|---|
| **DFU-Full** | 0.0163 | 0.0509 |
| **DFU-$\frac{3}{4}$** | 0.0171 | 0.0544 |
| **DFU-$\frac{1}{2}$** | 0.0178 | 0.0608 |
| **VAE** | 0.0168 | 0.0463 |

To evaluate the effect of generative model choice, we evaluate the four generative model configurations—three diffusion models of different UNet sizes and one VAE—on all original (noise-

free) OOD datasets in the time-series forecasting task. The results in Table 5 show that, for the diffusion-based variants of GAS-DRO, increasing the UNet width leads to a clear improvement: the average MSE decreases monotonically, and the worst-case MSE is also substantially reduced as the model becomes larger. This indicates that a stronger generative model can better characterize the ambiguity set, thereby enhancing both the robustness and accuracy of GAS-DRO. The VAE-based variant achieves performance between DFU-Full and DFU-$\frac{3}{4}$, further illustrating that GAS-DRO does not depend on diffusion models specifically; rather, it is insensitive to the choice of generative architecture and exhibits strong generalization across model types.

### D.4.6 COMPUTATIONAL OVERHEAD

Finally, we tested all methods on the time series prediction task, specifically evaluating their runtime, memory overhead, and GPU memory overhead during both the training and inference phases. All methods are evaluated on a single NVIDIA RTX 6000 Ada GPU, with 48 GB of memory, and 1 TB of system storage.

The training computational overhead is shown in Table 6. GAS-DRO (DFU) represents GAS-DRO using the diffusion model, GAS-DRO (VAE) represents GAS-DRO using the VAE model, and other baselines have their names in Section D.2.

As shown in Table 6, DML has a higher memory cost than ML due to the need to store the augmented dataset. Compared to ML and DML,

Table 6: Training Cost Comparison.

| Method | Time (s) | Memory (MB) | GPU Memory (MB) |
|---|---|---|---|
| **GAS-DRO (DFU)** | 1897 | 2274 | 12239 |
| **GAS-DRO (VAE)** | 494 | 2449 | 5849 |
| **DRAGEN** | 2191 | 2355 | 434 |
| **P-DRO** | 1799 | 2205 | 11576 |
| **KL-DRO** | 149 | 2314 | 417 |
| **W-DRO** | 2896 | 2304 | 417 |
| **DML** | 130 | 2316 | 417 |
| **ML** | 143 | 1352 | 33 |

GAS-DRO (DFU) and P-DRO both introduce significant additional training time overhead primarily due to sampling processes in diffusion models. The memory overhead of GAS-DRO and P-DRO is also substantially increased due to the large size of the U-Net and the need to store sampling trajectories. However, the higher computational overhead of GAS-DRO (DFU) and P-DRO is traded for superior generalization performance across OOD testing datasets, particularly for GAS-DRO, as shown in Table 1. GAS-DRO (VAE) and DRAGEN are based on VAEs, so they have less memory footprint. However, due to the complexity to search within the Wasserstein-based ambiguity set, they have the highest time overhead. KL-DRO is the most efficient DRO algorithm in terms of time and memory due to its efficient closed-form solution.

In test phase, however, since all methods use the same ML models for time-series prediction, they have the same runtime and memory for inference. To perform inference on a testing dataset (**BANC_22**) with 8760 samples, the inference runtime is 0.1s and GPU memory is 409 MB.

## E EXPERIMENTS ON IMAGE CLASSIFICATION

### E.1 TASK

We further compare the performance of GAS-DRO with baseline methods on an image classification task. Handwritten digit classification is a long-standing and fundamental benchmark in computer vision. It captures the essential structure of real-world image recognition tasks, including high-dimensional pixel dependencies, sensitivity to corruptions, and vulnerability to distributional shifts. As such, the handwritten digit classification has been a standard testbed for evaluating the robustness and generalization ability of machine learning models. Beyond that, handwritten digit recognition supports many real-world applications, including postal mail sorting, bank check processing, form digitization, and optical character recognition (OCR) systems.

In this experiment, we adopt handwritten digit recognition to assess whether GAS-DRO can effectively handle distribution shifts and image corruptions in vision domains. The task is particularly relevant for DRO evaluation because even small perturbations—such as noise, occlusions, or contrast changes—can significantly alter the underlying data distribution. A robust DRO method should

therefore maintain high accuracy under these shifts. This makes handwritten digit classification an important and well-defined environment for comparing different designs and validating the performance of `GAS-DRO`.

## E.2 DATASETS

This experiment is a standard image classification task, where the training set is the MNIST dataset Lecun et al. (1998), and the evaluation is conducted on both MNIST and an OOD dataset USPS Hull (1994) test sets. The USPS dataset, collected by the U.S. Postal Service for automatic mail sorting, serves as a natural OOD benchmark due to its distinct handwriting styles and distributional characteristics. In addition to the original test sets, we construct more OOD datasets by applying various perturbations to both MNIST and USPS, including Gaussian noise, Perlin noise, and Cutout noise, consistent with the noise types used in our time-series forecasting experiments. As shown in Table 7, for MNIST, the corrupted variants include **M-Gaussian**, **M-Perlin**, and **M-Cutout**, corresponding to Gaussian noise with a standard deviation of 0.4, Perlin noise with amplitude 2.5, and Cutout with a cover rate of 0.3, respectively. For USPS, we adopt milder perturbations due to its inherent OOD nature, resulting in **U-Gaussian**, **U-Perlin**, and **U-Cutout**, which come with Gaussian noise with a standard deviation of 0.03, Perlin noise with amplitude 0.5, and Cutout with a cover rate of 0.05, respectively. Both **M-Original** and **U-Original** mean the original dataset of MNIST and USPS. The training is performed on **M-Original**, and the classifiers are tested on other OOD datasets. We select the first 4096 images from the MNIST training dataset as the training set for classifier training in all algorithms. For evaluation, we use the original MNIST test set consisting of 10000 images, together with 7291 images from the USPS dataset. All corrupted test sets used in our experiments are generated by applying various perturbations to these 17291 images.

During data preprocessing, we apply the standard normalization to the MNIST dataset so that the mean of each pixel is 0.1307 and variance of each pixel is 0.3081. For the USPS dataset, we use the same data preprocessing method as Ren & Majumdar (2022): Each original $16 \times 16$ grayscale digit image is reconstructed from the HDF5 file, resized to $30 \times 30$, and then centered on a $48 \times 48$ black background to match the spatial layout of MNIST-like inputs. The images are subsequently downsampled to a $14 \times 14$ resolution, paired with their corresponding labels for training and evaluation. Since the classifier used in this experiment is a lightweight two-layer DNN, all images across datasets are ultimately downsampled to a $14 \times 14$ resolution to facilitate efficient training and testing.

## E.3 BASELINES

The baselines which are compared with `GAS-DRO` in our vision experiments are introduced as below.

**ML**: This method trains the standard ML model to minimize the classification error without using DRO.

**DML**: `DML` represents an `ML` model fine-tuned with data augmented training set. Unlike the `DML` setting described in Appendix D, here we apply Gaussian noise with a standard deviation of 0.02 to augment the training set for data augmentation.

**W-DRO**: We employ the same implementation of `W-DRO` described in Appendix D, with only minor modifications to its input and output formats to accommodate training and inference on image data.

**KL-DRO**: We employ the same implementation of `KL-DRO` described in Appendix D, with only minor modifications to its input and output formats to accommodate training and inference on image data.

**DRAGEN**: We employ the same implementation of `DRAGEN` described in Appendix D, with only minor modifications to its input and output formats to accommodate training and inference on image data.

## E.4 TRAINING SETUPS

**Predictor**: The classifiers in `GAS-DRO` and all the baselines share the same two-layer DNN architecture with a hidden layer of 8 neurons followed by a 10-dimensional output layer. We use MSE

instead of cross-entropy while training the predictor, because the predictor is a very small two-layer network, and MSE provides more stable gradients.

**VAE For `DRAGEN` & `GAS-DRO`**: In this experiment, both `DRAGEN` and `GAS-DRO` use the same VAE, and its architecture is identical to the VAE employed by `DRAGEN` in the time-series forecasting task in Section D.3. The only difference is that we increase the latent dimensionality to 8 in order to preserve more image information. Specifically, the encoder is a convolutional neural network that progressively downsamples the input image and maps it into a low-dimensional latent representation. It first applies a stack of 2 convolutional layers, each with a stride of 2 and ReLU activation, where the number of channels doubles at each layer (from 1 to an initial low-width channel count and then up to the specified inner channel width). After spatial downsampling, the resulting feature map is flattened and passed through an MLP. This MLP outputs the parameters of an 8-dimensional latent Gaussian distribution, producing both the mean and log-variance vectors through two separate linear heads.

The decoder first maps the latent vector into a low-resolution convolutional feature map using an MLP, effectively expanding the compressed latent representation back into a spatial structure. This feature map is then progressively upsampled through a sequence of layers, where each layer doubles the spatial resolution using interpolation and refines the features with a 3×3 convolution followed by ReLU activation. As the resolution increases, the number of channels is gradually reduced to match the structure of a natural image. Finally, a 1×1 convolution projects the features into a single output channel, and a Sigmoid activation constrains the pixel intensities to the [0,1] range. Based on the decoder output, we can model a Gaussian distribution for the image as $p_\theta(x|z) = \mathcal{N}\left(x; \mu_\theta(z), \sigma^2 I\right)$ where $\mu_\theta(z)$ is the decoder network. Given a pretrained encoder $q_\varphi(z|x)$, the reconstruction loss of VAE $J_{\text{VAE}} = \mathbb{E}_{P_0(x)}\mathbb{E}_{q_\varphi^*(z|x)}\left[-\log p_\theta(x \mid z)\right]$ is in a linear relationship with the expected squared error based on the latent distribution, i.e. $\mathbb{E}_{P_0(x)}\mathbb{E}_{q_\varphi(z|x)}\left[\|x - \mu_\theta(z)\|^2\right]$. Thus, we directly use this expected squared loss to replace the VAE reconstruction loss in equation 25 and obtain the inner-maximization objective following equation 27.

**Training For `GAS-DRO`**: Before training `GAS-DRO`, we freeze the parameters $\varphi$ of the pretrained VAE encoder. We choose $\epsilon = 65$ as `InnerMax`'s adversarial budget which gives the best average performance over all validation datasets. We choose $\eta = 0.1$ as the rate to update the Lagrangian parameter $\mu$ in Algorithm 1. We use the Adam optimizer with a learning rate of $1 \times 10^{-4}$ for the VAE training in the maximization and a learning rate of $2 \times 10^{-4}$ for the predictor update in minimization. The VAE model is trained for 10 inner epochs with a batch size of 64. The predictor is trained for 10 epochs with a batch size of 64. During the predictor training, we encode the training data into latent variable $z$ using the VAE encoder. Then we introduce a small perturbation to the latent space with the perturbation radius as $\epsilon_z = 0.25$. Next, we generate new samples by decoding the latent variable $z$ with the updated decoder $p_\theta$.

**Training For `DRAGEN`**: For `DRAGEN`, we adopt the Wasserstein distance with respect to $l_2-$norm and set the adversarial budget as $\epsilon = 1.5$. Then a `W-DRO` is applied on the latent space with a learning rate of $5 \times 10^{-5}$.

**Training For `W-DRO`**: For `W-DRO`, we consider the Wasserstein distance with respect to $l_2-$norm and set the adversarial budget as $\epsilon = 3$ with a learning rate of $1 \times 10^{-5}$.

**Training For `KL-DRO`**: For `KL-DRO`, we choose the adversarial budget $\epsilon = 1$ with a learning rate of $5 \times 10^{-3}$.

**Training For `DML`**: The training configuration of `DML` is identical to that of `ML`; the only difference is that we replace the original training samples with their Gaussian-noised versions, where noise with a standard deviation of 0.02 is added to each image.

All methods are trained for a total of 100 epochs; for `GAS-DRO` that adopt a nested training scheme, the product of the numbers of outer and inner iterations is also set to 100.

### E.5 EXPERIMENTAL RESULTS ON IMAGE CLASSIFICATION

We conducted comparative tests on the MNIST and USPS datasets. As shown in Table 7, `GAS-DRO` achieves the best overall performance with an average accuracy of 71.52%, outperforming `DRAGEN` by 3.27%, KL-DRO by 6.24%, and standard `ML` by 9.49%, demonstrating the strongest generaliza-

tion and robustness across all distributions and corruption types. `GAS-DRO` also achieves the best worst-case performance with an accuracy of 50.58%, clearly outperforming `DRAGEN` (46.84%), `KL-DRO` (45.38%), `W-DRO` (46.11%), `DML` (43.08%) and `ML` (42.38%), making it the only method that maintains above 50% accuracy under the most challenging OOD conditions. Across the OOD testing settings, `GAS-DRO` achieves the best or near-best accuracy on MNIST and consistently ranks first in all USPS OOD scenarios, demonstrating its superiority under stronger distribution shifts.

Compared with `DML`, which benefits only from data augmentation, `GAS-DRO` leverages both generative modeling and worst-case optimization, yielding significantly superior OOD generalization. More importantly, we note that the superiority of `GAS-DRO` persists even when it is implemented with the same VAE backbone used by `DRAGEN`, which demonstrates that the advantage of `GAS-DRO` rely on the design of its ambiguity set and optimization approach. Unlike `DRAGEN`, which still

Table 7: Image Classification Test.

| Dataset | Accuracy (%) | | | | | |
|---|---|---|---|---|---|---|
| | GAS-DRO | DRAGEN | KL-DRO | W-DRO | DML | ML |
| M-Original | 85.31 | 82.71 | **86.92** | 85.70 | 79.02 | 85.63 |
| M-Gaussian | **83.13** | 81.23 | 70.26 | 74.16 | 81.70 | 65.89 |
| M-Perlin | 81.52 | **81.57** | 71.74 | 75.11 | 79.68 | 67.15 |
| M-Cutout | 62.10 | **65.22** | 59.62 | 62.86 | 61.28 | 59.31 |
| U-Original | **71.81** | 64.09 | 63.87 | 65.49 | 63.05 | 60.22 |
| U-Gaussian | **70.81** | 63.49 | 62.23 | 63.98 | 62.94 | 58.62 |
| U-Perlin | **66.86** | 60.83 | 62.24 | 62.95 | 59.51 | 57.07 |
| U-Cutout | **50.58** | 46.84 | 45.38 | 46.11 | 42.08 | 42.38 |
| Average | **71.52** | 68.25 | 65.28 | 67.05 | 66.16 | 62.03 |
| Worst | **50.58** | 46.84 | 45.38 | 46.11 | 42.08 | 42.38 |

relaxes traditional Wasserstein DRO framework to perturb the latent space of VAE, `GAS-DRO` performs inner maximization directly in the parameterized generative model space, enabling more stable convergence and the construction of more realistic adversarial distributions. Moreover, the reconstruction-loss-based ambiguity set used in `GAS-DRO` avoids the issues in traditional ambiguity modeling, including the restrictions on the support space inherent to `KL-DRO` and the overly conservative adversaries introduced by relaxing `W-DRO`. Thus, it outperforms the traditional DRO methods and the generative DRO method based on traditional ambiguity modeling.

## F   SCORE-MATCHING BASED DIFFUSION MODELS

The score-based diffusion models learn the distributions through a forward process and a backward process. We introduce a score-based generative modeling by Stochastic Differential Equations (SDEs) Song et al. (2021).

**Forward Process.** The forward process incrementally injects noise into the data, generating a sequence of perturbed samples. It begins with an initial sample $X_0 \in \mathcal{R}^d$ drawn from a training dataset $S_0$, and evolves according to SDE defined as:

$$\mathrm{d}x = b(x,t)dt + g(t)dw \tag{66}$$

where $b(\cdot,t) : \mathcal{R}^d \to \mathcal{R}^d$ is a drift coefficient, $g(t) \in \mathcal{R}$, and $w$ is a standard Wiener process. By the SDE, we get variable $x_t$ which represents the data corrupted by noise at time $t$. We use $P_t$ to represent the distribution of $x_t$ and $P_{t|0}$ to denote the conditional distribution of $x_t$ given $x_0$. In the framework of score-based diffusion models, the forward diffusion process terminates at a sufficiently large time $T$, so that the marginal distribution $P_T$ approximates a tractable distribution $p'$ which is typically chosen as the standard Gaussian distribution $\mathcal{N}(0, \mathbf{I}_d)$.

**Backward Process.** By reversing the forward process in time, it's possible to recover the original data distribution starting from pure noise. The backward SDE reverses the time evolution of the forward equation in equation 66:

$$\mathrm{d}x = \left(b(x,t) - g(t)^2 \nabla_x \log P_t(x)\right) \mathrm{d}t + g(t)\mathrm{d}\bar{w} \tag{67}$$

where $\bar{w}$ is a standard Wiener process in the reverse-time direction, $\nabla_x \log P_t(x)$ is the time-dependent score function relying on the marginal probability density of $x_t$ in the forward process.

**Score Matching.** In the backward SDE, the score function $\nabla \log P_t(\cdot)$ plays a critical role in directing the generative dynamics. To estimate the score function $\nabla \log P_t(\cdot)$, we train a score-based model $s_\theta(x,t)$ based on samples generated from the forward diffusion process. The score-based model is optimized by minimizing the following denoising score-matching loss given a dataset $S_0$:

$$J(\theta, S_0) = \int_0^T \mathbb{E}_{x_0 \sim P_0} \mathbb{E}_{x_t \sim P_{t|0}} \left[\iota(t) \left\| \nabla_{x_t} \log P_{t|0}(x_t \mid x_0) - s_\theta(x_t,t) \right\|^2 \right] \mathrm{d}t \tag{68}$$

where $\iota(t) > 0$ is a weighting function.

## F.1 BOUND OF THE SCORE-MATCHING LOSS BY KL DIVERGENCE

**Lemma 3.** *Given the assumptions in Appendix F.1, if the score-matching loss satisfies $J(\theta, S_0) = J_{DSM}(\theta, \{r_t^2\}_{t=1}^T) \leq \epsilon$, the output distribution of the diffusion model $P_\theta$ satisfies*

$$D_{\mathrm{KL}}(P_0 \| P_\theta) \leq \epsilon + \mathbb{E}_{q(x_0)}[D_{\mathrm{KL}}(P_T \| p')] + C_1$$

*where $P_T$ is the forward process distribution at step $T$, $p'$ is the prior distribution (by design $P_T \approx p'$), and $C_1$ is a constant that does not depend on $\theta$ as below*

$$C_1 = \int_{t=0}^{T} g(t)^2 C_1'(x_0, x_t) \mathrm{d}t \tag{69}$$

*where $C_1'(x_0, x_t) = \mathbb{E}_{P_{0,t}(x_0,x_t)} \left[ \frac{1}{2} \|\nabla_{x_t} \log P_t(x_t)\|_2^2 - \frac{1}{2} \|\nabla_{x_t} \log P_{t|0}(x_t \mid x_0)\|_2^2 \right]$.*

To prove this lemma, we make the following assumptions for the SDE-based diffusion model in Section F.

**Assumptions**

1. $P_0(x)$ is a density function with continuous second-order derivatives and $\mathbb{E}_{x \sim P_0} \left[ \|x\|_2^2 \right] < \infty$.

2. The prior distribution $p'(x)$ is a density function with continuous second-order derivatives and $\mathbb{E}_{x \sim p'} \left[ \|x\|_2^2 \right] < \infty$.

3. $\forall t \in [0, T]$: $b(\cdot, t)$ is a function with continuous first order derivatives. $\exists C > 0, \forall x \in \mathcal{R}^d, t \in [0, T]$: $\|b(x, t)\|_2 \leq C(1 + \|x\|_2)$

4. $\exists C > 0, \forall x, y \in \mathcal{R}^d : \|b(x, t) - b(y, t)\|_2 \leq C\|x - y\|_2$.

5. $g$ is a continuous function and $\forall t \in [0, T], |r(t)| > 0$.

6. For any open bounded set $\mathcal{O}$, $\int_{t=0}^T \int_{\mathcal{O}} \|P_t(x)\|_2^2 + dr(t)^2 \|\nabla_x P_t(x)\|_2^2 \mathrm{d}x \mathrm{d}t$.

7. $\exists C > 0, \forall x \in \mathcal{R}^d, t \in [0, T] : \|\nabla_x \log P_t(x)\|_2 \leq C(1 + \|x\|_2)$.

8. $\exists C > 0, \forall x, y \in \mathcal{R}^d : \|\nabla_x \log P_t(x) - \nabla_x \log P_t(y)\|_2 \leq C(\|x - y\|_2)$.

9. $\exists C > 0, \forall x \in \mathcal{R}^d, t \in [0, T] : \|s_\theta(x, t)\|_2 \leq C(1 + \|x\|_2)$.

10. $\exists C > 0, \forall x, y \in \mathcal{R}^d : \|s_\theta(x, t) - s_\theta(y, t)\|_2 \leq C(\|x - y\|_2)$.

11. Novikov's condition: $\mathbb{E} \left[ \exp(\frac{1}{2} \int_{t=0}^T \|\nabla_x \log P_t(x) - s_\theta(x, t)\|_2^2 \mathrm{d}t) \right] < \infty$.

Given the assumptions, the following lemma is a re-statement of Theorem 1 in Song et al. (2021).

**Lemma 4.** *Let $P_0(x)$ be the underlining data distribution, $p'(x)$ be a known prior distribution, and $P_\theta$ be marginal distribution of $\hat{x}_\theta(0)$, the output of reverse-time SDE defined as below.*

$$\mathrm{d}\hat{x} = [b(\hat{x}, t) - r(t)^2 s_\theta(\hat{x}, t)]\mathrm{d}t + r(t)\mathrm{d}\bar{w}, \ \hat{x}_\theta(T) \sim p' \tag{70}$$

*With the assumptions in Section F.1, we have*

$$D_{\mathrm{KL}}(P_0 \| P_\theta) \leq J_{SM}(\theta, r(\cdot)^2) + D_{\mathrm{KL}}(P_T \| p') \tag{71}$$

*where $J_{SM}(\theta, r(\cdot)^2) = \frac{1}{2} \int_{t=0}^T \mathbb{E}_{p_t(x)} \left[ r(t) \|\nabla_x \log P_t(x) - s_\theta(x, t)\|_2^2 \right] \mathrm{d}t$.*

*Proof.* We denote the path measure of the forward outputs $\{x_t\}_{t \in [0,T]}$ as $p$ and the path measure of the backward outputs $\{\hat{x}_{\theta,t}\}_{t \in [0,T]}$ as $q$. By assumptions 1- 5, 9, 10, both $p$ and $q$ are uniquely given by the forward and backward SDEs, respectively. Consider a Markov kernel $M(\{z_t\}_{t \in [0,T]}, y) := \delta(z_0 = y)$ given any Markov chain $\{z_t\}_{t \in [0,T]}$. Since $x_0 \sim P_0$ and $\hat{x}_{\theta,0} \sim P_\theta$, we have

$$\int M(\{x_t\}_{t \in [0,T]}, x) \mathrm{d}p(\{x_t\}_{t \in [0,T]}) = P_0(x) \tag{72}$$

$$\int M(\{\hat{x}_{\theta,t}\}_{t\in[0,T]}, x)\mathrm{d}q(\hat{x}_{\theta,t}\}_{t\in[0,T]}) = P_\theta(x) \tag{73}$$

Here the Markov kernel $M$ essentially performs marginalization of path measures to obtain distributions at $t = 0$. We can use the data processing inequality with this Markov kernel to obtain

$$D_{\mathrm{KL}}(P_0\|P_\theta)$$

$$=D_{\mathrm{KL}}\left(\int M(\{x_t\}_{t\in[0,T]}, x)\mathrm{d}p(\{x_t\}_{t\in[0,T]})\|\int M(\{\hat{x}_{\theta,t}\}_{t\in[0,T]}, x)\mathrm{d}q(\hat{x}_{\theta,t}\}_{t\in[0,T]})\right) \tag{74}$$

$$\leq D_{\mathrm{KL}}(p, q)$$

Since $x_T \sim P_T$ and $\hat{x}_{\theta,T} \sim p'$. Leveraging the chain rule of KL divergence, we have

$$D_{\mathrm{KL}}(p, q) = \mathbb{E}_p\left[\log(\frac{p(x_{1:T}\mid x_T)P_T(x_T)}{q((\hat{x}_{\theta,1:T}\mid \hat{x}_{\theta,T})p'(\hat{x}_{\theta,T})})\right]$$

$$= D_{\mathrm{KL}}(P_T\|p') + \mathbb{E}_{z\sim P_T}\left[D_{\mathrm{KL}}(p(\cdot\mid x_T = z)\|q(\cdot\mid \hat{x}_{\theta,T} = z))\right] \tag{75}$$

Under assumptions 1,3-8, the SDE in Eqn. equation 66 has a corresponding reverse-time SDE as

$$\mathrm{d}x = [b(x, t) - r(t)^2 \nabla_x \log P_t(x)] + r(t)\mathrm{d}\bar{w} \tag{76}$$

Since Eqn. equation 76 is the time reversal of Eqn. equation 66, they share the same path measure $p$. Thus, $\mathbb{E}_{z\sim P_T}\left[D_{\mathrm{KL}}(p(\cdot\mid x_T = z)\|q(\cdot\mid \hat{x}_{\theta,T} = z))\right]$ can be viewed as the KL divergence between the path measures induced by the two SDEs in Eqn. equation 66 and Eqn. equation 70 with the same starting points $x_T = \hat{x}_{\theta,T} = z$.

The KL divergence between two SDEs with shared diffusion coefficients and starting points exists under assumptions 7,-11, and can be bounded by the Girsanov theorem

$$D_{\mathrm{KL}}(p(\cdot\mid x_T = z)\|q(\cdot\mid \hat{x}_{\theta,T} = z)) = \mathbb{E}_p\left[\log\frac{\mathrm{d}p}{\mathrm{d}q}\right]$$

$$=\mathbb{E}_p\left[\int_{t=0}^T r(t)(\nabla_x \log P_t(x) - s_\theta(x, t))\mathrm{d}\bar{w}_t + \frac{1}{2}\int_{t=0}^T r(t)^2\|\nabla_x \log P_t(x) - s_\theta(x, t)\|_2^2\mathrm{d}t\right]$$

$$=\mathbb{E}_p\left[\frac{1}{2}\int_{t=0}^T r(t)^2\|\nabla_x \log P_t(x) - s_\theta(x, t)\|_2^2\mathrm{d}t\right]$$

$$=\frac{1}{2}\int_{t=0}^T \mathbb{E}_{P_t(x)}\left[r(t)^2\|\nabla_x \log P_t(x) - s_\theta(x, t)\|_2^2\right]\mathrm{d}t = J_{SM}(\theta, r(\cdot)^2) \tag{77}$$

where the second equality holds by Girsanov Theorem II, and the third equality holds because $Y_s = \int_{t=0}^s r(t)(\nabla_x \log P_t(x) - s_\theta(x, t))\mathrm{d}\bar{w}_t$ is a continuous-time Martingale process ($\mathbb{E}[Y_s\mid Y_{\tau,\tau\leq s'}] = Y_{s'}, \forall s' \leq s$) and we have $\mathbb{E}[Y_s - Y_{s'}] = 0, \forall s' < s$. $\qquad\square$

*Proof.* The score matching loss $J(\theta, S_0)$ in equation 68 with $\iota(t) = g(t)^2$ is actually the denoising score matching loss $J_{DSM}(\theta, g(\cdot)^2)$, i.e.

$$J(\theta, S_0) = J_{DSM}(\theta, g(\cdot)^2) = \frac{1}{2}\int_0^T \mathbb{E}_{x_0\sim P_0}\mathbb{E}_{x_t\sim P_{t|0}}\left[g(t)^2\left\|\nabla_{x_t} \log P_{t|0}(x_t\mid x_0) - s_\theta(x_t, t)\right\|^2\right]\mathrm{d}t \tag{78}$$

which is used to compute the original score matching loss $J_{SM}(\theta, g(\cdot)^2)$ given a dataset $S_0$. The gap between $J_{DSM}(\theta, g(\cdot)^2)$ and $J_{SM}(\theta, g(\cdot)^2)$ is a constant $C_1$ that does not depend on $\theta$, which is shown as below.

The difference is expressed as

$$J_{SM}(\theta, g(\cdot)^2) - J_{DSM}(\theta, g(\cdot)^2)$$

$$=\frac{1}{2}\int_{t=0}^T \mathbb{E}_{P_{0,t}(x_0,x_t)}\left[g(t)^2\left(\|\nabla_{x_t} \log P_t(x_t) - s_\theta(x_t, t)\|_2^2 - \|\nabla_{x_t} \log P_{t|0}(x_t\mid x_0) - s_\theta(x_t, t)\|_2^2\right)\right]\mathrm{d}t$$

$$=\int_{t=0}^T g(t)^2\left(\underbrace{\mathbb{E}_{P_{0,t}(x_0,x_t)}\left[-\left\langle s_\theta(x_t, t), \nabla_{x_t} \log P_t(x_t) + \nabla_{x_t} \log P_{t|0}(x_t\mid x_0)\right\rangle\right]}_{(1)} + C_1'(x_0, x_t)\right)\mathrm{d}t, \tag{79}$$

where $P_{0,t}$ is the joint distribution of $x_0$ and $x_t$, and $C_1'(x_0, x_t) =$ $\mathbb{E}_{P_{0,t}(x_0,x_t)} \left[ \frac{1}{2} \|\nabla_{x_t} \log P_t(x_t)\|_2^2 - \frac{1}{2} \|\nabla_{x_t} \log P_{t|0}(x_t \mid x_0)\|_2^2 \right]$.

The first term (1) is zero because

$$
\begin{aligned}
\mathbb{E}_{P_{0,t}(x_0,x_t)} & \left[ \langle s_\theta(x_t, t), \nabla_{x_t} \log P_t(x_t) \rangle \right] = \mathbb{E}_{P_t(x_t)} \left[ \langle s_\theta(x_t, t), \nabla_{x_t} \log P_t(x_t) \rangle \right] \\
&= \int_{x_t} \left\langle s_\theta(x_t, t), \frac{1}{P_t(x_t)} \nabla_{x_t} P_t(x_t) \right\rangle P_t(x_t) \mathrm{d}x_t \\
&= \int_{x_t} \left\langle s_\theta(x_t, t), \nabla_{x_t} \int_{x_0} P_{t|0}(x_t \mid x_0) P_0(x_0) \mathrm{d}x_0 \right\rangle \mathrm{d}x_t \\
&= \int_{x_t} \left\langle s_\theta(x_t, t), \int_{x_0} P_{t|0}(x_t \mid x_0) P_0(x_0) \nabla_{x_t} \log(P_t(x_t \mid x_0)) \mathrm{d}x_0 \right\rangle \mathrm{d}x_t \\
&= \int_{x_0,x_t} P_{0,t}(x_0, x_t) \left\langle s_\theta(x_t, t), \nabla_{x_t} \log(P_{t|0}(x_t \mid x_0)) \right\rangle \mathrm{d}x_0 \mathrm{d}x_t \\
&= \mathbb{E}_{P_{0,t}(x_0,x_t)} \left[ \left\langle s_\theta(x_t, t), \nabla_{x_t} \log P_{t|0}(x_t \mid x_0) \right\rangle \right]
\end{aligned}
\tag{80}
$$

Thus, we can bound the gap between $J_{DSM}(\theta, g(\cdot)^2)$ and $J_{SM}(\theta, g(\cdot)^2)$ as

$$
C_1 = J_{SM}(\theta, g(\cdot)^2) - J_{DSM}(\theta, g(\cdot)^2) = \int_{t=0}^T g(t)^2 C_1'(x_0, x_t) \mathrm{d}t
\tag{81}
$$

where $C_1'(x_0, x_t) = \mathbb{E}_{P_{0,t}(x_0,x_t)} \left[ \frac{1}{2} \|\nabla_{x_t} \log P_t(x_t)\|_2^2 - \frac{1}{2} \|\nabla_{x_t} \log P_{t|0}(x_t \mid x_0)\|_2^2 \right]$.

Therefore, if $J(\theta, S_0) = J_{DSM}(\theta, g(\cdot)^2) \le \epsilon$, by Lemma 4, we have

$$
\begin{aligned}
D_{\mathrm{KL}}(P_0 \| P_\theta) &\le J_{SM}(\theta, g(\cdot)^2) + D_{\mathrm{KL}}(P_T \| p') \\
&= J_{DSM}(\theta, g(\cdot)^2) + D_{\mathrm{KL}}(P_T \| p') + C_1 \\
&\le \epsilon + D_{\mathrm{KL}}(P_T \| p') + C_1
\end{aligned}
\tag{82}
$$

which completes the proof. $\qquad\square$

## G   THE USE OF LARGE LANGUAGE MODELS (LLMS)

LLMs were used as general-purpose assistive tools to improve the clarity of English writing and to check grammar. No parts of the technical contributions, experimental design, or theoretical results were produced directly by LLMs. All mathematical derivations, proofs, algorithms, and final writing decisions were carried out by the authors. The use of LLMs was limited to supportive roles and did not constitute a substantive contribution to the research itself.

