# OpenReview forum: "Distributionally Robust Optimization via Generative Ambiguity Modeling"
_ICLR.cc/2026/Conference — ICLR 2026 Poster_

### Official Review · Reviewer_4i8h · 2025-10-14

**Soundness:** 2
**Presentation:** 2
**Contribution:** 2
**Rating:** 4
**Confidence:** 4

**Summary:**

This paper studies the DRO problems with generative model-based ambiguity sets. For the problems with new constraints, a new framework is proposed to solve the problems and numerical results are provided.

**Strengths:**

The paper provides both theoretical and numerical results for the proposed methods.

**Weaknesses:**

The writing can be improved. For example, the $s_\theta$ used in line 204 is not previously defined.  The definitions of $J_{VAE}, R_{VAE}$ are important and should be highlighted under equation (6). The smooth and Lipschitz assumptions used in the Theorems should be defined.

**Questions:**

While the presentation contains some minor issues, my main concern lies in the motivation and conceptual clarity of this paper.

1. The paper argues that for general DRO problems, in the case of $\varphi$-divergence DRO, the target distribution must be absolutely continuous with respect to the nominal distribution, and that Wasserstein-DRO formulations are computationally complex. The authors then propose an upper bound on the KL divergence in Lemma 1. However, I find this result puzzling: the left-hand side is expressed as a general function of $P_\theta$, while the right-hand side depends only on $\theta$. This mismatch suggests that the authors implicitly treat $P_\theta$  as a parameterized distribution determined by $\theta$. Consequently, the results in this paper seem to apply only to specific parameterized models, e.g.,  VAEs or diffusion models, rather than to general optimization problems in the DRO framework.

2. Even within the context of VAEs and diffusion models, I am not convinced of the necessity of this approach. Lemma 1 shows that the upper bound on the KL divergence is linear in $J$ and, given $P_0$,  the gap between KL and J is fixed. Therefore, the results produced by the proposed method may not differ substantially from those obtained through traditional KL-based methods, especially since the constraint functions are linear and the KL divergence is typically easier to compute.

---

> ### Author Response · Authors · 2025-11-22
>
> We thank the reviewer for the insightful feedback. Below, we provide detailed responses to the questions.
>
> **`The results in this paper seem to apply only to specific parameterized models, e.g., VAEs or diffusion models, rather than to general optimization problems in the DRO framework.`**
>
> Our framework is designed for  DRO problems with general optimization objectives. The parameterized generative models are used only to model the ambiguity set, whereas the DRO objective $f(w,x)$ in Eqn. (1) is fully general. Therefore, our framework is not restricted to specific loss functions.
>
> Rather than limiting the applicability of DRO, the introduction of parameterized generative models is intended to enhance robustness and generalization, as discussed in the last two paragraphs of the Introduction. Prior works [1,2,3] have already demonstrated that using generative models to represent adversarial distributions in ambiguity sets leads to improved performance compared to unparameterized distribution modeling approaches. Our method differs fundamentally from these works in that we redesign the ambiguity set directly in the parameterized space of generative models, instead of relying on traditional ambiguity sets such as those based on KL divergence or the Wasserstein metric.
>
> This new design avoids the limitations of traditional ambiguity sets (see the second paragraph of the Introduction) and results in a principled and directly tractable GAS-DRO algorithm (Algorithm 1), which is supported by both formal convergence guarantees (Theorems 1 and 2) and empirical superiority (Section 7).
>
> [1] Michel, P., Hashimoto, T., & Neubig, G. (2021). Modeling the second player in distributionally robust optimization. ICLR.
>
> [2] Michel, P., Hashimoto, T., & Neubig, G. (2022). Distributionally robust models with parametric likelihood ratios. ICLR.
>
> [3] Ren, A. Z., & Majumdar, A. (2022). Distributionally robust policy learning via adversarial environment generation. IEEE Robotics and Automation Letters, 7(2), 1379-1386.
>
> **`The results produced by the proposed method may not differ substantially from those obtained through traditional KL-based methods, especially since the constraint functions are linear and the KL divergence is typically easier to compute.`**
>
> We respectfully hold a different view from the reviewer’s comment. Please note that in Lemma 1, we show that the proposed reconstruction-loss-based ambiguity set implies the bound on the **reverse KL divergence** $D_{\mathrm{KL}}(P_0 \| \cdot)$ rather than the standard KL divergence $D_{\mathrm{KL}}(\cdot \| P_0)$ that is commonly adopted in existing KL-based DRO frameworks. While the standard KL divergence $D_{\mathrm{KL}}(\cdot \| P_0)$ leads to tractable DRO formulations, it requires all adversarial distributions to be absolutely continuous with respect to the nominal distribution $P_0$. As a result, the support of any adversarial distribution must be a subset of the support of the nominal data distribution, which has been shown to limit generalization performance under support-shift scenarios (a limitation documented in multiple prior studies [4,5,6]).
>
> In contrast, **the reverse KL divergence $D_{\mathrm{KL}}(P_0 \| P_{\theta})$ bounded in Lemma 1 does not require the adversarial distribution $P_{\theta}$ to share the same support with the nominal data distribution $P_0$**. This allows the generative model to identify more expressive and flexible worst-case distributions, thereby yielding improved robustness compared to traditional KL-divergence based methods.
>
> [4] Staib, M., & Jegelka, S. (2019). Distributionally robust optimization and generalization in kernel methods. NeurIPS.
>
> [5] Chen, R., & Paschalidis, I. C. (2020). Distributionally robust learning. Foundations and Trends® in Optimization, 4(1-2), 1-243.
>
> [6] Lu, M., Zhong, H., Zhang, T., & Blanchet, J. (2024). Distributionally robust reinforcement learning with interactive data collection: Fundamental hardness and near-optimal algorithms. NeurIPS.
>
> **`Clarification of Notations`**
>
> Thank you for the suggestion to clarify some notations regarding the generative models. The function $s_{\theta}(x_t, t)$ denotes a ML model parameterized by $\theta$ that predicts the additive noise used to construct the mean of $x_{t-1}$ from $x_t$ in the diffusion process. The term $J_{\mathrm{VAE}}$ represents the reconstruction loss, which encourages the learned model to regenerate data that is consistent with the original data distribution. The term $R_{\mathrm{VAE}}$ denotes the prior-matching loss, which measures how similar the learned variational distribution is to a prior belief held over latent variables. We have added these clarifications in the revised version in Page 5.
>
> **`Clarification of smooth and Lipschitz assumptions`**
>
> Thank you for the suggestion to clarify the assumptions in Theorem 1 and Theorem 2. We have formally described the assumptions in Assumption 1 and Assumption 2 in Section C.3 (Page 20).

---

> ### Author Response · Authors · 2025-11-26
>
> We sincerely thank the reviewer for their time and thoughtful feedback, and we would greatly appreciate their consideration of our response. We would like to reiterate that **our new ambiguity modeling is fundamentally different from traditional KL-based methods, as it formally guarantees a bound on the reverse KL divergence, thereby avoiding the support-matching limitations inherent in the tradition KL-based methods.** We would be more than happy to address any remaining concerns the reviewer may have.

---

### Official Review · Reviewer_WYi2 · 2025-10-27

**Soundness:** 3
**Presentation:** 3
**Contribution:** 4
**Rating:** 8
**Confidence:** 3

**Summary:**

The paper addresses the problem of distributionally robust optimization for enhancing robustness in statistical learning under out-of-distribution scenarios. The main idea is to construct a novel ambiguity set using likelihood-based generative models, such as diffusion models or VAEs, which allows for distributions that are consistent with the nominal distribution while enabling support shifts and tractable solutions. Key results include the GAS-DRO algorithm (Algorithm 1), which solves the minimax problem via dual learning and policy optimization. Theoretical contributions comprise Theorem 1, establishing convergence of the inner maximization to the optimal oracle with bounded KL divergence to the nominal distribution, and Theorem 2, proving stationary convergence of GAS-DRO under smoothness and Lipschitz assumptions. Empirical results on Electricity Maps datasets show superior OOD generalization compared to baselines like W-DRO, KL-DRO, and DRAGEN.

## Clarity

The paper is very well-written and clearly structured. The introduction provides excellent motivation by clearly outlining the limitations of existing DRO frameworks. The proposed method (GAS) is developed logically, building from the problem setup (Sec 3) to the generative model formulationand the algorithm.

## Originality / Novelty

This work is highly original. The central idea of defining a DRO ambiguity set using a generative model's reconstruction loss is, to my knowledge, new.

**Strengths:**

- The core idea of formulating the ambiguity set over the parameters of a generative model and, crucially, using the *reconstruction loss* $J(\theta, P_0) \le \epsilon$ as the constraint (Eq. 7) is a highly novel and elegant contribution. It reframes the problem from an intractable search over distributions to a tractable optimization over model parameters.

- The method provides a compelling solution to the fundamental tension between $\phi$-divergence DRO (no support shift) and Wasserstein-DRO (intractable/conservative). The use of generative models inherently allows for sampling beyond the nominal support space, while the constraint $J(\theta, P_0)$ ensures consistency with the nominal distribution.

- The paper proposes a concrete, end-to-end algorithm (Algorithm 1). The application of dual learning and policy optimization to solve the complex inner loop is a clever technical contribution that makes the framework practical.

- The authors provide non-trivial convergence guarantees for their proposed algorithm. Theorem 1 establishes the convergence of the inner-loop maximization oracle, and Theorem 2 proves stationary convergence for the full min-max procedure, lending strong theoretical weight to the method.

- The experiments, summarized in Table 1, show a clear and substantial performance improvement for GAS-DRO over all baselines. It achieves the lowest average error (0.0163) by a significant margin over the next-best baseline, DRAGEN (0.0230), and standard ML (0.0450). The ablation studies in Appendix D.2 are also thorough.

**Weaknesses:**

- The theoretical link (Lemma 1) between the proposed constraint $J(\theta, P_0) \le \epsilon$ and traditional divergence-based ambiguity sets is a one-way bound ($J \le \epsilon \implies D_{KL}(P_0 || P_\theta)$ is bounded). This ensures distributions in GAS are "consistent," but it's not clear if the set is sufficiently expressive. It's possible that a "worst-case" distribution $P^*$ that is close to $P_0$ (in a standard metric) might not be representable by any $P_\theta$ in the GAS, even with a powerful generative model. The paper relies on the "strong capability" of generative models, but this is not formally characterized.

- The analysis assumes bounded reconstruction loss and specific conditions on objectives (e.g., $\beta$-smooth, L-Lipschitz), which may not hold in all deep learning settings; this is a theoretical concern (Sec. 6, Thm. 2).

- The paper doesn't explore the connection between the adversary budget $\epsilon$ with regular ambiguity radius in DRO, like as $\mathcal{B}_\delta(P_0)$.

## Minor:

1. correct below typos:

  - “support **shit** testing cases”

  - “…restrict **the the** search of worst-case distributions…”

  - “The convergence of the Moreau **envelop** gradient norm…”


2. Repeated references like Staib & Jegelka (2017a)/(2017b) without disambiguation in prose.

3. "Ma et al. Ma et al. (2024)" — The citation "Ma et al." is repeated. Also "Wang et al. Wang et al. (2025)" — The citation "Wang et al." is repeated.

**Questions:**

Lemma 1 is doing a lot of work: it’s the bridge from small reconstruction loss” to bounded $D_{\mathrm{KL}}(P_0 | P_\theta)$.
But in the main text, the statement is fairly high-level:

    - It asserts $D_{\mathrm{KL}}(P_0 | P_\theta) \le J(\theta,P_0)+R(p',P_0)+C_1$, where $R(p',P_0)$ depends on the prior and $C_1$ depends on $-H(P_0)$.

    - It then argues that this allows support shift (since it’s the *reverse* KL, which only needs $P_0 \ll P_\theta)$.
However, several details are pushed to Appendix C.1 and are not visible in the main body we have:

- Are there assumptions on model capacity (i.e., is $P_\theta$ assumed expressive enough to approximate $P_0$ arbitrarily well)?
- Are there bounds on $R(p',P_0)$ in practice, especially for diffusion, where $p'$ is a fixed Gaussian prior?
- Is $C_1$ finite when $P_0$ is empirical with finite support (so $-H(P_0)$ is well-defined)?
  These questions matter because if $R(p',P_0)$ is large or uncontrolled, then “bounded reconstruction loss ⇒ bounded KL” may not actually constrain the adversary meaningfully. Right now, the story is intuitive but not fully airtight in the visible text.

---

> ### Author Response · Authors · 2025-11-22
>
> We thank the reviewer for the insightful feedback. Below, we provide detailed responses to the questions.
>
> **`It's not clear if the set is sufficiently expressive...The paper relies on the "strong capability" of generative models, but this is not formally characterized.`**
>
> We thank the reviewer for this insightful question.
> The expressive capacity of the generative model is important to the performance of GAS-DRO. To ensure robustness, the generative model must approximate the possible testing distributions with sufficiently small representation error. In particular, if the set of possible testing distributions contains the nominal distribution $P_0$ (which is typically the case), the generative model should be able to represent $P_0$ accurately.
>
> We analyze the effect of the expressivity of generative models on GAS-DRO in the added Section C.4 (Page 22). We introduce the notion of $\Gamma$-expressivity in **Definition 1** (Page 22), which quantifies the expressive capacity of the generative model using the Wasserstein distance. Based on this notion, we establish **Corollary 1** (Page 22), which shows the generative-model expressivity quantified by $\Gamma$ affects the inner maximization performance by introducing an additional error $L_x\cdot\Gamma$ where $L_x$ is the Lipschitz constant of the objective $f$ with respect to $x$.
>
> When the expressivity parameter $\Gamma$ is smaller-(more expressive generative model), GAS-DRO can identify adversarial distributions that are more consistent with the true worst-case testing distribution, as reflected by a smaller objective gap. In this case, optimizing the objective parameter $w$ with respect to this learned adversarial distribution enables GAS-DRO to achieve better robustness across the true testing distributions.
>
> Furthermore, when $\Gamma = 0$, the generative model can approximate the possible testing distributions arbitrarily well. In this ideal setting, the inner maximization error converges to zero as the number of inner iterations $K$ increases.
>
> In addition, Corollary 1 shows that the reverse KL divergence $D_{\mathrm{KL}}(P_0 \mid P_{\theta})$ remains bounded as long as there exists a feasible $\theta$ to satisfy the constraint on the reconstruction loss.
>
> Beyond the theoretical analysis, we empirically investigate how the expressive capacity of generative models affects the performance of GAS-DRO in Section D.4.5 (Table 5, Page 28). We evaluate GAS-DRO using generative models with different expressive capacities, and the results are summarized in the table below.
>
> | METHOD            | AVERAGE | MAXIMUM |
> |-------------------|---------|-----------|
> | GAS-DRO (DFU-full) | 0.0163  | 0.0509    |
> | GAS-DRO (DFU-3/4)  | 0.0171  | 0.0544    |
> | GAS-DRO (DFU-1/2)  | 0.0178  | 0.0608    |
>
> GAS-DRO (DFU-full) denotes the diffusion-based implementation using a full U-Net backbone. The full U-Net architecture consists of four downsampling layers with channel sizes of 128, 256, 256, and 256, respectively. DFU-3/4 and DFU-1/2 are lightweight diffusion variants that use U-Nets with $3/4$ and $1/2$ of the channel capacity at each layer, respectively. These models are ordered by expressive capacity as follows: DFU-full > DFU-3/4 > DFU-1/2.
> We observe that more expressive generative models consistently achieve stronger robustness under GAS-DRO, as they can more accurately approximate adversarial distributions and construct expressive uncertainty sets.

---

> ### Author Response · Authors · 2025-11-22
>
> **`The assumptions of bounded reconstruction loss, $\beta-$smoothness, and $L-$Lipschitz may not hold in all deep learning settings.`**
>
> We agree that the assumptions of bounded reconstruction loss, $\beta$-smoothness, and $L$-Lipschitz continuity may not hold across all deep learning settings. However, these assumptions apply to a broad class of practically relevant optimization or control problems [1,2]. Even in deep learning, smooth and continuous neural networks have been shown to be sufficiently expressive to model complex relationships. Many modern neural architectures are explicitly designed to preserve Lipschitz continuity properties for better generalization performance [3,4,5].
>
> More importantly, these assumptions are standard in theoretical optimization analysis and are widely adopted to obtain meaningful theoretical insights. For instance, [6] assume Novikov’s condition, which directly constrains the score-matching reconstruction loss of diffusion models to be finite. Moreover, $L$-Lipschitz continuity is a common assumption in the DRO literature (e.g., [7,8]).
> Furthermore, the considered DRO formulation is a non-convex and non-concave min-max optimization problem. These problems are believed to have the intrinsic hardness even under smoothness assumptions [9], so they are typically studied under smoothness assumptions [10,11].
>
>
> [1] Zhang, L., Jiang, W., Lu, S., & Yang, T. (2021). Revisiting smoothed online learning. NeurIPS.
>
> [2] Shi, G., Lin, Y., Chung, S. J., Yue, Y., & Wierman, A. (2020). Online optimization with memory and competitive control. NeurIPS.
>
> [3] Yoshida, Y., & Miyato, T. (2017). Spectral norm regularization for improving the generalizability of deep learning. arXiv preprint arXiv:1705.10941.
>
> [4] Gogianu, F., Berariu, T., Rosca, M. C., Clopath, C., Busoniu, L., & Pascanu, R. (2021, July). Spectral normalisation for deep reinforcement learning: an optimisation perspective. ICML.
>
> [5] Liu, H. T. D., Williams, F., Jacobson, A., Fidler, S., & Litany, O. (2022, July). Learning smooth neural functions via lipschitz regularization. In ACM SIGGRAPH 2022 Conference Proceedings (pp. 1-13).
>
> [6] Song, Y., Durkan, C., Murray, I., & Ermon, S. (2021). Maximum likelihood training of score-based diffusion models. Advances in neural information processing systems, 34, 1415-1428.
>
> [7] Mohajerin Esfahani, P., & Kuhn, D. (2018). Data-driven distributionally robust optimization using the Wasserstein metric: Performance guarantees and tractable reformulations. Mathematical Programming, 171(1), 115-166.
>
> [8] Chen, R., & Paschalidis, I. C. (2020). Distributionally robust learning. Foundations and Trends® in Optimization, 4(1-2), 1-243.
>
> [9] Daskalakis, C., Skoulakis, S., and Zampetakis, M.(2021). The complexity of constrained min-max optimization. STOC’21.
>
> [10] Diakonikolas, J., Daskalakis, C., & Jordan, M. I. (2021, March). Efficient methods for structured nonconvex-nonconcave min-max optimization. AISTATS.
>
> [11] Jin, C., Netrapalli, P., & Jordan, M. (2020, November). What is local optimality in nonconvex-nonconcave minimax optimization?. ICML.
>
>
> **`The connection between the adversary budget $\epsilon$ with regular ambiguity radius in DRO`**
>
> Lemma 1 establishes a formal relationship between the adversary budget and the ambiguity radius defined by the reverse KL divergence. Specifically, if the reconstruction loss is bounded by the adversarial budget $\epsilon$, then the reverse KL divergence  $D_{\mathrm{KL}}(P_0 \| P_{\theta})$ is also bounded by the sum of the adversarial budget $\epsilon$, the prior-matching loss $R(p', P_0)$, and a constant $C_1$ that is independent of the generative model parameters.
>
> Unlike the standard forward KL divergence, the reverse KL divergence imposes no restriction on the support of the adversarial distribution, while still effectively measuring the discrepancy between distributions. Therefore, the derived reverse KL divergence bound shows that the adversary budget controls the discrepancy between the learned adversarial distribution and the nominal distribution.
>
> We note that establishing similar guarantees under other distributional discrepancy measures, such as the Wasserstein distance, remains an interesting direction for future work.

---

> ### Author Response · Authors · 2025-11-22
>
> **`Are there assumptions on model capacity?`**
>
> We appreciate the reviewer’s question. Lemma 1 does not rely on the assumptions about model capacity, although a less expressive generative model may incur a larger reconstruction loss in the inverse KL divergence bound. Theorem 1 and Theorem 2 focus on analyzing the convergence performance of GAS-DRO in the parameterized ambiguity-set space, and do not require assumptions on the capacity of the underlying generative model.
>
> That said, the expressive capacity of the generative model is indeed important for the robustness performance of GAS-DRO. We formally analyze this effect in Corollary 1 in Section C.4 (Page 22), which characterizes how model expressivity influences the inner maximization error. In addition, we empirically demonstrate this effect in experiments in Section D.4.5 (Table 5, Page 28), where more expressive generative models lead to improved robustness.
>
> **`Are there bounds on $R(p',P_0)$ in practice, especially for diffusion, where $p'$ is a fixed Gaussian prior?`**
>
> The prior-matching loss depends on the design of the underlying generative model.
> In diffusion models, the prior distribution $p'(x_T)$ is typically chosen as a standard Gaussian, and the prior-matching loss depends on the design of the forward process.
> For variational diffusion models such as DDPM  (Ho, J. et al. 2020), the forward process defines
> $q(x_t \mid x_0) = \mathcal{N}\left(x_t; \sqrt{\alpha_t}x_0,\,(1-\alpha_t)\mathbf{I}\right)$.
> By choosing a decreasing noise schedule $\alpha_t$, the marginal distribution $q(x_T \mid x_0)$ converges to a standard Gaussian as $T \to \infty$. Consequently, the prior-matching loss
> $R(p',P_0) = \mathbb{E}\left[ D_{\mathrm{KL}}\big(q(x_T \mid x_0)\,\|\,p'(x_T)\big) \right]$
> approaches zero as the number of forward steps increases.
>
> For SDE-based diffusion processes, it can be proved that the forward process converges to a standard Gaussian distribution as the diffusion time tends to infinity (Song et al. (2021)). Therefore, the corresponding prior-matching loss in Lemma 3 (Page 32) also converges to zero in this limit.
>
> In VAEs, the prior-matching loss measures how well the encoder approximates the assumed prior distribution over latent variables. The prior is typically chosen to be a standard Gaussian distribution, and the prior-matching loss depends on how accurately the encoder learns to match this prior.
>
> **`Is $C_1$ finite when $P_0$ is empirical with finite support (so $-H(P_0)$ is well-defined)?`**
>
> The constant $C_1$ depends on the design of the generative model.
>
> For variational diffusion models and VAEs, $C_1$ depends on $-H(P_0)$, as shown in the proof of Lemma 1 in Section C.1 (Page 16). If $P_0$ is treated as a discrete distribution on the training dataset, its entropy is naturally bounded by $\log(|\mathcal{S}_0|)$, where $|\mathcal{S}_0|$ denotes the number of samples in the dataset. Beyond that, $P_0$ is also commonly viewed as the distribution from which the empirical samples are drawn, and its entropy is typically finite in practice.
>
> For score-matching-based diffusion models with SDE formulations (Section F), the constant $C_1$ is defined in Eqn.(69) (Lemma 3, Page 32) (based on the theoretical results in Song et al. (2021)). In this setting, $C_1$ depends on the gradients of the log-likelihood of the forward transition distributions. The finiteness of $C_1$ therefore depends on the properties of the chosen forward SDE process.
>
>
>
> **`Typos and repeated references`**
>
> We thank the reviewer for the careful reading of our paper and pointing out the typos and repeated reference issues. We have revised them in our revision.

---

> ### Author Response · Authors · 2025-11-26
>
> We sincerely thank the reviewer for their time and thoughtful feedback, and we would greatly appreciate their consideration of our response. In response to your insightful comments, we **have formally characterized the effects of the expressive capacity of generative models, justified the assumptions for our analysis, and discussed the bounds of each term in Lemma 1 across different generative models**. We would be more than happy to address any remaining concerns the reviewer may have.

---

> > ### Comment · Reviewer_WYi2 · 2025-11-27
> >
> > I appreciate the clear and thorough answers. The authors have adequately resolved the issues I pointed out, and I remain confident in the quality of the work. I will keep my initial positive evaluation.

---

> > > ### Author Response · Authors · 2025-11-27
> > >
> > > We sincerely thank the reviewer for the encouraging and constructive feedback. We are pleased that our responses have addressed the issues you pointed out and are happy to answer any further questions. We greatly appreciate your confidence in the quality of our work.

---

### Official Review · Reviewer_Focz · 2025-10-31

**Soundness:** 3
**Presentation:** 3
**Contribution:** 2
**Rating:** 6
**Confidence:** 4

**Summary:**

This paper proposes a novel approach to Distributionally Robust Optimization (DRO) that leverages generative models to define the ambiguity set of distributions. Instead of using a divergence or Wasserstein ball, the ambiguity set is parameterized by a generative model (e.g. a diffusion model or VAE) which can produce adversarial distributions beyond the support of the nominal data while still remaining similar to it. The authors introduce an algorithm called GAS-DRO (Generative Ambiguity Set DRO) that alternates between an inner loop (updating the generative model to find a worst-case distribution) and an outer loop (updating the primary model/decision variable on data sampled from that worst-case distribution). They enforce a constraint (via a dual Lagrange multiplier) to keep the generative distribution within a certain “distance” (ambiguity budget *ε*) of the nominal distribution. The paper provides theoretical guarantees – proving that the inner maximization converges to the optimal worst-case distribution and that the overall iterative procedure converges to a stationary point. Empirically, GAS-DRO is implemented with a diffusion model as the adversary and tested on a time-series forecasting task under distribution shift (electricity grid emissions data). It achieves significant improvements in out-of-distribution (OOD) generalization, outperforming several baselines like standard DRO with KL or Wasserstein ambiguity sets and a recent generative DRO method (DRAGEN). In summary, the paper’s main contributions are the introduction of generative-model-based ambiguity sets for DRO, a corresponding optimization algorithm with convergence analysis, and demonstration of improved robustness on real-world data.

**Strengths:**

1. The paper introduces a generative-model-based ambiguity set for DRO, which allows considering distributions outside the original support while maintaining similarity to the nominal distribution. The approach does not require too much prior condition and addresses the limitations of KL-divergence ambiguity (no support shift) and improves over Wasserstein ambiguity by providing a tractable, parameterized search space.
2. Empirical OOD Performance Gains: GAS-DRO demonstrates state-of-the-art OOD generalization performance on the Electricity Maps time-series datasets.
3. The writing is straightforward and easy to understand.

**Weaknesses:**

1. This paper investigates DRO, which is widespread considered as an approach for discriminative tasks. The method in the paper instead uses a generative method to address the classification problem but determining ambiguity set itself needs the sampling operation. Therefore, I wonder how much extra compuational cost this method has introduced compared to normal plug-and-play baselines? For instance, how much longer (in terms of training time or iterations) does GAS-DRO take compared to standard DRO or ERM on the same task?
2. How sensitive is GAS-DRO to the choice and quality of the generative model? For example, if one used a smaller diffusion model or a different type of generator (like a Variational Autoencoder, as mentioned, or even a GAN), would the method still perform as well? Any insight into how approximation errors in the generative model impact the robustness of the learned model would be valuable.
3. The empirical evaluation, while thorough for the chosen task, is focused on a single domain (renewable energy time-series forecasting on Electricity Maps data). It would strengthen the paper to see results in other domains to confirm that GAS-DRO’s benefits hold broadly. The current evidence of superior performance is strong for the scenario presented, but the approach's generality would be more convincing with a wider range of experiments.
4. Maybe adding more discussions on the use of generative modeling in common classification or OOD problems will be better. For example, a diffusion classifier [1] is adopted for image classification. Although authors have presented related work on IID settings, the description of OOD is somewhat lacking [2, 3].

[1] Li, Alexander C., et al. "Your diffusion model is secretly a zero-shot classifier." *Proceedings of the IEEE/CVF International Conference on Computer Vision*. 2023.
[2] Tong, Yunze, et al. "Latent Score-Based Reweighting for Robust Classification on Imbalanced Tabular Data." *Forty-second International Conference on Machine Learning*.
[3] Zhang, Hengrui, et al. "Mixed-type tabular data synthesis with score-based diffusion in latent space." *arXiv preprint arXiv:2310.09656* (2023).

**Questions:**

Please see Weaknesses.

---

> ### Author Response · Authors · 2025-11-22
>
> We thank the reviewer for the insightful feedback. Below, we provide detailed responses to the questions.
>
> **`How much extra computational cost this method has introduced compared to normal plug-and-play baselines? For instance, how much longer (in terms of training time or iterations) does GAS-DRO take compared to standard DRO or ERM on the same task?`**
>
> We conducted a detailed measurement of wall-clock time, CPU memory, and GPU memory consumption across all baselines under identical hardware settings in Section D.4.6 (Table 6 in Page 28). The comparisons with the plug-and-play baselines are summarized in the following table:
>
> | METHOD        | TIME(s) | MEMORY (MB)  | GPU MEMORY (MB) |
> |---------------|---------|--------------|-----------------|
> | GAS-DRO (DFU) | 1897    | 2274         | 12239           |
> | GAS-DRO (VAE) | 494     | 2449         | 5849            |
> | KL-DRO        | 149     | 2314         | 417             |
> | W-DRO         | 2896    | 2304         | 417             |
> | ML            | 143     | 1352         | 33              |
>
> Due to the closed-form solution for KL-DRO, it has the least training overhead among DRO baselines.   Given the difficulty in solving W-DRO, it has the highest training time, but its GPU memory footprint is low since it does not employ generative models.
> GAS-DRO (DFU) represents GAS-DRO based on a diffusion model, while GAS-DRO (VAE) represents GAS-DRO based on a VAE model.   Due to the long denoising process, GAS-DRO (DFU) has a higher training overhead than ML (ERM). However, GAS-DRO (VAE) is efficient, substantially faster than its Diffusion counterpart.  Overall, the additional computation cost of GAS-DRO is well-justified by the significant performance improvements achieved by GAS-DRO.
>
> **`How sensitive is GAS-DRO to the choice and quality of the generative model?`**
>
> We conducted an additional set of comparison experiments to study the effects of the approximation ability of generative models in Section D.4.5 (Table 5, Page 28).  The results are shown in the table below.
>
> | METHOD            | AVERAGE | MAXIMUM |
> |-------------------|---------|-----------|
> | GAS-DRO (DFU-full) | 0.0163  | 0.0509    |
> | GAS-DRO (DFU-3/4)  | 0.0171  | 0.0544    |
> | GAS-DRO (DFU-1/2)  | 0.0178  | 0.0608    |
> | GAS-DRO (VAE)     | 0.0168  | 0.0463    |
>
> GAS-DRO (DFU) uses DDPM [1] with U-Net as the backbone. GAS-DRO (DFU-full) denotes the diffusion-based implementation using a full U-Net backbone. The full U-Net architecture consists of four downsampling layers with channel sizes of 128, 256, 256, and 256, respectively. DFU-3/4 and DFU-1/2 are lightweight diffusion variants that use U-Nets with 3/4 and 1/2 of the channel capacity at each layer, respectively. GAS-DRO (VAE) refers to the VAE-based implementation where the encoder and decoder both have 2 convolutional layers with 16 channels followed by 2 MLP layers.
>
> From the empirical results, we observe that larger U-Net backbones consistently lead to better DRO performance. This suggests that more expressive generative models can better approximate adversarial distributions, thereby enhancing the robustness of GAS-DRO. In addition, the VAE-based variant achieves intermediate performance between GAS-DRO (DFU-full) and GAS-DRO (DFU-3/4) in average, demonstrating that GAS-DRO is not tied specifically to diffusion models and can be effectively implemented based on other generative models.
>
>
> [1] Ho, J., Jain, A., & Abbeel, P. (2020). Denoising diffusion probabilistic models. NeurIPS.

---

> ### Author Response · Authors · 2025-11-22
>
> **`It would strengthen the paper to see results in other domains to confirm that GAS-DRO’s benefits hold broadly.`**
>
> We evaluate GAS-DRO on an image classification task on the MNIST dataset in Section E. Specifically, we consider 10-class digit classification problem on the MNIST and USPS datasets. The models are trained on MNIST dataset and are evaluated on OOD testing datasets. USPS dataset [1] is a natural OOD benchmark. Additionally, we synthesize OOD testing datasets by adding various corruptions, including Gaussian noise, Perlin noise, and Cutout noise, into MNIST and the USPS dataset [1].
> We compare GAS-DRO with DRAGEN, KL-DRO, Wasserstein-based DRO (W-DRO), data-augmented ML(DML) and standard ERM (ML). Generative models used for GAS-DRO and DRAGEN share the same VAE architecture. Some results are summarized in the following table and more details can be found in Table 7 (Page 31).
>
> | Performance    | GAS-DRO | DRAGEN | KL-DRO | W-DRO  | DML    | ML     |
> |-----------------|---------|--------|--------|--------|--------|--------|
> | AVERAGE         | 71.52%  | 68.25% | 65.28% | 67.05% | 66.16% | 62.03% |
> | WORST           | 50.58%  | 46.84% | 45.38% | 46.11% | 42.08% | 42.38% |
>
> As shown in the table, GAS-DRO achieves higher classification accuracy than data augmentation method (DML), traditional DRO methods and the generative DRO baseline DRAGEN.  In the worst-case OOD settings, GAS-DRO also maintains the strongest performance. Although KL-DRO, W-DRO, and DRAGEN may perform well on certain specific OOD testing datasets, their overall accuracy remains inferior to GAS-DRO.
>
> The superior performance of GAS-DRO can be attributed to several key factors. First, GAS-DRO is not constrained by KL-divergence–based ambiguity sets. This allows GAS-DRO to fully exploit the expressive capacity of generative models to construct diverse adversarial distributions with support shifts, leading to improved robustness compared to KL-DRO.
> Second, the GAS-DRO formulation is defined directly in the parameterized space of generative models, thereby avoiding the need to relax Wasserstein constraints as required in W-DRO or DRAGEN. This results in more stable convergence behavior and stronger generalization performance.
>
> [1] Hull, J. J. (2002). A database for handwritten text recognition research. IEEE Transactions on pattern analysis and machine intelligence, 16(5), 550-554.
>
> **`Maybe adding more discussions on the use of generative modeling in common classification or OOD problems will be better.`**
>
> We thank the reviewer for the suggestion. We summarized some generative methods for OOD problems in related works of our original submission. We have added more discussions on  generative methods for broader classification and OOD problems including the mentioned three papers.

---

> ### Author Response · Authors · 2025-11-26
>
> We sincerely thank the reviewer for their time and thoughtful feedback, and we would greatly appreciate their consideration of our response. In response to your insightful comments, we have added **detailed comparisons and analysis of computational complexity, evaluations across different generative model capacities, and additional experiments on a vision task**. We would be more than happy to address any remaining concerns the reviewer may have.

---

### Official Review · Reviewer_u62m · 2025-11-01

**Soundness:** 3
**Presentation:** 2
**Contribution:** 2
**Rating:** 2
**Confidence:** 3

**Summary:**

This paper proposes GAS-DRO. It uses diffusion models and VAEs to build ambiguity sets for DRO. The method constrains the reconstruction loss of generative models instead of using KL divergence or Wasserstein distance. The authors provide convergence guarantees and test on time series forecasting tasks.

**Strengths:**

1. The paper identifies real problems with existing DRO methods. φ-divergence restricts support. Wasserstein distance is hard to optimize.

2. Theorems 1 and 2 provide convergence guarantees. Lemma 1 connects reconstruction loss to KL divergence.

3. GAS-DRO achieves 63.7% improvement over baseline ML. It outperforms existing DRO methods by significant margins.

**Weaknesses:**

1. Paper claims "first to model ambiguity sets in parameterized space of likelihood-based generative models" (page 2). This is incorrect. Michel et al. (2021) "Modeling the Second Player in Distributionally Robust Optimization" (ICLR 2021) already proposed Parametric DRO (P-DRO) that: (a) uses neural generative models q_ψ for adversary, (b) parameterizes uncertainty set with model weights ψ, (c) uses likelihood-based Transformers (evaluates log q_ψ(x,y) in Equation 8), (d) solves same min-max game structure, (e) tests on NLP tasks. The paper completely omits this citation. The distinction between "implicit" (GAN) and "explicit" (diffusion) generative models is insufficient—Michel et al. already used likelihood-based models. After removing false claims, only incremental contributions remain: using diffusion instead of Transformers, reconstruction loss instead of KL constraint, and time series application. No direct comparison with P-DRO is provided.

2. Limited experimental scope. Only one domain (time series forecasting). Michel et al. tested sentiment classification and toxicity detection. Need broader evaluation across vision or NLP tasks.

3. No computational cost analysis. Diffusion models are expensive. No runtime, memory, or scalability comparisons provided.

**Questions:**

See the weakness

---

> ### Author Response · Authors · 2025-11-22
>
> We thank the reviewer for the insightful feedback. Below, we provide detailed responses to the questions.
>
> **`Key contribution compared with Michel et al. (2021)`**
>
> We thank the reviewer for bringing Michel et al. (2021) to our attention.
> Michel et al. (2021) employ generative models to represent the adversarial distributions within a traditional KL-divergence–based ambiguity set. However, our contribution is fundamentally different: we design the ambiguity set itself which directly constrains the parameters of generative models, **without relying on any traditional ambiguity sets**.
> This design directly addresses several intrinsic limitations of traditional ambiguity sets (2nd paragraph of Introduction). Concretely, compared to P-DRO in Michel et al. (2021), our framework GAS-DRO offers the following advantages:
> +   **GAS-DRO has no support mismatch issue.**  In Michel et al. (2021), the ambiguity set is still based on the KL divergence, which requires all adversarial distributions to be absolutely continuous with respect to the nominal data distribution. Thus, the support of the adversarial distribution must be a subset of the nominal distribution support, which restricts the ability to model adversarial distributions with support shifts and ultimately weakens robustness (the support mismatch issue mentioned in many existing studies [1,2]).
> In contrast, GAS-DRO defines the ambiguity set through the constraints on the reconstruction-loss of a generative model. This reconstruction-loss-based ambiguity set is proved in Lemma 1 to **constrain the reverse KL divergence $D_{\mathrm{KL}}(P_0\|P_{\theta})$**. Unlike the KL divergence $D_{\mathrm{KL}}(\cdot \|P_0)$ used in Michel et al. (2021), the reverse KL divergence does not impose any requirement on the support of the adversarial distributions. This allows generative models to provide more expressive and flexible adversary representations, thereby leading to enhanced robustness.
> + **GAS-DRO is both directly tractable and theoretically explainable.**
> The KL-divergence–based ambiguity set in Michel et al. (2021) leads to a challenging optimization that requires several approximations (Eqns. (8)–(9)) to obtain a tractable formulation. As mentioned by Michel et al. (2021), Eqn. (8) relies on a crude approximation that replaces the KL divergence with its reverse KL divergence. The unexplainable approximations may create performance degradation of DRO and making it questionable when extending the method to broader applications and generative models, although it empirically works well for the NLP task with transformers.
> In contrast, GAS-DRO uses a reconstruction-loss–based ambiguity set that is **directly tractable without further approximations**. Moreover, this design provides **principled guarantees on optimization stability** while **explicitly constraining the reverse KL divergence between the adversarial and nominal distributions** (Theorems 1 and 2). As a result, GAS-DRO is both theoretically sound and computationally tractable.
>
> Beyond the new ambiguity modeling, our paper also introduces the following contributions and distinctions:
> + **A dual-learning method for solving GAS-DRO**.
> We propose a principled dual-learning approach that directly solves inner maximization of GAS-DRO (Algorithm 1). As established in Theorem 1, this method identifies a near-optimal worst-case distribution while ensuring its proximity to the nominal distribution through a reverse KL divergence constraint.
> + **Theoretical convergence guarantees.**
> Building on the proposed generative ambiguity modeling, we formally prove the convergence of GAS-DRO in Theorem 2, formally demonstrating the stability of GAS-DRO despite the challenges posed by the non-convex, non-concave min–max optimization.
> + **Implementation frameworks for diffusion models and VAEs.**
> We derive training objectives based on policy optimization techniques for diffusion models and VAEs. The empirical results demonstrate the advantages of GAS-DRO across both regression (Section D) and classification (Section E) tasks.
>
> **We carefully compared our work with Michel et al. (2021) and its follow-up work Michel et al. (2022) in Introduction (Section 1) and Related Work (Section 2)**. Together with Ren \& Majumdar (2022) in original Related Work, these methods are categorized as approaches that utilize generative models to represent adversarial distributions within traditional ambiguity modeling frameworks.
>
> In addition, we conducted **experimental comparisons with Michel et al. (2021)** on the time-series prediction task in Section 7, with further details provided in Section E.
>
> [1] Chen, R., & Paschalidis, I. C. (2020). Distributionally robust learning. Foundations and Trends® in Optimization, 4(1-2), 1-243.
>
> [2] Lu, M., Zhong, H., Zhang, T., & Blanchet, J. (2024). Distributionally robust reinforcement learning with interactive data collection: Fundamental hardness and near-optimal algorithms. NeurIPS.

---

> ### Author Response · Authors · 2025-11-22
>
> **`Explanation to the claim "first to model ambiguity sets in parameterized space of likelihood-based generative models".`**
>
> By this claim, we intend to highlight the novelty of **designing ambiguity sets that directly constrain the parameters of generative models, without relying on traditional distributional discrepancy measures** such as the KL divergence or the Wasserstein metric. We acknowledge, however, that this statement may be interpreted differently by readers from various research communities. To avoid potential ambiguity, we have added a clarifying footnote alongside the claim to explicitly disambiguate its intended meaning.
>
> **`The distinction between "implicit" (GAN) and "explicit" (diffusion) generative models is insufficient`**
>
>
> Our framework applies to generative models that satisfy the inverse KL divergence constraint established in Lemma 1. Typically, these generative models exactly or approximately optimize the likelihood. Since VAEs and diffusion models approximately maximize the likelihood, we derive inverse KL divergence constraints for these models (Lemma 1, restated for VAEs in Section B.1 on Page 15) and develop principled GAS-DRO algorithms based on them (Sections A and B).
>
> Autoregressive models and normalizing flows optimize the exact likelihood and therefore are natural candidates for this framework. Deriving concrete principled GAS-DRO implementation framework for these models is left as an interesting direction for future work. To our knowledge, GANs are implicit generative models that do not explicitly optimize likelihood and do not necessarily satisfy the conditions of Lemma 1. As a result, we cannot currently provide a principled GAS-DRO design based on GANs.
>
> Nevertheless, if future generative models can be shown to satisfy the inverse KL divergence constraint in Lemma 1, our framework can be naturally extended to construct new GAS-DRO variants with corresponding theoretical guarantees.
>
> **`Empirical comparison with P-DRO in Michel et al. (2021)`**
>
> We have added an empirical comparison with P-DRO on the time-series prediction task in Table 1 (Section 7, Page 9). A summary of the baseline results is shown below.
>
> | Performance | GAS-DRO | P-DRO |KL-DRO |W-DRO |ML |
> |---|---|---|---|---|---|
> |AVERAGE | 0.0163 | 0.0259 |0.0288 |0.0342 | 0.0450|
> |MAXIMUM | 0.0509 | 0.0820 |0.0831 |0.0879 | 0.0946|
>
> We implement P-DRO using a diffusion model as the generative component by replacing the probability model $q_{\varphi}$ in Objective (8) of Michel et al. (2021) with a diffusion-based probability model. The diffusion architecture is identical to the one used in GAS-DRO. All hyperparameters for the baselines are selected using a validation dataset.
>
> The results show that P-DRO outperforms traditional DRO methods based on KL divergence and Wasserstein distance, demonstrating improved training stability and the ability to construct adversarial distributions by diffusion models. However, GAS-DRO consistently outperforms P-DRO in both average-case and worst-case settings across a range of OOD testing datasets. One plausible explanation is that GAS-DRO can identify adversarial distributions whose support is not restricted by KL-divergence to coincide with that of the nominal distribution. In addition, the crude approximations used in P-DRO may affect the stability and accuracy of the inner maximization, leading to inferior robustness performance.

---

> ### Author Response · Authors · 2025-11-22
>
> **`Broader evaluation on a vision task`**
>
> We evaluate GAS-DRO and baseline methods on an image classification task using the MNIST and USPS benchmark. Experimental details are provided in Section E. Some results are summarized in the table below and more details can be found in Table 7 (Page 31).
>
> | Performance    | GAS-DRO | DRAGEN | KL-DRO | W-DRO  | DML    | ML     |
> |-----------------|---------|--------|--------|--------|--------|--------|
> | AVERAGE         | 71.52%  | 68.25% | 65.28% | 67.05% | 66.16% | 62.03% |
> | WORST           | 50.58%  | 46.84% | 45.38% | 46.11% | 42.08% | 42.38% |
>
> As shown in the table, GAS-DRO achieves higher average classification accuracy than the traditional DRO methods (KL-DRO and W-DRO), the data-augmentation baseline (DML), and the generative DRO baseline DRAGEN.  In the worst-case testing setting, GAS-DRO also maintains the strongest performance.  More details of the results can be found in Table 7 (Page 31).
> The superior performance of GAS-DRO is attributed to its reconstruction-loss-based ambiguity modeling, which is directly tractable without relaxation and can fully exploit the expressive capacity of generative models, resulting in more stable optimization and improved generalization under distribution shifts.
>
> **`Computation cost analysis and runtime, memory comparisons`**
>
> Algorithm 1 (GAS-DRO) is a min-max optimization procedure with two nested loops. The inner maximization loop performs $K$ iterations, while the outer minimization loop performs $I$ iterations, resulting in an overall computational complexity of $\mathcal{O}(K \cdot I)$.
>
> We additionally conducted a detailed empirical evaluation of wall-clock time, CPU and GPU memory usage for all baselines under identical hardware settings, as reported in Section D.4.6 (Table 6, Page 28). Some comparisons are summarized in the following table:
>
> | METHOD        | TIME(s) | MEMORY (MB)  | GPU MEMORY (MB) |
> |---------------|---------|--------------|-----------------|
> | GAS-DRO (DFU) | 1897    | 2274         | 12239           |
> | GAS-DRO (VAE) | 494     | 2449         | 5849            |
> | P-DRO          | 1799   | 2205         | 11576         |
> | KL-DRO        | 149     | 2314         | 417             |
> | W-DRO         | 2896    | 2304         | 417             |
> | ML            | 143     | 1352         | 33              |
>
> Thanks to its closed-form solution, KL-DRO incurs the lowest training overhead among the DRO baselines. In contrast, W-DRO exhibits the highest training time due to the inherent difficulty of searching in the Wasserstein-based ambiguity set, while maintaining a relatively low GPU memory footprint because it does not rely on generative models.
>
> Compared with standard empirical risk minimization (ML), both GAS-DRO (DFU) and P-DRO introduce substantial additional training-time overhead, primarily due to the sampling procedures required by diffusion models. Their memory consumption is also significantly higher, owing to the large U-Net backbones and the need to store sampling trajectories. GAS-DRO (VAE), which is based on a VAE, is more computationally efficient than diffusion-based DRO baselines.
> Overall, the additional computational cost of GAS-DRO (DFU) is justified by the substantial performance improvements it achieves.
>
> Since all baselines use the same predictive model during testing, they share identical inference runtime (0.1 s) and GPU memory usage (409 MB) when evaluated on a test set of 8,760 samples.

---

> ### Author Response · Authors · 2025-11-26
>
> We sincerely thank the reviewer for their time and thoughtful feedback, and we would greatly appreciate their consideration of our response. We would like to reiterate the key distinction from Michel et al. (2021): **Michel et al. (2021) relies on a traditional KL-divergence–based ambiguity set, which retains intrinsic support-mismatch issues and requires additional approximations, whereas our new ambiguity modeling overcomes the limitations of traditional ambiguity sets and is both theoretically justified and flexibly tractable**. We would be more than happy to address any remaining concerns the reviewer may have.

---

### Author Response · Authors · 2025-12-03

Dear Area Chairs,

We appreciate your efforts in leading the remaining review process.

During the rebuttal process, **Reviewer WYi2 has confirmed that our responses have adequately resolved the issues the reviewer pointed out. The reviewer remains confident in the quality of the work and keeps the initial positive evaluation**.
The other reviewers did not provide further feedback before the reverting, but we believe their concerns can be fully addressed if the discussion were to continue. Their major questions and our responses are summarized below.

**`Is the proposed reconstruction-loss-based ambiguity set an incremental contribution over traditional KL-based methods? ` (Reviewers u62m, 4i8h)**

The proposed method outperforms traditional KL-based methods in a principled manner. A key point missed by Reviewers u62m and 4i8h is that **the proposed reconstruction-loss-based ambiguity set provably constrains the reverse KL divergence $D_{\mathrm{KL}}(P_0 || P)$ (Lemma 1), rather than the KL divergence $D_{\mathrm{KL}}(P || P_0)$ commonly used in traditional KL-based methods such as Michel et al. (2021).** The KL divergence $D_{\mathrm{KL}}(P || P_0)$ in traditional methods requires the support of the adversarial distribution $P$ to be a subset of the support of the nominal distribution $P_0$, which restricts the ability of generative models to represent adversarial distributions and ultimately weakens robustness when the testing distributions have shifted support —a well-known **support mismatch issue** [1].
In contrast, the proposed ambiguity modeling allows the adversarial distribution $P$ to extend beyond the support of the nominal distribution $P_0$. This enables the generative model to capture worst-case distributions with shifted support, thereby improving robustness.


**`Distinctions compared to Michel et al. (2021)` (Reviewer u62m)**

Our method has clear distinctions from Michel et al. (2021): **Our method models new ambiguity sets directly in the parameterized space of generative models, whereas Michel et al. (2021) models adversarial distributions within traditional KL-based ambiguity sets.**

**Due to its reliance on traditional KL-based ambiguity sets, Michel et al. (2021) retains the intrinsic support-mismatch issues [1] mentioned above and requires the crude or biased approximations to obtain the key objective.**

By contrast, our method is both theoretically justified and flexibly tractable. The proposed ambiguity set is shown in Lemma 1 to constrain the reverse KL divergence, which, unlike the KL divergence in Michel et al. (2021), does not impose any restriction on the support of adversarial distributions. **This allows generative models to capture more expressive adversarial distributions, thereby overcoming the support-mismatch issue**.
Moreover, since the proposed ambiguity set directly constrains the generative model parameters, it is **directly tractable without additional approximations**, yielding a more accurate and theoretically justified solution.

Our contributions also include:
- A dual-learning method that provably identifies near-optimal worst-case distributions while enforcing a reverse KL divergence constraint (Theorem 1).
- Formal stable convergence guarantees for the proposed algorithm despite non-convex, non-concave min–max optimization (Theorem 2).
- Training objectives for diffusion models and VAEs derived based on policy optimization techniques.

**`Broad evaluations across other tasks` (Reviewers u62m, Focz)**

We include evaluations on image classification using the MNIST and USPS datasets (Section E, Table 7, Page 32), which confirm the superior performance of the proposed method on broader applications.

**`Computational complexity and measurement` (Reviewers u62m, Focz)**

We measured wall-clock time, CPU memory, and GPU memory for all baselines (Section D.4.6, Table 6, Page 29) and explained the computational complexity of all baselines.

**`What are the effects of the expressive capacity of generative models` (Reviewers Focz, WYi2)**

We demonstrate the impact of generative model expressivity by both theoretical analysis and empirical studies.

In Section C.4 (Page 23), we introduce the notion of $\Gamma$-expressivity (Definition 1) and establish Corollary 1, showing that the generative-model expressivity quantified by $\Gamma$ affects DRO performance by introducing an additional error that scales linearly with $\Gamma$.


Empirically, Section D.4.5 (Table 5, Page 29) shows that higher-capacity generative models consistently achieve better DRO performance, indicating that more expressive models can better represent the adversarial distributions and enhance robustness of the proposed method.



[1] Chen, R., & Paschalidis, I. C. (2020). Distributionally robust learning. Foundations and Trends® in Optimization, 4(1-2), 1-243.

---

### Meta-Review · Area_Chair_86j7 · 2025-12-12

**Summary:**

The paper studies Distributionally Robust Optimization using generative (VAEs & Diffusions) models to approximate/construct ambiguity sets, constraining the ELBO with reconstruction error bounds via Lagrange optimizers.   Reviewers mainly discussed novelty, relation to prior work, and experimental scope.  In particular, it was questioned whether the contribution is incremental over prior DRO methods.  But personally I believe that during  the rebuttal they succedded in comparing their approach w.r.t. the one in  Michel et al. (2021). Authors also included new experimetnal tasks (though simple MNIST, USPS) and evaluated computational running times. One of the strongest points is the convergence analysis of the GAS-DRO min max optimization method. Overall, given the initial scores and rebuttal, I believe this paper can be accepted. From a personal perspective, I believe it is interesting if the authors comment in the final version of the paper how the DRO approach is different from the methods implemented in adversarial training.

**Reviewer Concerns:**

**Concerns addressed by the rebuttal:**
- The novelty over KL-DRO and Wasserstein-DRO was clarified.
- The difference with Michel et al. (2021) was explained in detail.
- Additional experiments on image classification were added.
- Computational cost and runtime analysis were provided.
- Reviewer WYi2 confirmed that the rebuttal resolved their concerns.

**Concerns still outstanding:**
- In the opition of the AC, Reviewer 4i8h concerns were not directly addressed in a convincing way.

**Reviewer Scores:**

Reviewer u62m could increase the score from 2 to 4, or even to 6, after the clarifications.
Reviewer Focz (6) could slightly increase the score after seeing runtime results and extra tasks.
Reviewer WYi2 would remain at 8, as they already support the paper.
Reviewer 4i8h is unlikely to change the score (4), since the rebuttal does not directly answer the concerns.

This would give an average rating of 5.5-6, which may lay about the acceptance threshold. My personal assessment of the work goes in that direction.

---

### Decision · Program_Chairs · 2026-01-26

Accept (Poster)